# Increasing Both Batch Size and Learning Rate Accelerates Stochastic Gradient Descent

**Hikaru Umeda**                                                                                          *ee227115@meiji.ac.jp*
*Meiji University*

**Hideaki Iiduka**                                                                                          *iiduka@cs.meiji.ac.jp*
*Meiji University*

**Reviewed on OpenReview:** *https://openreview.net/forum?id=sbmp55k6iE*

## Abstract

The performance of mini-batch stochastic gradient descent (SGD) strongly depends on setting the batch size and learning rate to minimize the empirical loss in training the deep neural network. In this paper, we present theoretical analyses of mini-batch SGD with four schedulers: (i) constant batch size and decaying learning rate scheduler, (ii) increasing batch size and decaying learning rate scheduler, (iii) increasing batch size and increasing learning rate scheduler, and (iv) increasing batch size and warm-up decaying learning rate scheduler. We show that mini-batch SGD using scheduler (i) does not always minimize the expectation of the full gradient norm of the empirical loss, whereas it does using any of schedulers (ii), (iii), and (iv). Furthermore, schedulers (iii) and (iv) accelerate mini-batch SGD. The paper also provides numerical results of supporting analyses showing that using scheduler (iii) or (iv) minimizes the full gradient norm of the empirical loss faster than using scheduler (i) or (ii).

## 1 Introduction

Mini-batch stochastic gradient descent (SGD) (Robbins & Monro, 1951; Zinkevich, 2003; Nemirovski et al., 2009; Ghadimi & Lan, 2012; 2013) is a simple and useful deep-learning optimizer for finding appropriate parameters of a deep neural network (DNN) in the sense of minimizing the empirical loss defined by the mean of nonconvex loss functions corresponding to the training set.

The performance of mini-batch SGD strongly depends on how the batch size and learning rate are set. In particular, increasing batch size (Byrd et al., 2012; Balles et al., 2016; De et al., 2017; Smith et al., 2018; Goyal et al., 2018; Shallue et al., 2019; Zhang et al., 2019) is useful for training DNNs with mini-batch SGD. In (Smith et al., 2018), it was numerically shown that using an enormous batch size leads to a reduction in the number of parameter updates.

Decaying a learning rate (Wu et al., 2014; Ioffe & Szegedy, 2015; Loshchilov & Hutter, 2017; Hundt et al., 2019) is also useful for training DNNs with mini-batch SGD. In (Chen et al., 2020), theoretical results indicated that running SGD with a diminishing learning rate $\eta_t = O(\frac{1}{t})$ and a large batch size for sufficiently many steps leads to convergence to a stationary point. A practical example of a decaying learning rate with $\eta_{t+1} \le \eta_t$ for all $t \in \mathbb{N}$ is a constant learning rate $\eta_t = \eta > 0$ for all $t \in \mathbb{N}$. However, convergence of SGD with a constant learning rate is not guaranteed (Scaman & Malherbe, 2020). Other practical learning rates have been presented for training DNNs, including cosine annealing (Loshchilov & Hutter, 2017), cosine power annealing (Hundt et al., 2019), step decay (Lu, 2024), exponential decay (Wu et al., 2014), polynomial decay (Chen et al., 2018), and linear decay (Liu et al., 2020).

**Contribution:** The main contribution of the present paper is its theoretical analyses of mini-batch SGD with batch size and learning rate schedulers used in practice satisfying the following inequality:

$$
\min_{t\in[0:T-1]} \mathbb{E}\left[\|\nabla f(\boldsymbol{\theta}_t)\|\right] \leq \left\{ \frac{2(f(\boldsymbol{\theta}_0) - \underline{f^\star})}{2 - \bar{L}\eta_{\max}} \underbrace{\frac{1}{\sum_{t=0}^{T-1}\eta_t}}_{B_T} + \frac{\bar{L}\sigma^2}{2 - \bar{L}\eta_{\max}} \underbrace{\frac{1}{\sum_{t=0}^{T-1}\eta_t}\sum_{t=0}^{T-1}\frac{\eta_t^2}{b_t}}_{V_T} \right\}^{\frac{1}{2}},
$$

where $f$ is the empirical loss for $n$ training samples having $\bar{L}$-Lipschitz continuous gradient $\nabla f$ and lower bound $\underline{f^\star}$, $\sigma^2$ is an upper bound on the variance of the mini-batch stochastic gradient, and $(\boldsymbol{\theta}_t)_{t=0}^{T-1}$ is the sequence generated by mini-batch SGD with batch size $b_t$, learning rate $\eta_t \in [\eta_{\min}, \eta_{\max}] \subset [0, \frac{2}{\bar{L}})$, and total number of steps to train a DNN $T$.

| Scheduler | $B_T$ | $V_T$ | $O(\sqrt{B_T + V_T})$ |
|---|---|---|---|
| **Case (i) (Theorem 3.1; Section 3.1)** $b_t$: Constant; $\eta_t$: Decay | $\dfrac{H_1}{T}$ | $\dfrac{H_2}{b} + \dfrac{H_7}{bT}$ | $O\left(\sqrt{\dfrac{1}{T} + \dfrac{1}{b}}\right)$ |
| **Case (ii) (Theorem 3.2; Section 3.2)** $b_t$: Increase; $\eta_t$: Decay | $\dfrac{H_3}{T}$ | $\dfrac{H_4}{b_0 T}$ | $O\left(\dfrac{1}{\sqrt{T}}\right)$, $O\left(\dfrac{1}{\sqrt{M}}\right)$ |
| **Case (iii) (Theorem 3.3; Section 3.3)** $b_t$: Increase; $\eta_t$: Increase | $\dfrac{H_5}{\gamma^M}$ | $\dfrac{H_6}{b_0 \gamma^M}$ | $O\left(\dfrac{1}{\gamma^{\frac{M}{2}}}\right)$ $^{(*)\ \exists \bar{m}\forall M \geq \bar{m}}_{\frac{1}{\gamma^{\frac{M}{2}}} < \frac{1}{\sqrt{M}}}$ |
| **Case (iv) (Theorem 3.4; Section 3.4)** $b_t$: Increase; $\eta_t$: Increase $\to$ Decay | $\dfrac{H_5}{\gamma^M} \to \dfrac{H_3}{T}$ | $\dfrac{H_6}{b_0 \gamma^M} \to \dfrac{H_4}{b_0 T}$ | $O\left(\dfrac{1}{\gamma^{\frac{M}{2}}}\right) \to O\left(\dfrac{1}{\sqrt{T}}\right)$ |

$H_i$ ($i \in [6]$) (resp. $H_7$) is a positive (resp. nonnegative) number depending on $\eta_{\min}$ and $\eta_{\max}$. $\gamma$ and $\delta$ are such that $1 < \gamma^2 < \delta$ (e.g., $\delta = 2$ when batch size is doubly increasing every $E$ epochs). The total number of steps when batch size increases $M$ times is $T(M) = \sum_{m=0}^{M}\lceil\frac{n}{b_m}\rceil E \geq ME$. $O(\frac{1}{\gamma^{\frac{M}{2}}}) \to O(\frac{1}{\sqrt{T}})$ implies that the convergence rate changes from $O(\frac{1}{\gamma^{\frac{M}{2}}})$ to $O(\frac{1}{\sqrt{T}})$ when the learning rate $\eta_t$ changes from an increasing learning rate to a decaying learning rate ($\eta_t$: Increase $\to$ Decay).

**(i) Using constant batch size $b_t = b$ and decaying learning rate $\eta_t$ (Theorem 3.1; Section 3.1):** Using a constant batch size and practical decaying learning rates, such as constant, cosine-annealing, and polynomial decay learning rates, satisfies that, for a sufficiently large step $T$, the upper bound on $\min_{t\in[0:T-1]} \mathbb{E}[\|\nabla f(\boldsymbol{\theta}_t)\|]$ becomes approximately $O(\frac{1}{\sqrt{b}}) > 0$. This result provides an upper bound on the best iterate within $T$ iterations and characterizes how the expected gradient norm scales with the batch size $b$ under the considered learning rate schedules. However, this bound does not necessarily mean that mini-batch SGD does not converge to a stationary point in the long run, as it does not characterize the last iterate. Meanwhile, the analysis indicates that using the cosine-annealing and polynomial decay learning rates would decrease $\mathbb{E}[\|\nabla f(\boldsymbol{\theta}_t)\|]$ faster than using a constant learning rate (see (8)), which is supported by the numerical results in **Figure 1**.

**(ii) Using increasing batch size $b_t$ and decaying learning rate $\eta_t$ (Theorem 3.2; Section 3.2):** Although convergence analyses of SGD were presented in (Vaswani et al., 2019; Fehrman et al., 2020; Scaman & Malherbe, 2020; Loizou et al., 2021; Wang et al., 2021; Khaled & Richtárik, 2023), providing the theoretical performance of mini-batch SGD with increasing batch sizes that have been used in practice may not be sufficient. The present paper shows that mini-batch SGD has an $O(\frac{1}{\sqrt{T}})$ rate of convergence, where $T$ represents the total number of optimization steps, and $M$ represents the number of times the batch size is increased during training. Since the total number of training steps when batch size increases $M$ times satisfies $T(M) = \sum_{m=0}^{M}\lceil\frac{n}{b_m}\rceil E \geq ME$, the convergence rates can be analyzed either in terms of $T$ or $M$, giving $O(\frac{1}{\sqrt{T}})$ or $O(\frac{1}{\sqrt{M}})$, respectively. This result highlights that increasing batch sizes affects the variance

reduction in mini-batch SGD. By analyzing the convergence rate in terms of $M$ instead of $T$, we are able to make a more direct comparison with Case (iii), where the geometric batch size scheduling achieves an $O(\frac{1}{\gamma^{\frac{M}{2}}})$ convergence rate. Increasing batch size every $E$ epochs makes the polynomial decay and linear learning rates become small at an early stage of training (**Figure 2**(a)). Meanwhile, the cosine-annealing and constant learning rates are robust to increasing batch sizes (**Figure 2**(a)). Hence, it is desirable for mini-batch SGD using increasing batch sizes to use the cosine-annealing and constant learning rates, which is supported by the numerical results in **Figure 2**.

**(iii) Using increasing batch size $b_t$ and increasing learning rate $\eta_t$ (Theorem 3.3; Section 3.3):** From Case (ii), when batch sizes increase, keeping learning rates large is useful for training DNNs. Hence, we are interested in verifying whether mini-batch SGD with both the batch sizes and learning rates increasing can train DNNs. Let us consider a scheduler doubly increasing batch size (i.e., $\delta = 2$). We set $\gamma > 1$ such that $\gamma < \sqrt{\delta} = \sqrt{2}$ and we set an increasing learning rate scheduler such that the learning rate is multiplied by $\gamma$ every $E$ epochs (**Figure 3**(a)). This paper shows that, when batch size increases $M$ times, mini-batch SGD has an $O(\frac{1}{\gamma^{\frac{M}{2}}})$ convergence rate that is better than the $O(\frac{1}{\sqrt{M}})$ convergence rate in Case (ii). That is, *increasing both batch size and learning rate accelerates mini-batch SGD*. We give practical results (**Figure 3**(b); $\delta = 2$ and **Figures 5**, **8**, **9**(b); $\delta = 3, 4$) such that Case (iii) decreases $\|\nabla f(\boldsymbol{\theta}_t)\|$ faster than Case (ii) and tripling and quadrupling batch sizes ($\delta = 3, 4$) decrease $\|\nabla f(\boldsymbol{\theta}_t)\|$ faster than doubly increasing batch sizes ($\delta = 2$). The intuition for why increasing batch size and learning rate can provide fast convergence is as follows:

(1) Increasing batch size decreases the variance of the stochastic gradient, since the upper bound on the variance is inversely proportional to the batch size (see also Proposition A.1). Hence, increasing batch size improves convergence to stationary points of the empirical loss $f$. This fact is based on Case (i) that the upper bound of $\min_{t \in [0:T-1]} \mathbb{E}[\|\nabla f(\boldsymbol{\theta}_t)\|]$ is $O(\sqrt{\frac{1}{T} + \frac{1}{b}})$, where $b$ is the batch size.

(2) Mini-batch SGD does not work when learning rates are small at an early stage of training. This fact is supported by the numerical results in **Figure 1** (Case (i)) indicating that using the decaying learning rate $\eta_t = O(\frac{1}{\sqrt{t+1}})$ does not train a DNN. Hence, keeping learning rates large implies that SGD works well.

(3) From (1) and (2), increasing batch size and learning rate can provide fast convergence of SGD.

Here, let us compare Case (ii) with Case (iii). For simplicity, we use a scheduler tripling batch size, i.e., $b_t$ is multiplied by $\delta = 3$ at each step and consider Case (ii) with a constant learning rate $\eta_t = \eta$ satisfying $\eta_{t+1} \le \eta_t$ ($t \in \{0\} \cup \mathbb{N}$) and Case (iii) with the learning rate $\eta_t$ that is multiplied by $\gamma$ ($< \sqrt{\delta} = \sqrt{3}$) at each step. $B_T$ in Case (ii) is given by

$$B_T = \frac{1}{\sum_{t=0}^{T-1} \eta_t} = \frac{1}{\sum_{t=0}^{T-1} \eta} = O\left(\frac{1}{T}\right),$$

while $B_T$ in Case (iii) is given by

$$B_T = \frac{1}{\sum_{t=0}^{T-1} \eta_t} = \frac{1}{\sum_{t=0}^{T-1} \gamma^t \eta_0} = \frac{1}{\eta_0 \frac{\gamma^T - 1}{\gamma - 1}} = O\left(\frac{1}{\gamma^T}\right).$$

$V_T$ in Case (ii) is given by

$$V_T = \frac{1}{\sum_{t=0}^{T-1} \eta_t} \sum_{t=0}^{T-1} \frac{\eta_t^2}{b_t} = \frac{1}{\sum_{t=0}^{T-1} \eta} \sum_{t=0}^{T-1} \frac{\eta^2}{b_t} = \frac{\eta}{T} \sum_{t=0}^{T-1} \frac{1}{\delta^t b_0} \le \frac{\eta}{b_0 T} \frac{1}{1 - \frac{1}{\delta}} = O\left(\frac{1}{T}\right).$$

Meanwhile, $V_T$ in Case (iii) is given by

$$V_T = \frac{1}{\sum_{t=0}^{T-1} \eta_t} \sum_{t=0}^{T-1} \frac{\eta_t^2}{b_t} = \frac{1}{\sum_{t=0}^{T-1} \eta_t} \sum_{t=0}^{T-1} \frac{\gamma^{2t} \eta_0^2}{\delta^t b_0} = \frac{1}{\eta_0 \frac{\gamma^T - 1}{\gamma - 1}} \frac{\eta_0^2}{b_0} \sum_{t=0}^{T-1} \left(\frac{\gamma^2}{\delta}\right)^t \le \frac{1}{\frac{\gamma^T - 1}{\gamma - 1}} \frac{\eta_0}{b_0} \frac{1}{1 - \frac{\gamma^2}{\delta}} = O\left(\frac{1}{\gamma^T}\right),$$

where $\gamma^2 < \delta$ is used to guarantee $\sum_{t=0}^{+\infty} (\frac{\gamma^2}{\delta})^t < +\infty$. Therefore, Case (iii) has $\min_{t \in [0:T-1]} \mathbb{E}[\|\nabla f(\boldsymbol{\theta}_t)\|] = O(\frac{1}{\gamma^{\frac{T}{2}}})$, which is better than Case (ii) with $\min_{t \in [0:T-1]} \mathbb{E}[\|\nabla f(\boldsymbol{\theta}_t)\|] = O(\frac{1}{\sqrt{T}})$.

**(iv) Using increasing batch size $b_t$ and warm-up decaying learning rate $\eta_t$ (Theorem 3.4; Section 3.4):** One way to guarantee fast convergence of mini-batch SGD with increasing batch sizes is to increase learning rates (acceleration period; Case (iii)) during the first epochs and then decay the learning rates (convergence period; Case (ii)), that is, to use a decaying learning rate with warm-up (He et al., 2016; Vaswani et al., 2017; Goyal et al., 2018; Gotmare et al., 2019; He et al., 2019). We give numerical results (**Figure 4**; $\delta = 2$ and **Figure 10**; $\delta = 3$) indicating that using mini-batch SGD with increasing batch sizes and decaying learning rates with a warm-up minimizes $\|\nabla f(\boldsymbol{\theta}_t)\|$ faster than using a constant learning rate in Case (ii) or increasing learning rates in Case (iii). Our results are numerically supported by the previous results reported in (He et al., 2016; Goyal et al., 2018; Gotmare et al., 2019; He et al., 2019) indicating that a warm-up learning rate is useful for training deep neural networks, such as ResNets and Transformer networks (Vaswani et al., 2017).

## 2 Mini-batch SGD for Empirical Risk Minimization

### 2.1 Empirical Risk Minimization

Let $\boldsymbol{\theta} \in \mathbb{R}^d$ be a parameter of a deep neural network; let $S = \{(\boldsymbol{x}_1, \boldsymbol{y}_1), \ldots, (\boldsymbol{x}_n, \boldsymbol{y}_n)\}$ be the training set, where data point $\boldsymbol{x}_i$ is associated with label $\boldsymbol{y}_i$; and let $f_i(\cdot) := f(\cdot; (\boldsymbol{x}_i, \boldsymbol{y}_i)) \colon \mathbb{R}^d \to \mathbb{R}_+$ be the loss function corresponding to the $i$-th labeled training data $(\boldsymbol{x}_i, \boldsymbol{y}_i)$. Empirical risk minimization (ERM) minimizes the empirical loss defined for all $\boldsymbol{\theta} \in \mathbb{R}^d$ as $f(\boldsymbol{\theta}) = \frac{1}{n} \sum_{i \in [n]} f_i(\boldsymbol{\theta})$. This paper considers the following stationary point problem: Find $\boldsymbol{\theta}^\star \in \mathbb{R}^d$ such that $\nabla f(\boldsymbol{\theta}^\star) = \mathbf{0}$.

We assume that the loss functions $f_i$ ($i \in [n]$) satisfy the conditions in the following assumption (see Appendix A for definitions of functions, mappings, and notation used in this paper).

**Assumption 2.1** *Let $n$ be the number of training samples and let $L_i > 0$ ($i \in [n]$).*

(A1) *$f_i \colon \mathbb{R}^d \to \mathbb{R}$ ($i \in [n]$) is differentiable and $L_i$-smooth, and $f_i^\star := \inf\{f_i(\boldsymbol{\theta}) \colon \boldsymbol{\theta} \in \mathbb{R}^d\} \in \mathbb{R}$.*

(A2) *Let $\xi$ be a random variable that is independent of $\boldsymbol{\theta} \in \mathbb{R}^d$. $\nabla f_\xi \colon \mathbb{R}^d \to \mathbb{R}^d$ is the stochastic gradient of $\nabla f$ such that (i) for all $\boldsymbol{\theta} \in \mathbb{R}^d$, $\mathbb{E}_\xi[\nabla f_\xi(\boldsymbol{\theta})] = \nabla f(\boldsymbol{\theta})$ and (ii) there exists $\sigma \geq 0$ such that, for all $\boldsymbol{\theta} \in \mathbb{R}^d$, $\mathbb{V}_\xi[\nabla f_\xi(\boldsymbol{\theta})] = \mathbb{E}_\xi[\|\nabla f_\xi(\boldsymbol{\theta}) - \nabla f(\boldsymbol{\theta})\|^2] \leq \sigma^2$, where $\mathbb{E}_\xi[\cdot]$ denotes expectation with respect to $\xi$.*

(A3) *Let $b \in \mathbb{N}$ such that $b \leq n$; and let $\boldsymbol{\xi} = (\xi_1, \xi_2, \cdots, \xi_b)^\top$ comprise $b$ independent and identically distributed variables and be independent of $\boldsymbol{\theta} \in \mathbb{R}^d$. The full gradient $\nabla f(\boldsymbol{\theta})$ is estimated as the following mini-batch gradient at $\boldsymbol{\theta}$: $\nabla f_B(\boldsymbol{\theta}) := \frac{1}{b} \sum_{i=1}^b \nabla f_{\xi_i}(\boldsymbol{\theta})$.*

The $L_i$-smoothness of $f_i$ in Assumption (A1) is used to analyze mini-batch SGD (Garrigos & Gower, 2024, Assumption 4.3), since almost all of the analyses of mini-batch SGD have been based on the descent lemma (Beck, 2017, Lemma 5.7) that is satisfied under smoothness of $f_i$. If $f_i^\star := \inf\{f_i(\boldsymbol{\theta}) \colon \boldsymbol{\theta} \in \mathbb{R}^d\} = -\infty$ holds, then the loss function $f_i$ corresponding to the $i$-th labeled training data $(\boldsymbol{x}_i, \boldsymbol{y}_i)$ does not have any global minimizer, which implies that the empirical loss $f$ satisfies $f^\star := \inf\{f(\boldsymbol{\theta}) \colon \boldsymbol{\theta} \in \mathbb{R}^d\} = -\infty$. Hence, the interpolation property (Garrigos & Gower, 2024, Section 4.3.1) (i.e., there exists $\boldsymbol{\theta}^\star \in \mathbb{R}^d$ such that, for all $i \in [n]$, $f_i(\boldsymbol{\theta}^\star) = f_i^\star \in \mathbb{R}$) does not hold, whereas the interpolation property does hold for optimization of a linear model with the squared hinge loss for binary classification on linearly separable data (Vaswani et al., 2019, Section 2). Moreover, in the case where $f$ is convex with $f^\star = -\infty$, there are no stationary points of $f$, which implies that no algorithm ever finds stationary points of $f$. Accordingly, the condition $f_i^\star := \inf\{f_i(\boldsymbol{\theta}) \colon \boldsymbol{\theta} \in \mathbb{R}^d\} \in \mathbb{R}$ in (A1) is natural under training DNNs including the case where the empirical loss $f$ is the cross-entropy with $\boldsymbol{\theta}^\star \in \mathbb{R}^d$ such that $f(\boldsymbol{\theta}^\star) = \inf\{f(\boldsymbol{\theta}) \colon \boldsymbol{\theta} \in \mathbb{R}^d\} = 0$. Assumption (A2) is satisfied when (A1) holds and the random variable $\xi$ follows the uniform distribution that is used to train DNNs in practice (see Appendix A.1 for details). Assumption (A3) holds under sampling with replacement (see Appendix A.2 for details).

## 2.2 Mini-batch SGD

Given the $t$-th approximated parameter $\boldsymbol{\theta}_t \in \mathbb{R}^d$ of the deep neural network, mini-batch SGD uses $b_t$ loss functions $f_{\xi_{t,1}}, f_{\xi_{t,2}}, \cdots, f_{\xi_{t,b_t}}$ randomly chosen from $\{f_1, f_2, \cdots, f_n\}$ at each step $t$, where $\boldsymbol{\xi}_t = (\xi_{t,1}, \xi_{t,2}, \cdots, \xi_{t,b_t})^\top$ is independent of $\boldsymbol{\theta}_t$ and $b_t$ is a batch size satisfying $b_t \leq n$. The pseudo-code of the algorithm is shown as Algorithm 1.

---
**Algorithm 1** Mini-batch SGD algorithm
---
**Require:** $\boldsymbol{\theta}_0 \in \mathbb{R}^d$ (initial point), $b_t > 0$ (batch size), $\eta_t > 0$ (learning rate), $T \geq 1$ (steps)
**Ensure:** $(\boldsymbol{\theta}_t) \subset \mathbb{R}^d$
1: **for** $t = 0, 1, \ldots, T-1$ **do**
2:     $\nabla f_{B_t}(\boldsymbol{\theta}_t) := \frac{1}{b_t} \sum_{i=1}^{b_t} \nabla f_{\xi_{t,i}}(\boldsymbol{\theta}_t)$
3:     $\boldsymbol{\theta}_{t+1} := \boldsymbol{\theta}_t - \eta_t \nabla f_{B_t}(\boldsymbol{\theta}_t)$
4: **end for**

---

The following lemma can be proved using Proposition A.1, Assumption 2.1, and the descent lemma (Beck, 2017, Lemma 5.7): for all $\boldsymbol{\theta}_1, \boldsymbol{\theta}_2 \in \mathbb{R}^d$, $f(\boldsymbol{\theta}_2) \leq f(\boldsymbol{\theta}_1) + \langle \nabla f(\boldsymbol{\theta}_1), \boldsymbol{\theta}_2 - \boldsymbol{\theta}_1 \rangle + \frac{\bar{L}}{2}\|\boldsymbol{\theta}_2 - \boldsymbol{\theta}_1\|^2$, where Assumption 2.1(A1) ensures that $f$ is $\bar{L}$-smooth ($\bar{L} := \frac{1}{n}\sum_{i\in[n]} L_i$).

**Lemma 2.1** *Suppose that Assumption 2.1 holds and consider the sequence $(\boldsymbol{\theta}_t)$ generated by Algorithm 1 with $\eta_t \in [\eta_{\min}, \eta_{\max}] \subset [0, \frac{2}{\bar{L}})$ satisfying $\sum_{t=0}^{T-1} \eta_t \neq 0$, where $\bar{L} := \frac{1}{n}\sum_{i\in[n]} L_i$ and $\underline{f^\star} := \frac{1}{n}\sum_{i\in[n]} f_i^\star$. Then, for all $T \in \mathbb{N}$,*

$$\min_{t\in[0:T-1]} \mathbb{E}\left[\|\nabla f(\boldsymbol{\theta}_t)\|^2\right] \leq \frac{2(f(\boldsymbol{\theta}_0) - \underline{f^\star})}{2 - \bar{L}\eta_{\max}} \frac{1}{\sum_{t=0}^{T-1} \eta_t} + \frac{\bar{L}\sigma^2}{2 - \bar{L}\eta_{\max}} \frac{\sum_{t=0}^{T-1} \eta_t^2 b_t^{-1}}{\sum_{t=0}^{T-1} \eta_t},$$

*where $\mathbb{E}$ denotes the total expectation, defined by $\mathbb{E} := \mathbb{E}_{\boldsymbol{\xi}_0}\mathbb{E}_{\boldsymbol{\xi}_1}\cdots\mathbb{E}_{\boldsymbol{\xi}_t}$.*

The proof of Lemma 2.1 depends on the following standard inequality (Garrigos & Gower, 2024, (27)) from the literature on SGD analysis (Garrigos & Gower, 2024, Section 5.4, Theorem 5.12) using the descent lemma:

$$\min_{t\in[0:T-1]} \mathbb{E}\left[\|\nabla f(\boldsymbol{\theta}_t)\|^2\right] \leq \frac{f(\boldsymbol{\theta}_0) - \underline{f^\star}}{\eta \sum_{t=0}^{T-1} \alpha_t} + \eta \bar{L} L_{\max} \Delta_f^*, \tag{1}$$

where $(\alpha_t)$ is defined by $\alpha_t(1 + \eta^2 \bar{L} L_{\max}) = \alpha_{t-1}$, $L_{\max} := \max_{i\in[n]} L_i$, $f^\star$ is the optimal value of $f$ over $\mathbb{R}^d$, and $\Delta_f^* := f^\star - \underline{f^\star}$. While the existing approach (Garrigos & Gower, 2024) uses a sequence $(\alpha_t)$ satisfying $\sum_{t=0}^{T-1} \alpha_t \geq \frac{1}{2\eta^2 \bar{L} L_{\max}}$, the present paper uses a learning rate $\eta_t$ satisfying $\sum_{t=0}^{T-1} \eta_t \geq O(T)$ and an increasing batch size $b_t$ satisfying $\sum_{t=0}^{T-1} \frac{1}{b_t} < +\infty$ in Lemma 2.1. As a result, mini-batch SGD with $\eta > 0$ and the increasing batch size $b_t$ satisfy that $\min_{t\in[0:T-1]} \mathbb{E}[\|\nabla f(\boldsymbol{\theta}_t)\|] = O(\frac{1}{\sqrt{T}})$ (Theorem 3.2), which is better than the existing result that SGD with $\eta = \sqrt{\frac{2}{\bar{L} L_{\max} T}}$ satisfies $\min_{t\in[0:T-1]} \mathbb{E}[\|\nabla f(\boldsymbol{\theta}_t)\|] = O(\frac{1}{T^{\frac{1}{4}}})$ (Garrigos & Gower, 2024, Theorem 5.12). Section 3.5 compares our results with the existing ones in detail.

The theorems in the present paper are based on Lemma 2.1. Here, we sketch a proof of Lemma 2.1. The descent lemma under (A1) and the definition of $\boldsymbol{\theta}_{t+1}$ imply the following inequality:

$$f(\boldsymbol{\theta}_{t+1}) \leq f(\boldsymbol{\theta}_t) - \eta_t \langle \nabla f(\boldsymbol{\theta}_t), \nabla f_{B_t}(\boldsymbol{\theta}_t) \rangle + \frac{\bar{L}\eta_t^2}{2} \|\nabla f_{B_t}(\boldsymbol{\theta}_t)\|^2.$$

Under (A2), the mini-batch gradient $\nabla f_{B_t}(\boldsymbol{\theta}_t)$ is an unbiased estimator of $\nabla f(\boldsymbol{\theta}_t)$, i.e., $\mathbb{E}[\nabla f_{B_t}(\boldsymbol{\theta}_t)] = \nabla f(\boldsymbol{\theta}_t)$, and the upper bound on the variance of $\nabla f_{B_t}(\boldsymbol{\theta}_t)$ is inversely proportional to the batch size $b_t$, i.e., $\mathbb{V}[\nabla f_{B_t}(\boldsymbol{\theta}_t)] \leq \frac{\sigma^2}{b_t}$ (see Proposition A.1 for details). Using the properties of $\nabla f_{B_t}(\boldsymbol{\theta}_t)$, the above inequality leads to the following:

$$\mathbb{E}[f(\boldsymbol{\theta}_{t+1})] \leq \mathbb{E}[f(\boldsymbol{\theta}_t)] - \eta_t \mathbb{E}\left[\|\nabla f(\boldsymbol{\theta}_t)\|^2\right] + \frac{\bar{L}\eta_t^2}{2}\left(\frac{\sigma^2}{b_t} + \mathbb{E}\left[\|\nabla f(\boldsymbol{\theta}_t)\|^2\right]\right).$$

Finally, summing the above inequality from $t = 0$ to $t = T - 1$, together with $\underline{f}^\star \in \mathbb{R}$ under (A1) and $\sum_{t=0}^{T-1} \eta_t \neq 0$, leads to the assertion in Lemma 2.1. A detailed proof is given in Appendix A.2.

## 3 Convergence Analysis of Mini-batch SGD

### 3.1 Constant Batch Size and Decaying Learning Rate Scheduler

This section considers a constant batch size and a decaying learning rate:

$$b_t = b \ (t \in \mathbb{N}) \quad \text{and} \quad \eta_{t+1} \leq \eta_t \ (t \in \mathbb{N}). \tag{2}$$

Let $p > 0$ and $T, E \in \mathbb{N}$; and let $\eta_{\min}$ and $\eta_{\max}$ satisfy $0 \leq \eta_{\min} \leq \eta_{\max}$. Examples of decaying learning rates are as follows: for all $t \in [0 : T]$,

$$[\text{Constant LR}] \ \eta_t = \eta_{\max}, \tag{3}$$

$$[\text{Diminishing LR}] \ \eta_t = \frac{\eta_{\max}}{\sqrt{t+1}}, \tag{4}$$

$$[\text{Cosine-annealing LR}] \ \eta_t = \eta_{\min} + \frac{\eta_{\max} - \eta_{\min}}{2} \left( 1 + \cos \left\lfloor \frac{t}{K} \right\rfloor \frac{\pi}{E} \right), \tag{5}$$

$$[\text{Polynomial decay LR}] \ \eta_t = (\eta_{\max} - \eta_{\min}) \left( 1 - \frac{t}{T} \right)^p + \eta_{\min}, \tag{6}$$

where $K = \lceil \frac{n}{b} \rceil$ is the number of steps per epoch, $E$ is the total number of epochs, and the number of steps $T$ in (5) is given by $T = KE$. A simple, practical decaying learning rate is the constant learning rate defined by (3). A decaying learning rate used in theoretical analyses of deep-learning optimizers is the diminishing learning rate defined by (4). The cosine-annealing learning rate defined by (5) and the linear learning rate defined by (6) with $p = 1$ (i.e., an example of a polynomial decay learning rate) are used in practice. Note that the cosine-annealing learning rate is updated each epoch, whereas the polynomial decay learning rate is updated each step.

Lemma 2.1 leads to the following (the proof of the theorem is given in Appendix A.3).

**Theorem 3.1 (Upper bound on $\min_t \mathbb{E} \|\nabla f(\boldsymbol{\theta}_t)\|^2$ for SGD using (2))** *Under the assumptions in Lemma 2.1, Algorithm 1 using (2) satisfies that, for all $T \in \mathbb{N}$,*

$$\min_{t \in [0:T-1]} \mathbb{E} \left[ \|\nabla f(\boldsymbol{\theta}_t)\|^2 \right] \leq \frac{2(f(\boldsymbol{\theta}_0) - \underline{f}^\star)}{2 - \bar{L}\eta_{\max}} \underbrace{\frac{1}{\sum_{t=0}^{T-1} \eta_t}}_{B_T} + \frac{\bar{L}\sigma^2}{2 - \bar{L}\eta_{\max}} \underbrace{\frac{\sum_{t=0}^{T-1} \eta_t^2}{b \sum_{t=0}^{T-1} \eta_t}}_{V_T},$$

*where $p, \eta_{\min}, \eta_{\max}, K$, and $E$ are the parameters used in (3)–(6), $T = KE = \lceil \frac{n}{b} \rceil E$ for Cosine LR (5),*

$$B_T \leq \begin{cases} \dfrac{1}{\eta_{\max} T} & [\text{Constant LR (3)}] \\[2mm] \dfrac{1}{2\eta_{\max}(\sqrt{T+1} - 1)} & [\text{Diminishing LR (4)}] \\[2mm] \dfrac{2}{(\eta_{\min} + \eta_{\max})T} & [\text{Cosine LR (5)}] \\[2mm] \dfrac{p+1}{(p\eta_{\min} + \eta_{\max})T} & [\text{Polynomial LR (6)}], \end{cases} \tag{7}$$

$$V_T \leq \begin{cases} \dfrac{\eta_{\max}}{b} & [\text{Constant LR (3)}] \\[2mm] \dfrac{\eta_{\max}(1 + \log T)}{2b(\sqrt{T+1} - 1)} & [\text{Diminishing LR (4)}] \\[2mm] \dfrac{3\eta_{\min}^2 + 2\eta_{\min}\eta_{\max} + 3\eta_{\max}^2}{4(\eta_{\min} + \eta_{\max})b} + \dfrac{(\eta_{\max} - \eta_{\min})K}{bT} & [\text{Cosine LR (5)}] \\[2mm] \dfrac{2p^2\eta_{\min}^2 + 2p\eta_{\min}\eta_{\max} + (p+1)\eta_{\max}^2}{(2p+1)(p\eta_{\min} + \eta_{\max})b} + \dfrac{(p+1)(\eta_{\max}^2 - \eta_{\min}^2)}{(p\eta_{\min} + \eta_{\max})bT} & [\text{Polynomial LR (6)}]. \end{cases}$$

Let us consider using Constant LR (3), Cosine LR (5), or Polynomial LR (6). Theorem 3.1 indicates that the bias term including $B_T$ converges to 0 as $O(\frac{1}{T})$, whereas the variance term including $V_T$ does not always converge to 0. Hence, the upper bound on $\min_{t \in [0:T-1]} \mathbb{E}[\|\nabla f(\boldsymbol{\theta}_t)\|^2]$ does not converge to 0. In fact, Theorem 3.1 with $\eta = \eta_{\max}$ and $\eta_{\min} = 0$ implies that

$$\limsup_{T \to +\infty} \min_{t \in [0:T-1]} \mathbb{E}\left[\|\nabla f(\boldsymbol{\theta}_t)\|^2\right] \leq \frac{\bar{L}\sigma^2}{(2 - \bar{L}\eta)b} \times \begin{cases} \eta & \text{[Constant LR (3)]} \\ \dfrac{3\eta}{4} & \text{[Cosine LR (5)]} \\ \dfrac{(p+1)\eta}{(2p+1)} & \text{[Polynomial LR (6)].} \end{cases} \tag{8}$$

Since $\frac{3\eta}{4} < \eta$ and $\frac{(p+1)\eta}{(2p+1)} < \eta$ ($p > 0$), using the cosine-annealing learning rate or the polynomial decay learning rate is better than using the constant learning rate in the sense of minimizing the upper bound on $\min_{t \in [0:T-1]} \mathbb{E}[\|\nabla f(\boldsymbol{\theta}_t)\|^2]$. Theorem 3.1 also indicates that Algorithm 1 using Diminishing LR (4) converges to 0 with the convergence rate $\min_{t \in [0:T-1]} \mathbb{E}[\|\nabla f(\boldsymbol{\theta}_t)\|] = O(\frac{\sqrt{\log T}}{T^{\frac{1}{4}}})$. However, since Diminishing LR (4) defined by $\eta_t = \frac{\eta}{\sqrt{t+1}}$ decays rapidly (see Figure 1(a)), it would not be useful for training DNNs in practice.

### 3.2 Increasing Batch Size and Decaying Learning Rate Scheduler

An increasing batch size is used to train DNNs in practice (Byrd et al., 2012; Balles et al., 2016; De et al., 2017; Smith et al., 2018; Goyal et al., 2018). This section considers an increasing batch size and a decaying learning rate following one of (3)–(6):

$$b_t \leq b_{t+1} \ (t \in \mathbb{N}) \quad \text{and} \quad \eta_{t+1} \leq \eta_t \ (t \in \mathbb{N}). \tag{9}$$

Examples of $b_t$ are, for example, for all $m \in [0:M]$ and all $t \in S_m = \mathbb{N} \cap [\sum_{k=0}^{m-1} K_k E_k, \sum_{k=0}^{m} K_k E_k)$ ($S_0 := \mathbb{N} \cap [0, K_0 E_0)$),

$$[\text{Polynomial growth BS}] \ b_t = \left( am \left\lceil \frac{t}{\sum_{k=0}^{m} K_k E_k} \right\rceil + b_0 \right)^c, \tag{10}$$

$$[\text{Exponential growth BS}] \ b_t = \delta^{m \left\lceil \frac{t}{\sum_{k=0}^{m} K_k E_k} \right\rceil} b_0, \tag{11}$$

where $a \in \mathbb{R}_{++}$, $c, \delta > 1$, and $E_m$ and $K_m$ are the numbers of, respectively, epochs and steps per epoch when the batch size is $(am + b_0)^c$ or $\delta^m b_0$. For example, the exponential growth batch size defined by (11) with $\delta = 2$ makes batch size double each $E_m$ epochs. We may modify the parameters $a$ and $\delta$ to $a_t$ and $\delta_t$ monotone increasing with $t$. The total number of steps for the batch size to increase $M$ times is $T = \sum_{m=0}^{M} K_m E_m$. An analysis of Algorithm 1 with a constant batch size $b_t = b$ and decaying learning rates satisfying (9) is given in Section 3.1.

Lemma 2.1 leads to the following them (the proof of the theorem and the result for Polynomial BS (10) are given in Appendix A.3).

**Theorem 3.2 (Convergence rate of SGD using (9))** *Under the assumptions in Lemma 2.1, Algorithm 1 using (9) satisfies that, for all $M \in \mathbb{N}$,*

$$\min_{t \in [0:T-1]} \mathbb{E}\left[\|\nabla f(\boldsymbol{\theta}_t)\|^2\right] \leq \frac{2(f(\boldsymbol{\theta}_0) - \underline{f}^\star)}{2 - \bar{L}\eta_{\max}} \underbrace{\frac{1}{\sum_{t=0}^{T-1} \eta_t}}_{B_T} + \frac{\bar{L}\sigma^2}{2 - \bar{L}\eta_{\max}} \underbrace{\frac{1}{\sum_{t=0}^{T-1} \eta_t} \sum_{t=0}^{T-1} \frac{\eta_t^2}{b_t}}_{V_T},$$

where $T = \sum_{m=0}^{M} K_m E_m$, $E_{\max} = \sup_{M \in \mathbb{N}} \sup_{m \in [0:M]} E_m < +\infty$, $K_{\max} = \sup_{M \in \mathbb{N}} \sup_{m \in [0:M]} K_m < +\infty$, $B_T$ is defined as in (7), and $V_T$ is bounded as

$$V_T \leq \begin{cases} \dfrac{\delta \eta_{\max} K_{\max} E_{\max}}{(\delta - 1) b_0 T} & \text{[Constant LR (3)]} \\[2mm] \dfrac{\delta \eta_{\max} K_{\max} E_{\max}}{2(\delta - 1) b_0 (\sqrt{T+1} - 1)} & \text{[Diminishing LR (4)]} \\[2mm] \dfrac{2\delta \eta_{\max}^2 K_{\max} E_{\max}}{(\delta - 1)(\eta_{\min} + \eta_{\max}) b_0 T} & \text{[Cosine LR (5)]} \\[2mm] \dfrac{(p+1)\delta \eta_{\max}^2 K_{\max} E_{\max}}{(\delta - 1)(\eta_{\max} + \eta_{\min} p) b_0 T} & \text{[Polynomial LR (6)].} \end{cases} \quad (\text{[Exponential BS (11)]})$$

That is, Algorithm 1 using Exponential BS (11) has the convergence rate

$$\min_{t \in [0:T-1]} \mathbb{E}\left[\|\nabla f(\boldsymbol{\theta}_t)\|\right] = \begin{cases} O\left(\dfrac{1}{\sqrt{T}}\right) & \text{[Constant LR (3), Cosine LR (5), Polynomial LR (6)]} \\[3mm] O\left(\dfrac{1}{T^{\frac{1}{4}}}\right) & \text{[Diminishing LR (4)].} \end{cases}$$

Theorem 3.2 (Theorem A.1) indicates that, with increasing batch sizes such as Polynomial BS (10) and Exponential BS (11), Algorithm 1 using each of Constant LR (3), Cosine LR (5), and Polynomial LR (6) has the convergence rate $O(\frac{1}{\sqrt{T}})$, in contrast to Theorem 3.1.

### 3.3 Increasing Batch Size and Increasing Learning Rate Scheduler

This section considers an increasing batch size and an increasing learning rate:

$$b_t \leq b_{t+1} \ (t \in \mathbb{N}) \quad \text{and} \quad \eta_t \leq \eta_{t+1} \ (t \in \mathbb{N}). \tag{12}$$

Example of $b_t$ and $\eta_t$ satisfying (12) is as follows: for all $m \in [0:M]$ and all $t \in S_m = \mathbb{N} \cap [\sum_{k=0}^{m-1} K_k E_k, \sum_{k=0}^{m} K_k E_k)$ $(S_0 = \mathbb{N} \cap [0, K_0 E_0))$,

$$[\text{Exponential growth BS and LR}] \ b_t = \delta^{m \left\lceil \frac{t}{\sum_{k=0}^{m} K_k E_k} \right\rceil} b_0, \ \eta_t = \gamma^{m \left\lceil \frac{t}{\sum_{k=0}^{m} K_k E_k} \right\rceil} \eta_0, \tag{13}$$

where $\delta, \gamma > 1$ such that $\gamma^2 < \delta$; and $E_m$ and $K_m$ are defined as in (11). We may modify the parameters $\gamma$ and $\delta$ to be monotone increasing parameters in $t$. The total number of steps when both batch size and learning rate increase $M$ times is $T = \sum_{m=0}^{M} K_m E_m$.

Lemma 2.1 leads to the following theorem (the proof of the theorem and the result for Polynomial growth BS and LR (27) are given in Appendix A.3).

**Theorem 3.3 (Convergence rate of SGD using (12))** *Under the assumptions in Lemma 2.1, Algorithm 1 using (12) satisfies that, for all $M \in \mathbb{N}$,*

$$\min_{t \in [0:T-1]} \mathbb{E}\left[\|\nabla f(\boldsymbol{\theta}_t)\|^2\right] \leq \frac{2(f(\boldsymbol{\theta}_0) - \underline{f}^\star)}{2 - \bar{L}\eta_{\max}} \underbrace{\frac{1}{\sum_{t=0}^{T-1} \eta_t}}_{B_T} + \frac{\bar{L}\sigma^2}{2 - \bar{L}\eta_{\max}} \underbrace{\frac{1}{\sum_{t=0}^{T-1} \eta_t} \sum_{t=0}^{T-1} \frac{\eta_t^2}{b_t}}_{V_T},$$

*where $T$, $E_{\max}$, and $K_{\max}$ are defined as in Theorem 3.2, $E_{\min} = \inf_{M \in \mathbb{N}} \inf_{m \in [0:M]} E_m < +\infty$, $K_{\min} = \inf_{M \in \mathbb{N}} \inf_{m \in [0:M]} K_m < +\infty$, $\hat{\gamma} = \frac{\gamma^2}{\delta} < 1$,*

$$B_T \leq \frac{\delta}{\eta_0 K_{\min} E_{\min} \gamma^M}, \ V_T \leq \frac{K_{\max} E_{\max} \eta_0 \delta}{K_{\min} E_{\min} b_0 (1 - \hat{\gamma}) \gamma^M}.$$

*That is, Algorithm 1 has the convergence rate*

$$\min_{t \in [0:T-1]} \mathbb{E}\left[\|\nabla f(\boldsymbol{\theta}_t)\|\right] = O\left(\frac{1}{\gamma^{\frac{M}{2}}}\right) \text{ [Exponential growth BS and LR (13)].}$$

Under Exponential BS (11), using Exponential LR (13) improves the convergence rate from $O(\frac{1}{\sqrt{M}})$ with Constant LR (3), Cosine LR (5), or Polynomial LR (6) (Theorem 3.2) to $O(\frac{1}{\gamma^{\frac{M}{2}}})$ $(\gamma > 1)$.

### 3.4 Increasing Batch Size and Warm-up Decaying Learning Rate Scheduler

This section considers an increasing batch size and a decaying learning rate with warm-up for a given $T_w = \sum_{m=0}^{M_w} K_m E_m > 0$ (learning rate increases $M_w$ times):

$$b_t \leq b_{t+1} \ (t \in \mathbb{N}) \quad \text{and} \quad \eta_t \leq \eta_{t+1} \ (t \in [T_w - 1]) \wedge \eta_{t+1} \leq \eta_t \ (t \geq T_w). \tag{14}$$

Examples of $b_t$ in (14) are Exponential BS (13) and Polynomial BS (27). Examples of $\eta_t$ in (14) can be obtained by combining (13) with (3)–(6). For example, for all $m \in [0:M]$ and all $t \in S_m$,

$$\text{[Constant LR with warm-up]} \ \eta_t = \begin{cases} \gamma^{m\left\lceil \frac{t}{\sum_{k=0}^{m} K_k E_k} \right\rceil} \eta_0 & (m \in [M_w]) \\ \gamma^{M_w} \eta_0 & (m \in [M_w : M]) \end{cases} \tag{15}$$

and [Cosine LR with warm-up]

$$\eta_t = \begin{cases} \gamma^{m\left\lceil \frac{t}{\sum_{k=0}^{m} K_k E_k} \right\rceil} \eta_0 & (m \in [M_w]) \\ \eta_{\min} + \dfrac{\eta_{\max} - \eta_{\min}}{2} \\ \quad \times \left\{ 1 + \cos\left( \sum_{k=0}^{m-1} E_k + \left\lfloor \dfrac{t - \sum_{k=0}^{m-1} K_k E_k}{K_m} \right\rfloor - E_w \right) \dfrac{\pi}{E_M - E_w} \right\} & (m \in [M_w : M]), \end{cases} \tag{16}$$

where $E_M$ is the total number of epochs when batch size increases $M$ times, $E_w$ is the number of warm-up epochs, $\eta_{\min} \geq 0$, $\eta_{\max} = \gamma^{M_w} \eta_0$, and $\gamma$ is defined as in (13).

Theorems 3.2 and 3.3 lead to the following theorem.

**Theorem 3.4 (Convergence rate of SGD using (14))** *Under the assumptions in Lemma 2.1, Algorithm 1 using (14) satisfies that, for all $M \in \mathbb{N}$,*

$$\min_{t \in [0:T-1]} \mathbb{E}\left[ \|\nabla f(\boldsymbol{\theta}_t)\|^2 \right] \leq \frac{2(f(\boldsymbol{\theta}_0) - f^\star)}{2 - \bar{L}\eta_{\max}} \underbrace{\frac{1}{\sum_{t=0}^{T-1} \eta_t}}_{B_T} + \frac{\bar{L}\sigma^2}{2 - \bar{L}\eta_{\max}} \underbrace{\frac{1}{\sum_{t=0}^{T-1} \eta_t} \sum_{t=0}^{T-1} \frac{\eta_t^2}{b_t}}_{V_T},$$

*where $b_t$ is the exponential growth batch size defined by (13) with $\delta, \gamma > 1$ such that $\gamma^2 < \delta$; $K_{\min}$, $K_{\max}$, $E_{\min}$, and $E_{\max}$ are defined as in Theorems 3.2 and 3.3;*

$$B_T \leq \begin{cases} \dfrac{\delta}{\eta_0 K_{\min} E_{\min} \gamma^{M_w}} + \dfrac{1}{\eta_{\max}(T - T_w)} & \text{[Constant LR (15)]} \\ \dfrac{\delta}{\eta_0 K_{\min} E_{\min} \gamma^{M_w}} + \dfrac{2}{(\eta_{\min} + \eta_{\max})(T - T_w)} & \text{[Cosine LR (16)]} \end{cases}$$

$$V_T \leq \begin{cases} \dfrac{K_{\max} E_{\max} \eta_0 \delta}{K_{\min} E_{\min} b_0 (1 - \hat{\gamma}) \gamma^{M_w}} + \dfrac{\delta \eta_{\max} K_{\max} E_{\max}}{(\delta - 1) b_0 (T - T_w)} & \text{[Constant LR (15)]} \\ \dfrac{K_{\max} E_{\max} \eta_0 \delta}{K_{\min} E_{\min} b_0 (1 - \hat{\gamma}) \gamma^{M_w}} + \dfrac{2\delta \eta_{\max}^2 K_{\max} E_{\max}}{(\delta - 1)(\eta_{\min} + \eta_{\max}) b_0 (T - T_w)} & \text{[Cosine LR (16)].} \end{cases}$$

*That is, Algorithm 1 has the convergence rate*

$$\min_{t \in [T_w : T-1]} \mathbb{E}\left[ \|\nabla f(\boldsymbol{\theta}_t)\| \right] = O\left( \frac{1}{\sqrt{T - T_w}} \right) \text{ [Constant LR (15), Cosine LR (16)]}.$$

Since Algorithm 1 with (15) and (16) uses increasing batch sizes and decaying learning rates for $t \geq T_w$, it has the same convergence rate as using (9) in Theorem 3.2. Meanwhile, since Algorithm 1 with (15) and (16) uses the warm-up learning rates for $t \in [T_w]$, Algorithm 1 speeds up during the warm-up period, based on Theorem 3.3. As a result, for increasing batch sizes, Algorithm 1 using decaying learning rates with warm-up minimizes $\mathbb{E}[\|\nabla f(\boldsymbol{\theta}_t)\|]$ faster than using decaying learning rates in Theorem 3.2.

### 3.5 Comparisons of Our Convergence Rate Results with Existing Ones

This section compares our results with the existing analyses of SGD for nonconvex optimization. The comparisons are summarized in Table 1. Let us consider the case where a learning rate $\eta_t$ is constant, i.e., $\eta_t = \eta > 0$. Theorem 11 in (Scaman & Malherbe, 2020) indicated that SGD with $\eta = O(\frac{1}{\sqrt{LT}})$ satisfies $\min_{t \in [0:T-1]} \mathbb{E}[\|\nabla f(\boldsymbol{\theta}_t)\|] = O(\frac{1}{T^{\frac{1}{4}}})$. Corollary 1 in (Khaled & Richtárik, 2023) showed that, under a weaker condition (the expected smoothness (Khaled & Richtárik, 2023, Assumption 2)) than (A2), SGD with $\eta = O(\frac{1}{\sqrt{LT}})$ satisfies $\min_{t \in [0:T-1]} \mathbb{E}[\|\nabla f(\boldsymbol{\theta}_t)\|] = O(\frac{1}{\sqrt{T}})$. Meanwhile, Theorem 3.2 indicates that SGD with $\eta = O(\frac{1}{L})$ and an increasing batch size $b_t$ satisfies $\min_{t \in [0:T-1]} \mathbb{E}[\|\nabla f(\boldsymbol{\theta}_t)\|] = O(\frac{1}{\sqrt{T}})$. For example, let us consider training a DNN on the CIFAR-100 dataset ($n = 50000$) over $E = 200$ epochs. When the batch size $b_0$ is $2^5$, the number of steps per epoch is $K = \lceil \frac{n}{b_0} \rceil = 1563$. Hence, we have $T = KE = 312600$. Since the Lipschitz constant $\bar{L}$ of $\nabla f$ would be large, SGD with too small a learning rate $\eta = O(\frac{1}{\sqrt{LT}})$ would not work in practice. Meanwhile, since the learning rate $\eta = O(\frac{1}{L})$ is constant with respect to $T$, SGD with $\eta = O(\frac{1}{L})$ will work well.

Table 1: Comparisons of convergence analyses of SGD for nonconvex optimization. "Noise" in the Gradient column means that SGD uses noisy observation, i.e., $\boldsymbol{g}(\boldsymbol{\theta}) = \nabla f(\boldsymbol{\theta}) + (\text{Noise})$, of the full gradient $\nabla f(\boldsymbol{\theta})$, where $\sigma^2$ is the upper bound on (Noise), while "Mini-batch" in the Gradient column means that SGD uses a mini-batch gradient $\nabla f_{B_t}(\boldsymbol{\theta})$. "Strong Growth" in the Additional Assumption column means that there exists $\kappa > 0$ such that, for all $t \in \mathbb{N}$ and all $i \in [n]$, $\|\nabla f_i(\boldsymbol{\theta}_t)\| \geq \kappa \|\nabla f(\boldsymbol{\theta}_t)\|^2$. "Bounded Gradient" in the Additional Assumption column means that there exists $G > 0$ such that, for all $t \in \mathbb{N}$, $\mathbb{E}[\|\nabla f_{B_t}(\boldsymbol{\theta}_t)\|] \leq G$. "Polyak-Łojasiewicz" in the Additional Assumption column means that there exists $\rho > 0$ such that, for all $t \in \mathbb{N}$, $\|\nabla f(\boldsymbol{\theta}_t)\|^2 \geq 2\rho(f(\boldsymbol{\theta}_t) - f^\star)$, where $f^\star$ is the optimal value of $f$ over $\mathbb{R}^d$. "Armijo" in the Learning Rate column means that $\eta_t$ satisfies the Armijo line search condition. "Step Decay" in the Learning Rate column means that $\eta_t$ is step decay. "Polyak" in the Learning Rate column means that $\eta_t$ is a stochastic Polyak learning rate. Here, we let $\mathbb{E}\|\nabla f_T\| := \min_{t \in [0:T-1]} \mathbb{E}[\|\nabla f(\boldsymbol{\theta}_t)\|]$ and $\mathbb{E}[f_T] := \mathbb{E}[f(\boldsymbol{\theta}_T)]$. $\nu \in (0,1)$ and $c$ is a positive constant.

| Reference and Theorem | Gradient | Additional Assumption | Learning Rate | Convergence Analysis |
|---|---|---|---|---|
| (Scaman & Malherbe, 2020) | Noise | —— | $\eta = O\left(\frac{1}{\sqrt{LT}}\right)$ | $\mathbb{E}\|\nabla f_T\| = O\left(\frac{1}{T^{\frac{1}{4}}}\right)$ |
| (Vaswani et al., 2019) | Noise | Strong Growth | Armijo | $\mathbb{E}\|\nabla f_T\| = O\left(\frac{1}{\sqrt{T}}\right)$ |
| (Wang et al., 2021) | Noise | Bounded Gradient | Step Decay | $\mathbb{E}\|\nabla f_T\| = O\left(\frac{\sqrt{\log T}}{T^{\frac{1}{4}}}\right)$ |
| (Loizou et al., 2021) | Noise | Polyak-Łojasiewicz | Polyak | $\mathbb{E}[f_T] = f^\star + O(\nu^T + \sigma^2)$ |
| (Khaled & Richtárik, 2023) | Mini-batch | —— | $\eta = O\left(\frac{1}{\sqrt{LT}}\right)$ | $\mathbb{E}\|\nabla f_T\| = O\left(\frac{1}{\sqrt{T}}\right)$ |
| Theorem 3.2 (**Case (ii)**) | Mini-batch | —— | $\eta = O\left(\frac{1}{L}\right)$ | $\mathbb{E}\|\nabla f_T\| = O\left(\frac{1}{\sqrt{T}}\right)$ |
| Theorem 3.3 (**Case (iii)**) | Mini-batch | —— | Increasing | $\mathbb{E}\|\nabla f_T\| = O\left(\frac{1}{\gamma^{\frac{M}{2}}}\right)$ |
| Theorem 3.4 (**Case (iv)**) | Mini-batch | —— | Warm-up | $\mathbb{E}\|\nabla f_T\| = O\left(\frac{1}{\sqrt{T}}\right)$ |

The previous results reported in (Vaswani et al., 2019; Wang et al., 2021; Loizou et al., 2021) showed convergence of SGD with specialized learning rates, such as the Armijo line search learning rate, step decay

learning rate, and stochastic Polyak learning rate. Our results indicate that SGD using practical learning rates, such as the cosine-annealing and polynomial decay learning rates, minimizes $\min_{t \in [0:T-1]} \mathbb{E}[\|\nabla f(\boldsymbol{\theta}_t)\|]$ in the sense of a rate of convergence $O(\frac{1}{\sqrt{T}})$ (Theorem 3.2). In addition, we would like to emphasize that, using an increasing batch size, SGD with an increasing learning rate accelerates SGD with a constant learning rate (Theorem 3.3). The acceleration of SGD is guaranteed during $\eta_t < \frac{2}{L}$ and the convergence of SGD is not guaranteed during $\eta_t > \frac{2}{L}$. Therefore, using a warm-up constant learning rate (Case (iv)) is appropriate to guarantee fast convergence of SGD.

### 3.6 Comparisons of Convergence Rates under Nonconvexity with Ones under Convexity

While Sections 3.1–3.5 consider the case where $f$ is not always convex, this section considers the case where $f$ is convex and compares convergence rates under nonconvexity with ones under convexity. Table 2 summarizes the convergence rates under nonconvexity and convexity. The left column in Table 2 is obtained from the results in Sections 3.1–3.4 (see also the table in Section 1). A well-known performance measure of Algorithm 1 when $f$ is convex is $\min_{t \in [0:T-1]} \mathbb{E}[f(\boldsymbol{\theta}_t) - f^\star]$ (Garrigos & Gower, 2024, Section 9.1), where $f^\star$ is the optimal value of minimizing a convex function $f$. The right column in Table 2 gives the results under convexity of $f$. An upper bound of $\min_{t \in [0:T-1]} \mathbb{E}[f(\boldsymbol{\theta}_t) - f^\star]$ for the sequence $(\boldsymbol{\theta}_t)$ generated by Algorithm 1 can be obtained by using Lemma 2.1, that is, the result when $f$ is not always convex (the proof of the results for convexity is given in Appendix A.4). While the performance measure $\min_{t \in [0:T-1]} \mathbb{E}[\|\nabla f(\boldsymbol{\theta}_t)\|]$ for nonconvexity of $f$ differs from the measure $\min_{t \in [0:T-1]} \mathbb{E}[f(\boldsymbol{\theta}_t) - f^\star]$ for convexity of $f$, Algorithm 1 under, for example, Case (ii) with convexity of $f$ satisfies that $\min_{t \in [0:T-1]} \mathbb{E}[f(\boldsymbol{\theta}_t) - f^\star] = O(\frac{1}{T})$.

Table 2: Comparisons of convergence rates under nonconvexity (**Left**) with ones under convexity (**Right**). $f^\star$ is the optimal value of the minimization problem for a convex function $f$, and **Case (i)**–**Case (iv)** are the learning rates and batch size schedulers considered in Section 3.1–3.4 (see the table in Section 1 for the definitions of $b$, $\gamma$, $M$, and $T$)

| Schedular | Upper bound of $\min_{t \in [0:T-1]} \mathbb{E}[\|\nabla f(\boldsymbol{\theta}_t)\|]$ ($f$ is nonconvex) | Upper bound of $\min_{t \in [0:T-1]} \mathbb{E}[f(\boldsymbol{\theta}_t) - f^\star]$ ($f$ is convex) |
|---|---|---|
| **Case (i)** | $O\left(\sqrt{\frac{1}{T} + \frac{1}{b}}\right)$ | $O\left(\frac{1}{T} + \frac{1}{b}\right)$ |
| **Case (ii)** | $O\left(\frac{1}{\sqrt{T}}\right), \quad O\left(\frac{1}{\sqrt{M}}\right)$ | $O\left(\frac{1}{T}\right), \quad O\left(\frac{1}{M}\right)$ |
| **Case (iii)** | $O\left(\frac{1}{\gamma^{\frac{M}{2}}}\right)$ | $O\left(\frac{1}{\gamma^M}\right)$ |
| **Case (iv)** | $O\left(\frac{1}{\gamma^{\frac{M}{2}}}\right) \to O\left(\frac{1}{\sqrt{T}}\right)$ | $O\left(\frac{1}{\gamma^M}\right) \to O\left(\frac{1}{T}\right)$ |

## 4 Numerical Results

We examined training ResNet-18 on the CIFAR100 dataset by using Algorithm 1 (see Appendices A.9 and A.10 for training Wide-ResNet-28-10 on CIFAR100 and ResNet-18 on Tiny ImageNet). The experimental environment was two NVIDIA GeForce RTX 4090 GPUs and Intel Core i9 13900KF CPU. The software environment was Python 3.10.12, PyTorch 2.1.0, and CUDA 12.2. The code is available at `https://github.com/iiduka-researches/incr_both_bs_lr`.

We set the total number of epochs $E = 300$, the initial learning rate $\eta_0 = 0.1$, and the minimum learning rate $\eta_{\min} = 0$ in (5) and (6). The solid line in the figure represents the mean value, and the shaded area in the figure represents the maximum and minimum over three runs.

Let us first consider the case (Figure 1(a)) of a constant batch size ($b = 2^7$) and decaying learning rates $\eta_t$ defined by (3)–(6) discussed in Section 3.1, where "linear" in Figure 1 denotes Polynomial LR (6) with $p = 1$. Figure 1(b)–(d) indicate that using Diminishing LR (4) did not work well, since it decayed rapidly and was very small (Figure 1(a)). Figure 1(b)–(d) also indicate that Cosine LR (5) and Polynomial LR (6) performed better than Constant LR (3), as promised in the theoretical results in Theorem 3.1 and (8).

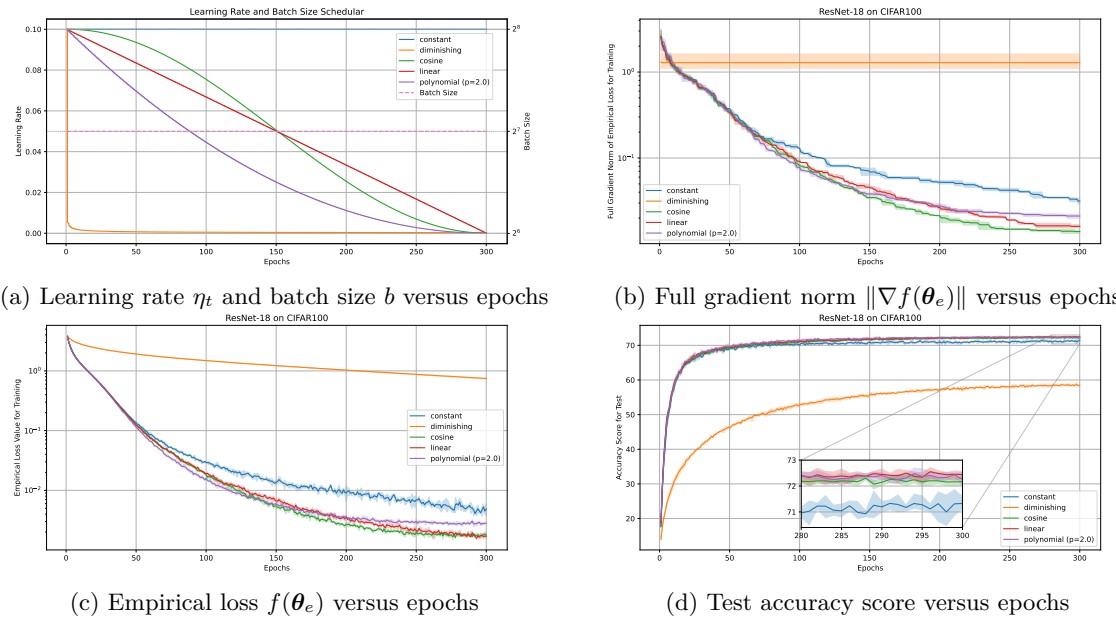

(a) Learning rate $\eta_t$ and batch size $b$ versus epochs

(b) Full gradient norm $\|\nabla f(\boldsymbol{\theta}_e)\|$ versus epochs

(c) Empirical loss $f(\boldsymbol{\theta}_e)$ versus epochs

(d) Test accuracy score versus epochs

Figure 1: (a) Decaying learning rates (constant, diminishing, cosine, linear, and polynomial) and constant batch size, (b) full gradient norm of empirical loss, (c) empirical loss value, and (d) accuracy score in testing for SGD to train ResNet-18 on CIFAR100 dataset.

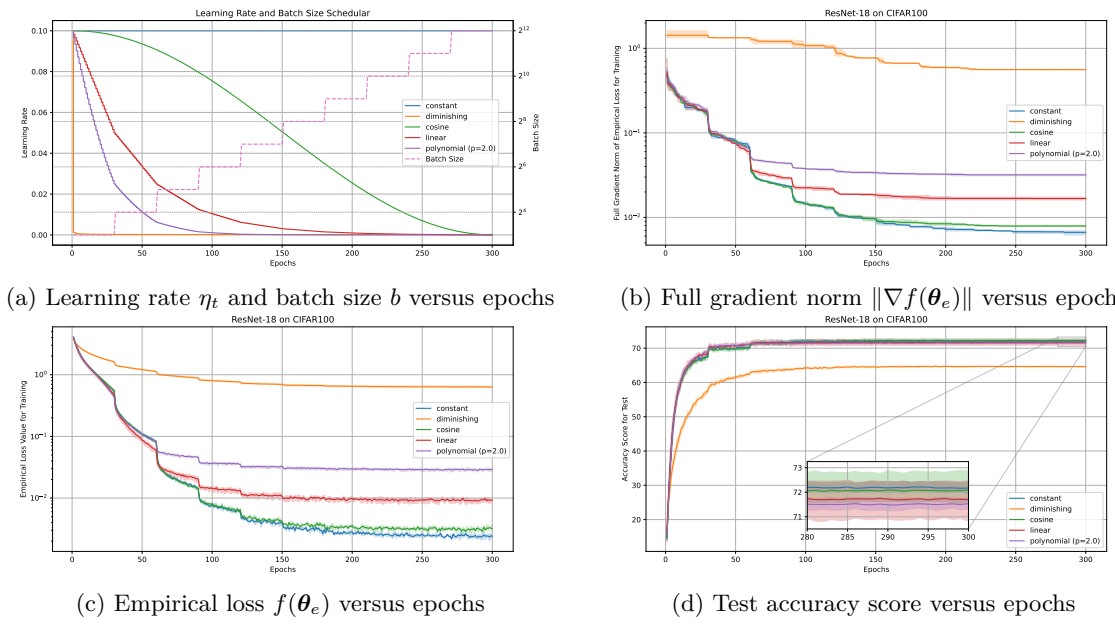

(a) Learning rate $\eta_t$ and batch size $b$ versus epochs

(b) Full gradient norm $\|\nabla f(\boldsymbol{\theta}_e)\|$ versus epochs

(c) Empirical loss $f(\boldsymbol{\theta}_e)$ versus epochs

(d) Test accuracy score versus epochs

Figure 2: (a) Decaying learning rates and doubly increasing batch size every 30 epochs, (b) full gradient norm of empirical loss, (c) empirical loss value, and (d) accuracy score in testing for SGD to train ResNet-18 on CIFAR100 dataset.

Next, let us consider the case (Figure 2(a)) of doubly increasing batch size every 30 epochs from an initial batch size $b_0 = 2^3$ and decaying learning rates $\eta_t$ defined by (3)–(6). Figure 2(a) indicates that the learning rate of Polynomial LR (6) updated each step ("linear" and "polynomial ($p = 2.0$)") becomes small at an early stage of training. This is because the smaller the batch size $b_t$ is, the larger the required number of steps $K_t = \lceil \frac{n}{b_t} \rceil$ per epoch becomes and the smaller the decaying learning rate $\eta_t$ becomes. Hence, in practice, increasing batch size is not compatible with Polynomial LR (6) updated each step. Meanwhile, Figure 2(a) indicates Constant LR (3) ("constant") and Cosine LR (5) ("cosine") were compatible with increasing batch size, since Constant LR (3) and Cosine LR (5) updated each epoch maintain large learning rates even for small batch sizes. In particular, Figure 2(b)–(d) indicate that using Constant LR (3) performed well.

Let us consider the case (Figure 3(a)) of doubly increasing batch size ($\delta = 2$) every 30 epochs and increasing learning rates defined by Exponential growth LR (13) with $\eta_0 = 0.1$ . The parameters $\gamma$ in the increasing learning rates considered here were (i) $\gamma \approx 1.080$ when $\eta_{\max} = 0.2$, (ii) $\gamma \approx 1.196$ when $\eta_{\max} = 0.5$, and (iii) $\gamma \approx 1.292$ when $\eta_{\max} = 1.0$, which satisfy the condition $\gamma^2 < \delta \ (= 2)$ to guarantee the convergence of Algorithm 1 (see Theorem 3.3). Figure 3 compares the result for "constant" in Figure 2 with the ones for the increasing learning rates (i)–(iii). Figure 3(b) indicates that the larger the learning rate $\eta_t$ was, the smaller the full gradient norm $\|\nabla f(\boldsymbol{\theta}_e)\|$ became and that Algorithm 1 with increasing learning rates minimized the full gradient norm faster than Algorithm 1 with a constant learning rate ("constant" in Figures 2 and 3).

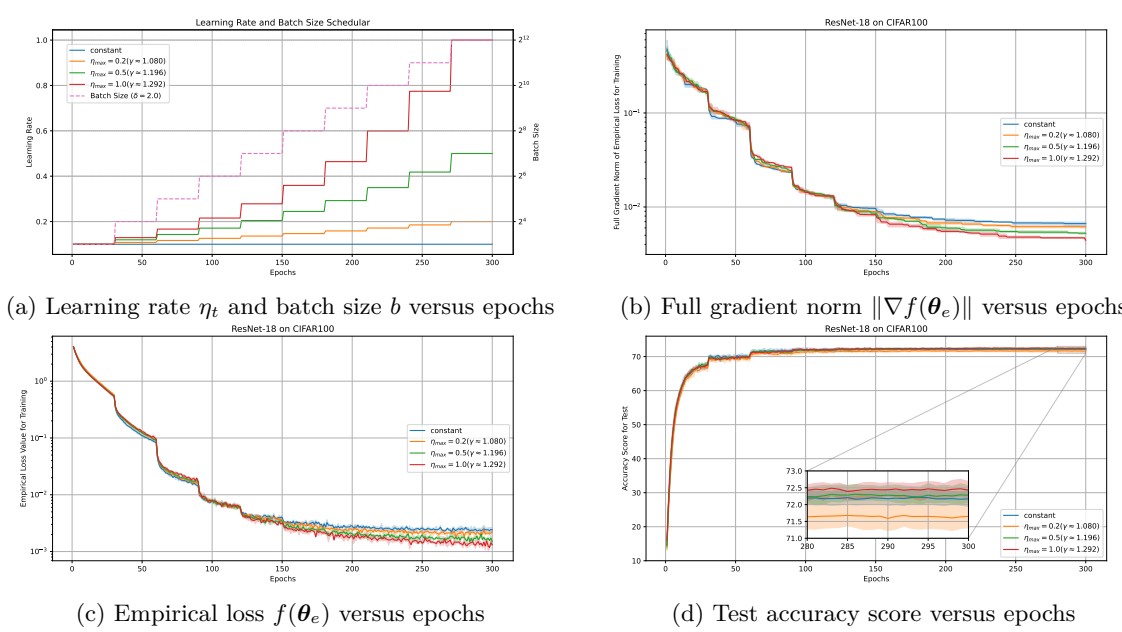

(a) Learning rate $\eta_t$ and batch size $b$ versus epochs

(b) Full gradient norm $\|\nabla f(\boldsymbol{\theta}_e)\|$ versus epochs

(c) Empirical loss $f(\boldsymbol{\theta}_e)$ versus epochs

(d) Test accuracy score versus epochs

Figure 3: (a) Increasing learning rates ($\eta_{\max} = 0.2, 0.5, 1.0$) and doubly increasing batch size every 30 epochs, (b) full gradient norm of empirical loss, (c) empirical loss value, and (d) accuracy score in testing for SGD to train ResNet-18 on CIFAR100 dataset.

Let us consider the case (Figure 4(a)) of a doubly increasing batch size and decaying learning rates (Constant LR (3) and Cosine LR (5)) with warm-up based on Figure 3(a). Figure 4(b) indicates that using decaying learning rates with warm-up accelerated Algorithm 1 more than using only increasing learning rates in Figure 3(b) and only a constant learning rate in Figure 2(b).

From the sufficient condition $\gamma^2 < \delta$ to guarantee convergence of Algorithm 1 with both batch size and learning rate increasing (Theorem 3.3), we can set a larger $\gamma$ when $\delta$ is large. Since Algorithm 1 has an $O(\frac{1}{\gamma^{\frac{M}{2}}})$ convergence rate (Theorem 3.3), using triply ($\gamma = 1.5 < \sqrt{\delta} = \sqrt{3}$) and quadruply ($\gamma = 1.9 < \sqrt{\delta} = \sqrt{4}$) increasing batch sizes theoretically decreases $\|\nabla f(\boldsymbol{\theta}_e)\|$ faster than doubly increasing batch sizes ($\gamma = 1.080 < \sqrt{\delta} = \sqrt{2}$ when $\eta_{\max} = 0.2$; Figure 3). Finally, we would like to verify whether the theoretical result holds in practice. The scheduler was as in Figure 5(a) with $\eta_0 = 0.1$ and $\eta_{\max} = 0.2$, where schedulers were modified such that batch sizes belong to $[2^3, 2^{12}]$ and learning rates belong to $[0.1, 0.2]$ (e.g.,

$b_e = a\delta^{\lfloor \frac{e}{30} \rfloor} + b$ and $\eta_e = c\gamma^{\lfloor \frac{e}{30} \rfloor} + d$, where $a \approx 0.2077$, $b \approx 7.7923$, $c \approx 0.00267$, and $d \approx 0.09733$ when $\delta = 3$ and $\gamma = 1.50$ and $a \approx 0.0155$, $b \approx 7.9844$, $c \approx 0.00031$, and $d \approx 0.09969$ when $\delta = 4$ and $\gamma = 1.90$). Figure 5(a) and (b) indicate that the larger the increasing rate of batch size was (the cases of $\delta = 3, 4$ after 180 epochs), the larger the increasing rate of the learning rate became ($\gamma = 1.5, 1.9$ when $\delta = 3, 4$) and the smaller $\|\nabla f(\boldsymbol{\theta}_e)\|$ became. That is, using increasing learning rates based on tripling and quadrupling batch sizes minimizes $\|\nabla f(\boldsymbol{\theta}_e)\|$ faster than using increasing learning rates based on doubly increasing batch sizes (see also Appendix A.6). Figure 5(c) and (d) indicate that using $\delta = 3, 4$ was better than using $\delta = 2$ in the sense of minimizing $f(\boldsymbol{\theta}_e)$ and achieving high test accuracy.

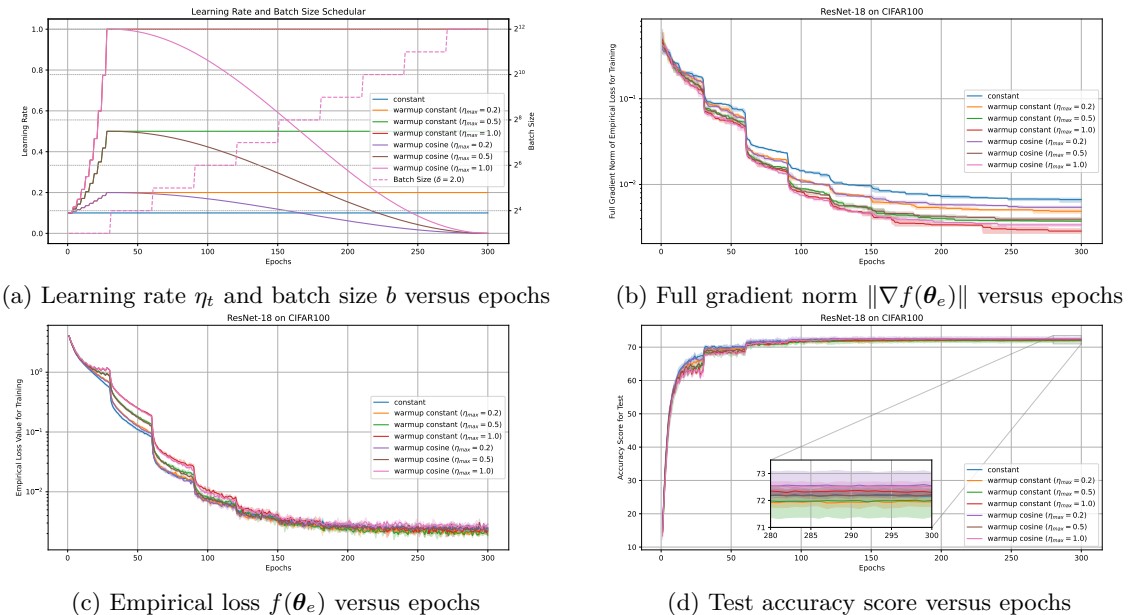

(a) Learning rate $\eta_t$ and batch size $b$ versus epochs   (b) Full gradient norm $\|\nabla f(\boldsymbol{\theta}_e)\|$ versus epochs

(c) Empirical loss $f(\boldsymbol{\theta}_e)$ versus epochs   (d) Test accuracy score versus epochs

Figure 4: (a) Warm-up learning rates and doubly increasing batch size every 30 epochs, (b) full gradient norm of empirical loss, (c) empirical loss value, and (d) accuracy score in testing for SGD to train ResNet-18 on CIFAR100 dataset.

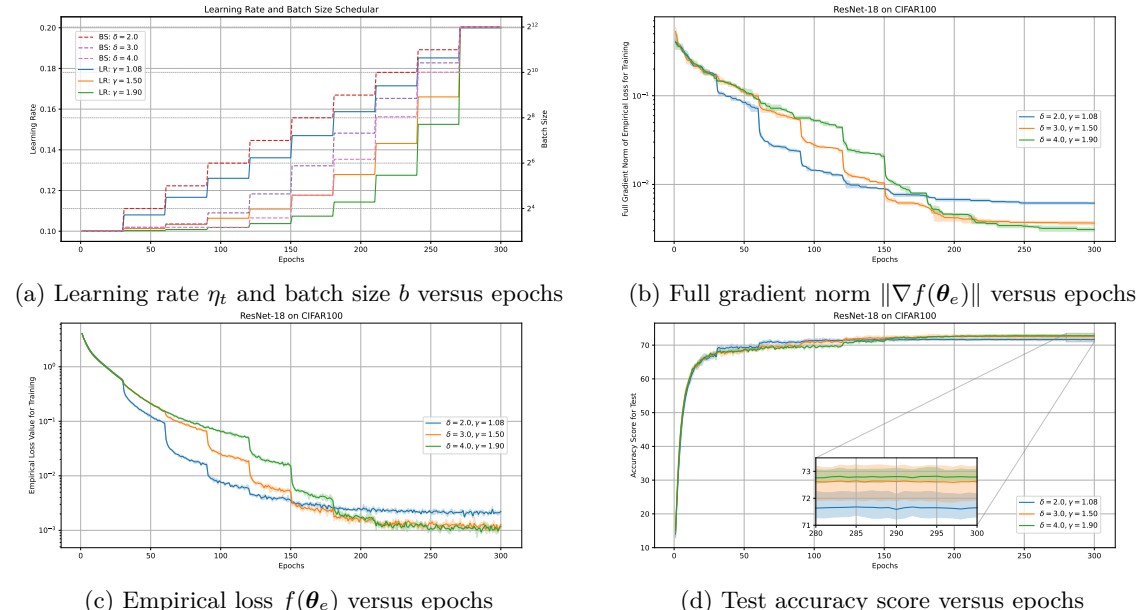

(a) Learning rate $\eta_t$ and batch size $b$ versus epochs   (b) Full gradient norm $\|\nabla f(\boldsymbol{\theta}_e)\|$ versus epochs

(c) Empirical loss $f(\boldsymbol{\theta}_e)$ versus epochs   (d) Test accuracy score versus epochs

Figure 5: (a) Increasing learning rates and increasing batch sizes based on $\delta = 2, 3, 4$, (b) full gradient norm of empirical loss, (c) empirical loss value, and (d) accuracy score in testing for SGD to train ResNet-18 on CIFAR100 dataset.

To better understand the impact of increasing both batch size and learning rate, we compare the full gradient norm across the four cases **Case (i)**–**Case (iv)** using ResNet-18 on CIFAR-100 in Figure 6. The results highlight that **Case (iv)**, which employs both a larger batch size and a warm-up learning rate, exhibits the most favorable convergence properties. Specifically, **Case (iv)** achieves a significantly reduced full gradient norm throughout training compared to the other cases, confirming its effectiveness. Building on these observations, we validate the scalability of **Case (iv)** on the larger ImageNet dataset. The results, presented in Appendix A.5, further highlight the advantages of this scaling strategy in improving training efficiency and accelerating convergence on large-scale tasks.

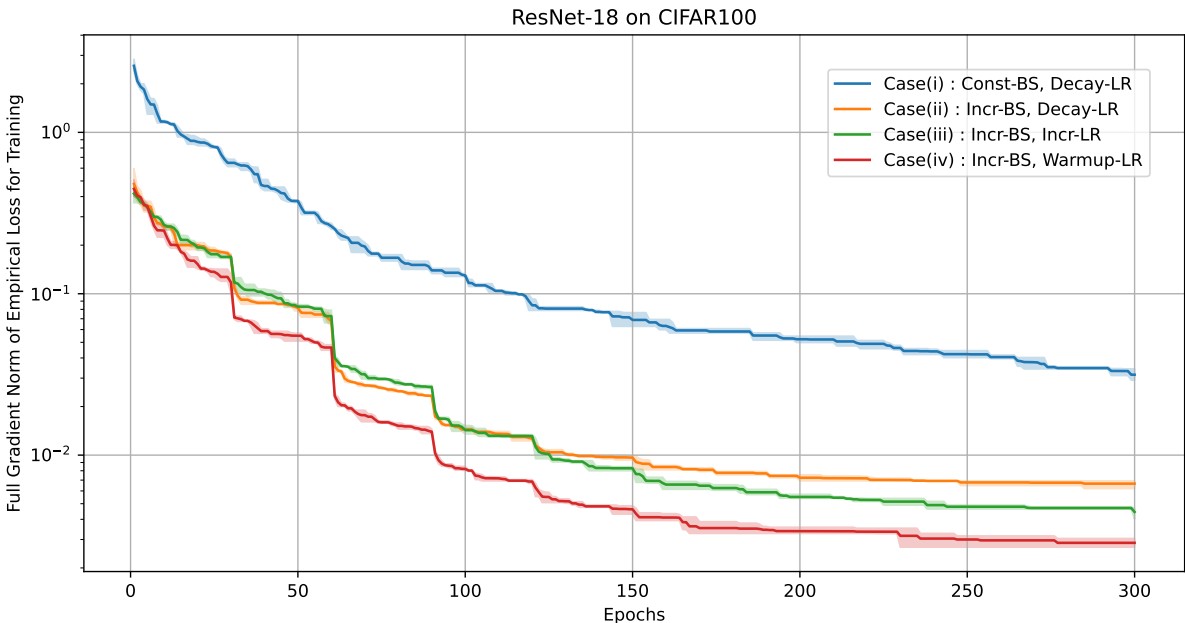

Figure 6: Performance comparison of best configurations for full gradient norm using Case (i) to Case (iv) to train ResNet-18 on CIFAR100 dataset.

## 5    Conclusion

This paper presented theoretical analyses of mini-batch SGD under batch size and learning rate schedulers used in practice. Our results indicated that using increasing batch sizes and decaying learning rates guarantees convergence of mini-batch SGD and using both batch sizes and learning rates that increase accelerates mini-batch SGD. That is, using increasing batch sizes and decaying learning rates with warm-up guarantees fast convergence of mini-batch SGD in the sense of minimizing the expectation of the full gradient norm of the empirical loss. This paper also provided numerical results to support the analysis results that increasing both batch sizes and learning rates accelerates mini-batch SGD. One limitation of this study is that the numbers of models and datasets in the experiments were limited. Hence, we should conduct similar experiments with larger numbers of models and datasets to support our theoretical results.

## Acknowledgments

We are sincerely grateful to the Action Editor, Ali Ramezani-Kebrya, and the three anonymous reviewers for helping us improve the original manuscript. This research is partly supported by the computational resources of the DGX A100 named TAIHO at Meiji University. This work was supported by the Japan Society for the Promotion of Science (JSPS) KAKENHI Grant Number 24K14846 awarded to Hideaki Iiduka. The authors thank FORTE Science Communications (`https://www.forte-science.co.jp/`) for English language editing.

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

# A Appendix

We here give the notation and state some definitions. Let $\mathbb{N}$ be the set of natural numbers. Define $[n] := \{1, 2, \cdots, n\}$ and $[0 : n] := \{0, 1, \cdots, n\}$ for $n \in \mathbb{N}$. Let $\mathbb{R}^d$ be the $d$-dimensional Euclidean space with inner product $\langle \boldsymbol{\theta}_1, \boldsymbol{\theta}_2 \rangle = \boldsymbol{\theta}_1^\top \boldsymbol{\theta}_2$ $(\boldsymbol{\theta}_1, \boldsymbol{\theta}_2 \in \mathbb{R}^d)$ and its induced norm $\|\boldsymbol{\theta}\| := \sqrt{\langle \boldsymbol{\theta}, \boldsymbol{\theta} \rangle}$ $(\boldsymbol{\theta} \in \mathbb{R}^d)$. Let $\mathbb{R}_+^d := \{\boldsymbol{\theta} = (\theta_1, \theta_2, \ldots, \theta_d)^\top \in \mathbb{R}^d \colon \theta_i \geq 0 \ (i \in [d])\}$ and $\mathbb{R}_{++}^d := \{\boldsymbol{\theta} = (\theta_1, \theta_2, \ldots, \theta_d)^\top \in \mathbb{R}^d \colon \theta_i > 0 \ (i \in [d])\}$. The gradient of a differentiable function $f \colon \mathbb{R}^d \to \mathbb{R}$ at $\boldsymbol{\theta} \in \mathbb{R}^d$ is denoted by $\nabla f(\boldsymbol{\theta})$. Let $L > 0$. A differentiable function $f \colon \mathbb{R}^d \to \mathbb{R}$ is said to be $L$-smooth if the gradient $\nabla f \colon \mathbb{R}^d \to \mathbb{R}^d$ is Lipschitz continuous, i.e., for all $\boldsymbol{\theta}_1, \boldsymbol{\theta}_2 \in \mathbb{R}^d$, $\|\nabla f(\boldsymbol{\theta}_1) - \nabla f(\boldsymbol{\theta}_2)\| \leq L\|\boldsymbol{\theta}_1 - \boldsymbol{\theta}_2\|$. Let $(x_t), (y_t) \subset \mathbb{R}_+$ be sequences. Let $O$ be Landau's symbol, i.e., $y_t = O(x_t)$ if there exist $c \in \mathbb{R}_+$ and $t_0 \in \mathbb{N}$ such that, for all $t \geq t_0$, $y_t \leq cx_t$.

## A.1 Example of Stochastic Gradient satisfying (A2) under (A1)

Suppose that (A1) holds and a random variable $\xi$ follows a uniform distribution. Then, let us show that (A2) holds, that is, for all $\boldsymbol{\theta} \in \mathbb{R}^d$, (i) $\mathbb{E}_\xi[\nabla f_\xi(\boldsymbol{\theta})] = \nabla f(\boldsymbol{\theta})$ and (ii) $\mathbb{V}_\xi[\nabla f_\xi(\boldsymbol{\theta})] \leq \sigma^2$.

(i) Since $\xi$ is independent of $\boldsymbol{\theta}$, we have that

$$\mathbb{E}_\xi[\nabla f_\xi(\boldsymbol{\theta})] = \frac{1}{n} \sum_{i \in [n]} \nabla f_i(\boldsymbol{\theta}) = \nabla \left( \frac{1}{n} \sum_{i \in [n]} f_i \right)(\boldsymbol{\theta}) = \nabla f(\boldsymbol{\theta}).$$

(ii) Let $\bar{\boldsymbol{\theta}} := \boldsymbol{\theta} - \frac{1}{L_i}\nabla f_i(\boldsymbol{\theta})$. The descent lemma and $f_i^\star = \inf\{f_i(\boldsymbol{\theta}) \colon \boldsymbol{\theta} \in \mathbb{R}^d\} \in \mathbb{R}$ ensure that

$$f_i^\star \leq f_i(\bar{\boldsymbol{\theta}}) \leq f_i(\boldsymbol{\theta}) + \langle \nabla f_i(\boldsymbol{\theta}), \bar{\boldsymbol{\theta}} - \boldsymbol{\theta} \rangle + \frac{L_i}{2}\|\bar{\boldsymbol{\theta}} - \boldsymbol{\theta}\|^2$$

$$= f_i(\boldsymbol{\theta}) - \frac{1}{L_i}\|\nabla f_i(\boldsymbol{\theta})\|^2 + \frac{1}{2L_i}\|\nabla f_i(\boldsymbol{\theta})\|^2$$

$$= f_i(\boldsymbol{\theta}) - \frac{1}{2L_i}\|\nabla f_i(\boldsymbol{\theta})\|^2,$$

which implies that, for all $i \in [n]$,

$$\|\nabla f_i(\boldsymbol{\theta})\|^2 \leq 2L_i(f_i(\boldsymbol{\theta}) - f_i^\star) \leq 2L_i M_i,$$

where $M_i := \sup\{f_i(\boldsymbol{\theta}) - f_i^\star \colon \boldsymbol{\theta} \in \mathbb{R}^d\}$. Hence, from (i),

$$\mathbb{V}_\xi[\nabla f_\xi(\boldsymbol{\theta})] = \mathbb{E}_\xi[\|\nabla f_\xi(\boldsymbol{\theta}) - \nabla f(\boldsymbol{\theta})\|^2]$$

$$= \mathbb{E}_\xi[\|\nabla f_\xi(\boldsymbol{\theta})\|^2] - 2\mathbb{E}_\xi[\langle \nabla f_\xi(\boldsymbol{\theta}), \nabla f(\boldsymbol{\theta}) \rangle] + \mathbb{E}_\xi[\|\nabla f(\boldsymbol{\theta})\|^2]$$

$$= \mathbb{E}_\xi[\|\nabla f_\xi(\boldsymbol{\theta})\|^2] - 2\|\nabla f(\boldsymbol{\theta})\|^2 + \|\nabla f(\boldsymbol{\theta})\|^2$$

$$= \frac{1}{n} \sum_{i \in [n]} \|\nabla f_i(\boldsymbol{\theta})\|^2 - \|\nabla f(\boldsymbol{\theta})\|^2$$

$$\leq \frac{2}{n} \sum_{i \in [n]} L_i M_i.$$

## A.2 Proofs of Proposition A.1 and Lemma 2.1

The following proposition holds for the mini-batch gradient.

**Proposition A.1** *Let $t \in \mathbb{N}$ and $\boldsymbol{\xi}_t$ be a random variable that is independent of $\boldsymbol{\xi}_j$ $(j \in [0 : t-1])$; let $\boldsymbol{\theta}_t \in \mathbb{R}^d$ be independent of $\boldsymbol{\xi}_t$; let $\nabla f_{B_t}(\boldsymbol{\theta}_t)$ be the mini-batch gradient defined by Algorithm 1, where $f_{\xi_{t,i}}$ $(i \in [b_t])$ is the stochastic gradient (see Assumption 2.1(A2)). Then, the following hold:*

$$\mathbb{E}_{\boldsymbol{\xi}_t}\left[\nabla f_{B_t}(\boldsymbol{\theta}_t)\Big|\hat{\boldsymbol{\xi}}_{t-1}\right] = \nabla f(\boldsymbol{\theta}_t) \ and \ \mathbb{V}_{\boldsymbol{\xi}_t}\left[\nabla f_{B_t}(\boldsymbol{\theta}_t)\Big|\hat{\boldsymbol{\xi}}_{t-1}\right] \leq \frac{\sigma^2}{b_t},$$

*where $\mathbb{E}_{\boldsymbol{\xi}_t}[\cdot|\hat{\boldsymbol{\xi}}_{t-1}]$ and $\mathbb{V}_{\boldsymbol{\xi}_t}[\cdot|\hat{\boldsymbol{\xi}}_{t-1}]$ are respectively the expectation and variance with respect to $\boldsymbol{\xi}_t$ conditioned on $\boldsymbol{\xi}_{t-1} = \hat{\boldsymbol{\xi}}_{t-1}$.*

The first equation in Proposition A.1 indicates that the mini-batch gradient $\nabla f_{B_t}(\boldsymbol{\theta}_t)$ is an unbiased estimator of the full gradient $\nabla f(\boldsymbol{\theta}_t)$. The second inequality in Proposition A.1 indicates that the upper bound on the variance of the mini-batch gradient $\nabla f_{B_t}(\boldsymbol{\theta}_t)$ is inversely proportional to the batch size $b_t$.

Assumption 2.1(A3) holds under sampling with replacement, where $\boldsymbol{\xi} = (\xi_1, \xi_2, \cdots, \xi_b)^\top$ are independent and identically distributed (i.i.d.). In practice, however, sampling without replacement is commonly used to improve data efficiency and diversity. Under sampling without replacement, minor dependencies arise between samples, particularly when $b \approx n$. These dependencies reduce the conditional variance $\mathbb{V}_{\boldsymbol{\xi}_t}[\nabla f_{B_t}(\boldsymbol{\theta}_t)|\hat{\boldsymbol{\xi}}_{t-1}]$, resulting in behavior that closely resembles full-batch gradient computation. On the other hand, when $b \ll n$, the dependencies introduced by sampling without replacement become negligible, and the behavior of the mini-batch gradient closely approximates that of i.i.d. samples. Under such conditions, the theoretical properties of Assumption 2.1(A3) are approximately satisfied. Therefore, in large-scale datasets, the approximation of i.i.d. is sufficient to ensure that the theoretical predictions remain valid under sampling without replacement. The experimental results (Appendix A.8) further confirm the validity of this approximation, as the observed behavior aligns closely with the theoretical predictions.

The proof of Proposition A.1 is based on standard results from the literature on SGD analysis (Garrigos & Gower, 2024, Section 6).

*Proof of Proposition A.1:* Assumption 2.1(A3) and the independence of $b_t$ and $\boldsymbol{\xi}_t$ ensure that

$$\mathbb{E}_{\boldsymbol{\xi}_t}\left[\nabla f_{B_t}(\boldsymbol{\theta}_t)\Big|\hat{\boldsymbol{\xi}}_{t-1}\right] = \mathbb{E}_{\boldsymbol{\xi}_t}\left[\frac{1}{b_t}\sum_{i=1}^{b_t}\nabla f_{\xi_{t,i}}(\boldsymbol{\theta}_t)\Big|\hat{\boldsymbol{\xi}}_{t-1}\right] = \frac{1}{b_t}\sum_{i=1}^{b_t}\mathbb{E}_{\xi_{t,i}}\left[\nabla f_{\xi_{t,i}}(\boldsymbol{\theta}_t)\Big|\hat{\boldsymbol{\xi}}_{t-1}\right],$$

which, together with Assumption 2.1(A2)(i) and the independence of $\boldsymbol{\xi}_t$ and $\boldsymbol{\xi}_{t-1}$, implies that

$$\mathbb{E}_{\boldsymbol{\xi}_t}\left[\nabla f_{B_t}(\boldsymbol{\theta}_t)\Big|\hat{\boldsymbol{\xi}}_{t-1}\right] = \frac{1}{b_t}\sum_{i=1}^{b_t}\nabla f(\boldsymbol{\theta}_t) = \nabla f(\boldsymbol{\theta}_t). \tag{17}$$

Assumption 2.1(A3), the independence of $b_t$ and $\boldsymbol{\xi}_t$, and (17) imply that

$$\mathbb{V}_{\boldsymbol{\xi}_t}\left[\nabla f_{B_t}(\boldsymbol{\theta}_t)\Big|\hat{\boldsymbol{\xi}}_{t-1}\right] = \mathbb{E}_{\boldsymbol{\xi}_t}\left[\|\nabla f_{B_t}(\boldsymbol{\theta}_t) - \nabla f(\boldsymbol{\theta}_t)\|^2\Big|\hat{\boldsymbol{\xi}}_{t-1}\right]$$

$$= \mathbb{E}_{\boldsymbol{\xi}_t}\left[\left\|\frac{1}{b_t}\sum_{i=1}^{b_t}\nabla f_{\xi_{t,i}}(\boldsymbol{\theta}_t) - \nabla f(\boldsymbol{\theta}_t)\right\|^2\Big|\hat{\boldsymbol{\xi}}_{t-1}\right]$$

$$= \frac{1}{b_t^2}\mathbb{E}_{\boldsymbol{\xi}_t}\left[\left\|\sum_{i=1}^{b_t}\left(\nabla f_{\xi_{t,i}}(\boldsymbol{\theta}_t) - \nabla f(\boldsymbol{\theta}_t)\right)\right\|^2\Big|\hat{\boldsymbol{\xi}}_{t-1}\right].$$

From the independence of $\xi_{t,i}$ and $\xi_{t,j}$ $(i \neq j)$ and Assumption 2.1(A2)(i), for all $i, j \in [b_t]$ such that $i \neq j$,

$$\mathbb{E}_{\xi_{t,i}}[\langle \nabla f_{\xi_{t,i}}(\boldsymbol{\theta}_t) - \nabla f(\boldsymbol{\theta}_t), \nabla f_{\xi_{t,j}}(\boldsymbol{\theta}_t) - \nabla f(\boldsymbol{\theta}_t)\rangle|\hat{\boldsymbol{\xi}}_{t-1}]$$

$$= \langle \mathbb{E}_{\xi_{t,i}}[\nabla f_{\xi_{t,i}}(\boldsymbol{\theta}_t)|\hat{\boldsymbol{\xi}}_{t-1}] - \mathbb{E}_{\xi_{t,i}}[\nabla f(\boldsymbol{\theta}_t)|\hat{\boldsymbol{\xi}}_{t-1}], \nabla f_{\xi_{t,j}}(\boldsymbol{\theta}_t) - \nabla f(\boldsymbol{\theta}_t)\rangle$$

$$= 0.$$

Hence, Assumption 2.1(A2)(ii) guarantees that

$$\mathbb{V}_{\boldsymbol{\xi}_t}\left[\nabla f_{B_t}(\boldsymbol{\theta})\Big|\hat{\boldsymbol{\xi}}_{t-1}\right] = \frac{1}{b_t^2}\sum_{i=1}^{b_t}\mathbb{E}_{\xi_{t,i}}\left[\|\nabla f_{\xi_{t,i}}(\boldsymbol{\theta}_t) - \nabla f(\boldsymbol{\theta}_t)\|^2\Big|\hat{\boldsymbol{\xi}}_{t-1}\right] \leq \frac{\sigma^2 b_t}{b_t^2} = \frac{\sigma^2}{b_t},$$

which completes the proof. $\qquad\square$

*Proof of Lemma 2.1:* The $\bar{L}$-smoothness of $f$ implies that the descent lemma holds; i.e., for all $t \in \mathbb{N}$,

$$f(\boldsymbol{\theta}_{t+1}) \leq f(\boldsymbol{\theta}_t) + \langle \nabla f(\boldsymbol{\theta}_t), \boldsymbol{\theta}_{t+1} - \boldsymbol{\theta}_t \rangle + \frac{\bar{L}}{2} \|\boldsymbol{\theta}_{t+1} - \boldsymbol{\theta}_t\|^2,$$

which, together with $\boldsymbol{\theta}_{t+1} := \boldsymbol{\theta}_t - \eta_t \nabla f_{B_t}(\boldsymbol{\theta}_t)$, implies that

$$f(\boldsymbol{\theta}_{t+1}) \leq f(\boldsymbol{\theta}_t) - \eta_t \langle \nabla f(\boldsymbol{\theta}_t), \nabla f_{B_t}(\boldsymbol{\theta}_t) \rangle + \frac{\bar{L}\eta_t^2}{2} \|\nabla f_{B_t}(\boldsymbol{\theta}_t)\|^2. \tag{18}$$

Proposition A.1 guarantees that

$$
\begin{aligned}
\mathbb{E}_{\boldsymbol{\xi}_t} \left[ \|\nabla f_{B_t}(\boldsymbol{\theta}_t)\|^2 \,|\hat{\boldsymbol{\xi}}_{t-1} \right] &= \mathbb{E}_{\boldsymbol{\xi}_t} \left[ \|\nabla f_{B_t}(\boldsymbol{\theta}_t) - \nabla f(\boldsymbol{\theta}_t) + \nabla f(\boldsymbol{\theta}_t)\|^2 \left| \hat{\boldsymbol{\xi}}_{t-1} \right] \right. \\
&= \mathbb{E}_{\boldsymbol{\xi}_t} \left[ \|\nabla f_{B_t}(\boldsymbol{\theta}_t) - \nabla f(\boldsymbol{\theta}_t)\|^2 \left| \hat{\boldsymbol{\xi}}_{t-1} \right] \right. \\
&\quad + 2\mathbb{E}_{\boldsymbol{\xi}_t} \left[ \langle \nabla f_{B_t}(\boldsymbol{\theta}_t) - \nabla f(\boldsymbol{\theta}_t), \nabla f(\boldsymbol{\theta}_t) \rangle \left| \hat{\boldsymbol{\xi}}_{t-1} \right] + \mathbb{E}_{\boldsymbol{\xi}_t} \left[ \|\nabla f(\boldsymbol{\theta}_t)\|^2 \left| \hat{\boldsymbol{\xi}}_{t-1} \right] \right. \\
&\leq \frac{\sigma^2}{b_t} + \|\nabla f(\boldsymbol{\theta}_t)\|^2.
\end{aligned}
\tag{19}
$$

Taking the expectation conditioned on $\boldsymbol{\xi}_{t-1} = \hat{\boldsymbol{\xi}}_{t-1}$ on both sides of (18), together with Proposition A.1 and (19), guarantees that, for all $t \in \mathbb{N}$,

$$
\begin{aligned}
\mathbb{E}_{\boldsymbol{\xi}_t} \left[ f(\boldsymbol{\theta}_{t+1}) \Big| \hat{\boldsymbol{\xi}}_{t-1} \right] &\leq f(\boldsymbol{\theta}_t) - \eta_t \mathbb{E}_{\boldsymbol{\xi}_t} \left[ \langle \nabla f(\boldsymbol{\theta}_t), \nabla f_{B_t}(\boldsymbol{\theta}_t) \rangle \Big| \hat{\boldsymbol{\xi}}_{t-1} \right] + \frac{\bar{L}\eta_t^2}{2} \mathbb{E}_{\boldsymbol{\xi}_t} \left[ \|\nabla f_{B_t}(\boldsymbol{\theta}_t)\|^2 \Big| \hat{\boldsymbol{\xi}}_{t-1} \right] \\
&\leq f(\boldsymbol{\theta}_t) - \eta_t \|\nabla f(\boldsymbol{\theta}_t)\|^2 + \frac{\bar{L}\eta_t^2}{2} \left( \frac{\sigma^2}{b_t} + \|\nabla f(\boldsymbol{\theta}_t)\|^2 \right).
\end{aligned}
$$

Hence, taking the total expectation on both sides of the above inequality ensures that, for all $t \in \mathbb{N}$,

$$\eta_t \left( 1 - \frac{\bar{L}\eta_t}{2} \right) \mathbb{E} \left[ \|\nabla f(\boldsymbol{\theta}_t)\|^2 \right] \leq \mathbb{E}\left[ f(\boldsymbol{\theta}_t) - f(\boldsymbol{\theta}_{t+1}) \right] + \frac{\bar{L}\sigma^2\eta_t^2}{2b_t}.$$

Let $T \in \mathbb{N}$. Summing the above inequality from $t = 0$ to $t = T - 1$ ensures that

$$\sum_{t=0}^{T-1} \eta_t \left( 1 - \frac{\bar{L}\eta_t}{2} \right) \mathbb{E} \left[ \|\nabla f(\boldsymbol{\theta}_t)\|^2 \right] \leq \mathbb{E}\left[ f(\boldsymbol{\theta}_0) - f(\boldsymbol{\theta}_T) \right] + \frac{\bar{L}\sigma^2}{2} \sum_{t=0}^{T-1} \frac{\eta_t^2}{b_t},$$

which, together with Assumption 2.1(A1) (the lower bound $\underline{f}^\star := \frac{1}{n} \sum_{i \in [n]} f_i^\star$ of $f$), implies that

$$\sum_{t=0}^{T-1} \eta_t \left( 1 - \frac{\bar{L}\eta_t}{2} \right) \mathbb{E} \left[ \|\nabla f(\boldsymbol{\theta}_t)\|^2 \right] \leq f(\boldsymbol{\theta}_0) - \underline{f}^\star + \frac{\bar{L}\sigma^2}{2} \sum_{t=0}^{T-1} \frac{\eta_t^2}{b_t}.$$

Since $\eta_t \in [\eta_{\min}, \eta_{\max}]$, we have that

$$\left( 1 - \frac{\bar{L}\eta_{\max}}{2} \right) \sum_{t=0}^{T-1} \eta_t \mathbb{E} \left[ \|\nabla f(\boldsymbol{\theta}_t)\|^2 \right] \leq f(\boldsymbol{\theta}_0) - \underline{f}^\star + \frac{\bar{L}\sigma^2}{2} \sum_{t=0}^{T-1} \frac{\eta_t^2}{b_t},$$

which, together with $\eta_t \in [\eta_{\min}, \eta_{\max}] \subset [0, \frac{2}{L})$, implies that

$$\sum_{t=0}^{T-1} \eta_t \mathbb{E} \left[ \|\nabla f(\boldsymbol{\theta}_t)\|^2 \right] \leq \frac{2(f(\boldsymbol{\theta}_0) - \underline{f}^\star)}{2 - \bar{L}\eta_{\max}} + \frac{\bar{L}\sigma^2}{2 - \bar{L}\eta_{\max}} \sum_{t=0}^{T-1} \frac{\eta_t^2}{b_t}. \tag{20}$$

Therefore, from $\sum_{t=0}^{T-1} \eta_t \neq 0$, we have

$$\min_{t \in [0:T-1]} \mathbb{E}[\|\nabla f(\boldsymbol{\theta}_t)\|^2] \leq \frac{2(f(\boldsymbol{\theta}_0) - \underline{f}^\star)}{2 - \bar{L}\eta_{\max}} \frac{1}{\sum_{t=0}^{T-1} \eta_t} + \frac{\bar{L}\sigma^2}{2 - \bar{L}\eta_{\max}} \frac{\sum_{t=0}^{T-1} \eta_t^2 b_t^{-1}}{\sum_{t=0}^{T-1} \eta_t}, \tag{21}$$

which implies that the assertion in Lemma 2.1 holds. $\qquad\square$

### A.3 Proofs of Theorems

We can also consider the case where batch sizes decay. For simplicity, let us set a constant learning rate $\eta_t = \eta > 0$ and a decaying batch size $b_t = \frac{b}{t+1}$, where $b > 0$. Then, we have that $V_T \leq \frac{\eta}{T} \sum_{t=0}^{T-1} \frac{1}{b_t} = \frac{\eta(T+1)}{2b} \to +\infty$ $(T \to +\infty)$, which implies that convergence of mini-batch SGD is not guaranteed. Accordingly, this paper focuses on the four cases in the main text.

*Proof of Theorem 3.1:* Let $\eta_{\max} = \eta$.

[Constant LR (3)] We have that

$$B_T = \frac{1}{\sum_{t=0}^{T-1} \eta} = \frac{1}{\eta T}, \; V_T = \frac{\sum_{t=0}^{T-1} \eta^2}{b \sum_{t=0}^{T-1} \eta} = \frac{\eta}{b}.$$

[Diminishing LR (4)] We have that

$$\sum_{t=0}^{T-1} \frac{1}{\sqrt{t+1}} \geq \int_0^T \frac{dt}{\sqrt{t+1}} = 2(\sqrt{T+1} - 1),$$

which implies that

$$B_T = \frac{1}{\sum_{t=0}^{T-1} \frac{\eta}{\sqrt{t+1}}} \leq \frac{1}{2\eta(\sqrt{T+1} - 1)}.$$

We also have that

$$\sum_{t=0}^{T-1} \frac{1}{t+1} \leq 1 + \int_0^{T-1} \frac{dt}{t+1} = 1 + \log T,$$

which implies that

$$V_T = \frac{\eta \sum_{t=0}^{T-1} \frac{1}{t+1}}{b \sum_{t=0}^{T-1} \frac{1}{\sqrt{t+1}}} \leq \frac{\eta(1 + \log T)}{2b(\sqrt{T+1} - 1)}.$$

[Cosine LR (5)] We have

$$\sum_{t=0}^{KE-1} \eta_t = \eta_{\min} KE + \frac{\eta_{\max} - \eta_{\min}}{2} KE + \frac{\eta_{\max} - \eta_{\min}}{2} \sum_{t=0}^{KE-1} \cos \left\lfloor \frac{t}{K} \right\rfloor \frac{\pi}{E}.$$

From $\sum_{t=0}^{KE} \cos \lfloor \frac{t}{K} \rfloor \frac{\pi}{E} = K - 1$, we have

$$\sum_{t=0}^{KE-1} \cos \left\lfloor \frac{t}{K} \right\rfloor \frac{\pi}{E} = K - 1 - \cos \pi = K. \tag{22}$$

We thus have

$$\sum_{t=0}^{KE-1} \eta_t = \eta_{\min} KE + \frac{\eta_{\max} - \eta_{\min}}{2} KE + \frac{\eta_{\max} - \eta_{\min}}{2} K$$

$$= \frac{1}{2} \{(\eta_{\min} + \eta_{\max})KE + (\eta_{\max} - \eta_{\min})K\}$$

$$\geq \frac{(\eta_{\min} + \eta_{\max})KE}{2}.$$

Moreover, we have that

$$\sum_{t=0}^{KE-1} \eta_t^2 = \eta_{\min}^2 KE + \eta_{\min}(\eta_{\max} - \eta_{\min}) \sum_{t=0}^{KE-1} \left(1 + \cos \left\lfloor \frac{t}{K} \right\rfloor \frac{\pi}{E}\right)$$

$$+ \frac{(\eta_{\max} - \eta_{\min})^2}{4} \sum_{t=0}^{KE-1} \left(1 + \cos \left\lfloor \frac{t}{K} \right\rfloor \frac{\pi}{E}\right)^2,$$

which implies that

$$\sum_{t=0}^{KE-1} \eta_t^2 = \eta_{\min}\eta_{\max}KE + \frac{(\eta_{\max} - \eta_{\min})^2}{4}KE + \eta_{\min}(\eta_{\max} - \eta_{\min}) \sum_{t=0}^{KE-1} \cos \left\lfloor \frac{t}{K} \right\rfloor \frac{\pi}{E}$$

$$+ \frac{(\eta_{\max} - \eta_{\min})^2}{2} \sum_{t=0}^{KE-1} \cos \left\lfloor \frac{t}{K} \right\rfloor \frac{\pi}{E} + \frac{(\eta_{\max} - \eta_{\min})^2}{4} \sum_{t=0}^{KE-1} \cos^2 \left\lfloor \frac{t}{K} \right\rfloor \frac{\pi}{E}.$$

From

$$\sum_{t=0}^{KE} \cos^2 \left\lfloor \frac{t}{K} \right\rfloor \frac{\pi}{E} = \frac{1}{2} \sum_{t=0}^{KE} \left(1 + \cos 2 \left\lfloor \frac{t}{K} \right\rfloor \frac{\pi}{E}\right)$$

$$= \frac{1}{2}(KE + 1) + \frac{1}{2}$$

$$= \frac{KE}{2} + 1,$$

we have

$$\sum_{t=0}^{KE-1} \cos^2 \left\lfloor \frac{t}{K} \right\rfloor \frac{\pi}{E} = \frac{KE}{2} + 1 - \cos^2 \pi = \frac{KE}{2}.$$

From (22), we have

$$\sum_{t=0}^{KE-1} \eta_t^2 = \frac{(\eta_{\min} + \eta_{\max})^2}{4}KE + \eta_{\min}(\eta_{\max} - \eta_{\min})K + \frac{(\eta_{\max} - \eta_{\min})^2}{2}K + \frac{(\eta_{\max} - \eta_{\min})^2}{4}\frac{KE}{2}$$

$$= \frac{3\eta_{\min}^2 + 2\eta_{\min}\eta_{\max} + 3\eta_{\max}^2}{8}KE + \frac{(\eta_{\max} - \eta_{\min})(\eta_{\max} + \eta_{\min})}{2}K.$$

Hence, we have

$$B_T = \frac{1}{\sum_{t=0}^{KE-1} \eta_t} \leq \frac{2}{(\eta_{\min} + \eta_{\max})KE}$$

and

$$V_T = \frac{\sum_{t=0}^{KE-1} \eta_t^2}{b \sum_{t=0}^{KE-1} \eta_t} \leq \frac{3\eta_{\min}^2 + 2\eta_{\min}\eta_{\max} + 3\eta_{\max}^2}{4(\eta_{\min} + \eta_{\max})b} + \frac{\eta_{\max} - \eta_{\min}}{bE}.$$

[Polynomial LR (6)] Since $f(x) = (1 - x)^p$ is monotone decreasing for $x \in [0, 1)$, we have that

$$\int_0^1 (1 - x)^p \mathrm{d}x < \frac{1}{T} \sum_{t=0}^{T-1} \left(1 - \frac{t}{T}\right)^p,$$

which implies that

$$T \int_0^1 (1 - x)^p \mathrm{d}x < \sum_{t=0}^{T-1} \left(1 - \frac{t}{T}\right)^p. \tag{23}$$

Since $\int_0^1 (1-x)^p \mathrm{d}x = \frac{1}{p+1}$, (23) implies that

$$\sum_{t=0}^{T-1} \left(1 - \frac{t}{T}\right)^p > \frac{T}{p+1}.$$

Accordingly,

$$\begin{aligned}
\sum_{t=0}^{T-1} \eta_t &= (\eta_{\max} - \eta_{\min}) \sum_{t=0}^{T-1} \left(1 - \frac{t}{T}\right)^p + \eta_{\min} T \\
&> (\eta_{\max} - \eta_{\min}) \frac{T}{p+1} + \eta_{\min} T \\
&= \left(\frac{\eta_{\max} - \eta_{\min}}{p+1} + \eta_{\min}\right) T \\
&= \frac{\eta_{\max} + \eta_{\min} p}{p+1} T.
\end{aligned}$$

Since $f(x) = (1-x)^p$ and $g(x) = (1-x)^{2p}$ are monotone decreasing for $x \in [0,1)$, we have that

$$\frac{1}{T} \sum_{t=0}^{T-1} \left(1 - \frac{t}{T}\right)^p < \frac{1}{T} + \int_0^1 (1-x)^p \mathrm{d}x, \quad \frac{1}{T} \sum_{t=0}^{T-1} \left(1 - \frac{t}{T}\right)^{2p} < \frac{1}{T} + \int_0^1 (1-x)^{2p} \mathrm{d}x,$$

which imply that

$$\sum_{t=0}^{T-1} \left(1 - \frac{t}{T}\right)^p < 1 + T \int_0^1 (1-x)^p \mathrm{d}x, \quad \sum_{t=0}^{T-1} \left(1 - \frac{t}{T}\right)^{2p} < 1 + T \int_0^1 (1-x)^{2p} \mathrm{d}x. \tag{24}$$

Since we have that $\int_0^1 (1-x)^p \mathrm{d}x = \frac{1}{p+1}$ and $\int_0^1 (1-x)^{2p} \mathrm{d}x = \frac{1}{2p+1}$, (24) ensures that

$$\sum_{t=0}^{T-1} \left(1 - \frac{t}{T}\right)^p < 1 + \frac{T}{p+1}, \quad \sum_{t=0}^{T-1} \left(1 - \frac{t}{T}\right)^{2p} < 1 + \frac{T}{2p+1}.$$

Hence,

$$\begin{aligned}
\sum_{t=0}^{T-1} \eta_t^2 &= (\eta_{\max} - \eta_{\min})^2 \sum_{t=0}^{T-1} \left(1 - \frac{t}{T}\right)^{2p} + 2(\eta_{\max} - \eta_{\min}) \sum_{t=0}^{T-1} \left(1 - \frac{t}{T}\right)^p \eta_{\min} + \eta_{\min}^2 T \\
&< (\eta_{\max} - \eta_{\min})^2 \left(1 + \frac{T}{2p+1}\right) + 2(\eta_{\max} - \eta_{\min}) \left(1 + \frac{T}{p+1}\right) \eta_{\min} + \eta_{\min}^2 T \\
&= \frac{\eta_{\max}^2 (p+1)(2p+T+1) + 2\eta_{\max}\eta_{\min} pT + \eta_{\min}^2 (2p^2(T-1) - 3p - 1)}{(p+1)(2p+1)}.
\end{aligned}$$

Therefore,

$$B_T = \frac{1}{\sum_{t=0}^{T-1} \eta_t} \leq \frac{p+1}{(\eta_{\max} + \eta_{\min} p) T}$$

and

$$\begin{aligned}
V_T &= \frac{\sum_{t=0}^{T-1} \eta_t^2}{b \sum_{t=0}^{T-1} \eta_t} \\
&= \frac{\eta_{\max}^2 (p+1)(2p+T+1) + 2\eta_{\max}\eta_{\min} pT + \eta_{\min}^2 (2p^2(T-1) - 3p - 1)}{(2p+1)(\eta_{\max} + \eta_{\min} p) bT}
\end{aligned}$$

$$= \frac{2p^2\eta_{\min}^2 + 2p\eta_{\min}\eta_{\max} + (p+1)\eta_{\max}^2}{(2p+1)(p\eta_{\min}+\eta_{\max})b} + \frac{(p+1)(2p+1)\eta_{\max}^2 - (p+1)(2p+1)\eta_{\min}^2}{(2p+1)(p\eta_{\min}+\eta_{\max})bT}$$

$$= \frac{2p^2\eta_{\min}^2 + 2p\eta_{\min}\eta_{\max} + (p+1)\eta_{\max}^2}{(2p+1)(p\eta_{\min}+\eta_{\max})b} + \frac{(p+1)(\eta_{\max}^2 - \eta_{\min}^2)}{(p\eta_{\min}+\eta_{\max})bT}.$$

This completes the proof. □

We will now show the following theorem, which includes Theorem 3.2.

**Theorem A.1 (Convergence rate of SGD using (9))** *Under the assumptions in Lemma 2.1, Algorithm 1 using (9) satisfies that, for all $M \in \mathbb{N}$,*

$$\min_{t\in[0:T-1]} \mathbb{E}\left[\|\nabla f(\boldsymbol{\theta}_t)\|^2\right] \le \frac{2(f(\boldsymbol{\theta}_0) - \underline{f^\star})}{2 - \bar{L}\eta_{\max}} \underbrace{\frac{1}{\sum_{t=0}^{T-1}\eta_t}}_{B_T} + \frac{\bar{L}\sigma^2}{2 - \bar{L}\eta_{\max}} \underbrace{\frac{1}{\sum_{t=0}^{T-1}\eta_t}\sum_{t=0}^{T-1}\frac{\eta_t^2}{b_t}}_{V_T},$$

*where $T = \sum_{m=0}^{M} K_m E_m$, $E_{\max} = \sup_{M\in\mathbb{N}}\sup_{m\in[0:M]} E_m < +\infty$, $K_{\max} = \sup_{M\in\mathbb{N}}\sup_{m\in[0:M]} K_m < +\infty$, $\underline{a} = \min\{a, b_0\}$, $B_T$ is defined as in (7), and $V_T$ is given by*

$$V_T \le \begin{cases} \dfrac{3\eta_{\max}K_{\max}E_{\max}}{\underline{a}^c T} & \text{[Constant LR (3)]} \\[2mm] \dfrac{3\eta_{\max}K_{\max}E_{\max}}{2\underline{a}^c(\sqrt{T+1}-1)} & \text{[Diminishing LR (4)]} \\[2mm] \dfrac{6\eta_{\max}^2 K_{\max}E_{\max}}{\underline{a}^c(\eta_{\min}+\eta_{\max})T} & \text{[Cosine LR (5)]} \\[2mm] \dfrac{3(p+1)\eta_{\max}^2 K_{\max}E_{\max}}{\underline{a}^c(\eta_{\max}+\eta_{\min}p)T} & \text{[Polynomial LR (6)]} \end{cases} \qquad \text{([Polynomial BS (10)])}$$

$$V_T \le \begin{cases} \dfrac{\delta\eta_{\max}K_{\max}E_{\max}}{(\delta-1)b_0 T} & \text{[Constant LR (3)]} \\[2mm] \dfrac{\delta\eta_{\max}K_{\max}E_{\max}}{2(\delta-1)b_0(\sqrt{T+1}-1)} & \text{[Diminishing LR (4)]} \\[2mm] \dfrac{2\delta\eta_{\max}^2 K_{\max}E_{\max}}{(\delta-1)(\eta_{\min}+\eta_{\max})b_0 T} & \text{[Cosine LR (5)]} \\[2mm] \dfrac{(p+1)\delta\eta_{\max}^2 K_{\max}E_{\max}}{(\delta-1)(\eta_{\max}+\eta_{\min}p)b_0 T} & \text{[Polynomial LR (6)].} \end{cases} \qquad \text{([Exponential BS (11)])}$$

*That is, Algorithm 1 using each of Polynomial BS (10) and Exponential BS (11) has the convergence rate*

$$\min_{t\in[0:T-1]} \mathbb{E}\left[\|\nabla f(\boldsymbol{\theta}_t)\|\right] = \begin{cases} O\left(\dfrac{1}{\sqrt{T}}\right) & \text{[Constant LR (3), Cosine LR (5), Polynomial LR (6)]} \\[3mm] O\left(\dfrac{1}{T^{\frac{1}{4}}}\right) & \text{[Diminishing LR (4)].} \end{cases}$$

*Proof of Theorem A.1:* Let $M \in \mathbb{N}$ and $T = \sum_{m=0}^{M} K_m E_m$, where $E_{\max} = \sup_{M\in\mathbb{N}}\sup_{m\in[0:M]} E_m < +\infty$, $K_{\max} = \sup_{M\in\mathbb{N}}\sup_{m\in[0:M]} K_m < +\infty$, $S_0 := \mathbb{N} \cap [0, K_0 E_0)$, and $S_m = \mathbb{N} \cap [\sum_{k=0}^{m-1} K_k E_k, \sum_{k=0}^{m} K_k E_k)$ $(m \in [M])$. Let us consider using (10). Let $\eta_{\max} = \eta$ and $\underline{a} = \min\{a, b_0\}$.

[Constant LR (3)] Let $m \in [M]$. We have that

$$\sum_{t\in S_m} \frac{1}{b_t} = \sum_{t\in S_m} \frac{1}{\left(am\left\lceil \frac{t}{\sum_{k=0}^{m} K_k E_k}\right\rceil + b_0\right)^c} \le \sum_{t\in S_m} \frac{1}{a^c m^c \left\lceil \frac{t}{\sum_{k=0}^{m} K_k E_k}\right\rceil^c}$$

$$\le \sum_{t\in S_m} \frac{1}{a^c m^c} \le \frac{1}{a^c m^c} K_m E_m \le \frac{K_{\max}E_{\max}}{a^c}\frac{1}{m^c} \le \frac{K_{\max}E_{\max}}{\underline{a}^c}\frac{1}{m^c}$$

and

$$\sum_{t \in S_0} \frac{1}{b_t} = \sum_{t \in S_0} \frac{1}{b_0^c} \leq \frac{K_{\max} E_{\max}}{\underline{a}^c}.$$

Accordingly, we have that

$$\sum_{m=0}^{M} \sum_{t \in S_m} \frac{1}{b_t} \leq \frac{K_{\max} E_{\max}}{\underline{a}^c} \left( 1 + \sum_{m=1}^{M} \frac{1}{m^c} \right) \leq \frac{K_{\max} E_{\max}}{\underline{a}^c} \left( 1 + \sum_{m=1}^{+\infty} \frac{1}{m^c} \right) \qquad (25)$$
$$\leq \frac{3 K_{\max} E_{\max}}{\underline{a}^c}.$$

Hence, we have that

$$V_T = \frac{1}{\sum_{t=0}^{T-1} \eta} \sum_{t=0}^{T-1} \frac{\eta^2}{b_t} \leq \frac{3 \eta K_{\max} E_{\max}}{\underline{a}^c T}.$$

[Diminishing LR (4)] From (25), we have that

$$V_T = \frac{1}{\sum_{t=0}^{T-1} \frac{\eta}{\sqrt{t+1}}} \sum_{t=0}^{T-1} \frac{\eta^2}{(t+1) b_t}$$
$$\leq \frac{\eta}{2(\sqrt{T+1}-1)} \sum_{t=0}^{T-1} \frac{1}{b_t} \leq \frac{3 \eta K_{\max} E_{\max}}{2 \underline{a}^c (\sqrt{T+1}-1)}.$$

[Cosine LR (5)] The cosine LR is defined for all $m \in [0:M]$ and all $t \in S_m$ by

$$\eta_t = \eta_{\min} + \frac{\eta_{\max} - \eta_{\min}}{2} \left\{ 1 + \cos \left( \sum_{k=0}^{m-1} E_k + \left\lfloor \frac{t - \sum_{k=0}^{m-1} K_k E_k}{K_m} \right\rfloor \right) \frac{\pi}{E_M} \right\}.$$

We have that

$$\sum_{t=0}^{T-1} \frac{\eta_t^2}{b_t} \leq \eta_{\max}^2 \sum_{t=0}^{T-1} \frac{1}{b_t},$$

which, together with (25), implies that

$$\sum_{t=0}^{T-1} \frac{\eta_t^2}{b_t} \leq \frac{3 \eta_{\max}^2 K_{\max} E_{\max}}{\underline{a}^c}.$$

Hence, we have that

$$V_T = \frac{1}{\sum_{t=0}^{T-1} \eta_t} \sum_{t=0}^{T-1} \frac{\eta_t^2}{b_t} \leq \frac{6 \eta_{\max}^2 K_{\max} E_{\max}}{\underline{a}^c (\eta_{\min} + \eta_{\max}) T}.$$

[Polynomial LR (6)] We have that

$$\sum_{t=0}^{T-1} \frac{\eta_t^2}{b_t} = \sum_{t=0}^{T-1} \frac{1}{b_t} \left\{ (\eta_{\max} - \eta_{\min}) \left( 1 - \frac{t}{T} \right)^p + \eta_{\min} \right\}^2 \leq \eta_{\max}^2 \sum_{t=0}^{T-1} \frac{1}{b_t},$$

which, together with (25), implies that

$$\sum_{t=0}^{T-1} \frac{\eta_t^2}{b_t} \leq \frac{3 \eta_{\max}^2 K_{\max} E_{\max}}{\underline{a}^c}.$$

Hence, we have that

$$V_T = \frac{1}{\sum_{t=0}^{T-1} \eta_t} \sum_{t=0}^{T-1} \frac{\eta_t^2}{b_t} \le \frac{3(p+1)\eta_{\max}^2 K_{\max} E_{\max}}{\underline{a}^c (\eta_{\max} + \eta_{\min} p) T}.$$

Let us consider using (11). Let $\eta_{\max} = \eta$.

[Constant LR (3)] We have that

$$\sum_{t \in S_m} \frac{1}{b_t} = \sum_{t \in S_m} \frac{1}{m \left\lceil \frac{t}{\sum_{k=0}^{m} K_k E_k} \right\rceil b_0} \le \sum_{t \in S_m} \frac{1}{\delta^m b_0} \le \frac{K_{\max} E_{\max}}{\delta^m b_0},$$

which implies that

$$\sum_{m=0}^{M} \sum_{t \in S_m} \frac{1}{b_t} \le \frac{K_{\max} E_{\max}}{b_0} \sum_{m=0}^{M} \frac{1}{\delta^m} \le \frac{K_{\max} E_{\max} \delta}{b_0 (\delta - 1)}. \tag{26}$$

Hence, we have that

$$V_T = \frac{1}{\sum_{t=0}^{T-1} \eta} \sum_{t=0}^{T-1} \frac{\eta^2}{b_t} \le \frac{\eta K_{\max} E_{\max} \delta}{b_0 (\delta - 1) T}.$$

[Diminishing LR (4)] From (26), we have that

$$V_T = \frac{1}{\sum_{t=0}^{T-1} \frac{\eta}{\sqrt{t+1}}} \sum_{t=0}^{T-1} \frac{\eta^2}{(t+1) b_t} \le \frac{\eta}{2(\sqrt{T+1}-1)} \sum_{t=0}^{T-1} \frac{1}{b_t} \le \frac{\eta K_{\max} E_{\max} \delta}{2(\sqrt{T+1}-1) b_0 (\delta - 1)}.$$

[Cosine LR (5)] We have that

$$\sum_{t=0}^{T-1} \frac{\eta_t^2}{b_t} \le \eta_{\max}^2 \sum_{t=0}^{T-1} \frac{1}{b_t},$$

which, together with (26), implies that

$$\sum_{t=0}^{T-1} \frac{\eta_t^2}{b_t} \le \frac{\eta_{\max}^2 K_{\max} E_{\max} \delta}{b_0 (\delta - 1)}.$$

Hence, we have that

$$V_T = \frac{1}{\sum_{t=0}^{T-1} \eta_t} \sum_{t=0}^{T-1} \frac{\eta_t^2}{b_t} \le \frac{2 \eta_{\max}^2 K_{\max} E_{\max} \delta}{(\delta - 1)(\eta_{\min} + \eta_{\max}) b_0 T}.$$

[Polynomial LR (6)] We have that

$$\sum_{t=0}^{T-1} \frac{\eta_t^2}{b_t} = \sum_{t=0}^{T-1} \frac{1}{b_t} \left\{ (\eta_{\max} - \eta_{\min}) \left(1 - \frac{t}{T}\right)^p + \eta_{\min} \right\}^2 \le \eta_{\max}^2 \sum_{t=0}^{T-1} \frac{1}{b_t},$$

which, together with (26), implies that

$$\sum_{t=0}^{T-1} \frac{\eta_t^2}{b_t} \le \frac{\eta_{\max}^2 K_{\max} E_{\max} \delta}{b_0 (\delta - 1)}.$$

Hence, we have that

$$V_T = \frac{1}{\sum_{t=0}^{T-1} \eta_t} \sum_{t=0}^{T-1} \frac{\eta_t^2}{b_t} \leq \frac{(p+1)\eta_{\max}^2 K_{\max} E_{\max} \delta}{(\delta-1)(\eta_{\max} + \eta_{\min} p) b_0 T}.$$

$\square$

Example of $b_t$ and $\eta_t$ satisfying (12) is as follows:

[Polynomial growth BS and LR]

$$b_t = \left( a_1 m \left\lceil \frac{t}{\sum_{k=0}^m K_k E_k} \right\rceil + b_0 \right)^{c_1}, \; \eta_t = \left( a_2 m \left\lceil \frac{t}{\sum_{k=0}^m K_k E_k} \right\rceil + \eta_0 \right)^{c_2}, \tag{27}$$

where $a_1, a_2 > 0$; $c_1 > 1$, $c_2 > 0$ such that $c_1 - 2c_2 > 1$.

We next show the following theorem, which includes Theorem 3.3.

**Theorem A.2 (Convergence rate of SGD using (12))** *Under the assumptions in Lemma 2.1, Algorithm 1 using (12) satisfies that, for all $M \in \mathbb{N}$,*

$$\min_{t \in [0:T-1]} \mathbb{E}\left[ \|\nabla f(\boldsymbol{\theta}_t)\|^2 \right] \leq \frac{2(f(\boldsymbol{\theta}_0) - \underline{f}^\star)}{2 - \bar{L}\eta_{\max}} \underbrace{\frac{1}{\sum_{t=0}^{T-1} \eta_t}}_{B_T} + \frac{\bar{L}\sigma^2}{2 - \bar{L}\eta_{\max}} \underbrace{\frac{1}{\sum_{t=0}^{T-1} \eta_t} \sum_{t=0}^{T-1} \frac{\eta_t^2}{b_t}}_{V_T},$$

*where $T = \sum_{m=0}^M K_m E_m$, $E_{\max} = \sup_{M \in \mathbb{N}} \sup_{m \in [0:M]} E_m < +\infty$, $E_{\min} = \inf_{M \in \mathbb{N}} \inf_{m \in [0:M]} E_m < +\infty$, $K_{\max} = \sup_{M \in \mathbb{N}} \sup_{m \in [0:M]} K_m < +\infty$, $K_{\min} = \inf_{M \in \mathbb{N}} \inf_{m \in [0:M]} K_m < +\infty$, $\underline{\eta} = \min\{a_2, \eta_0\}$, $\bar{\eta} = \max\{a_2, \eta_0\}$, $\underline{b} = \min\{a_1, b_0\}$, $\hat{\gamma} = \frac{\gamma^2}{\delta} < 1$,*

$$B_T \leq \begin{cases} \dfrac{1 + c_2}{\underline{\eta}^{c_2} K_{\min} E_{\min} M^{1+c_2}} & \text{[Polynomial growth BS and LR (27)]} \\[4mm] \dfrac{\delta}{\eta_0 K_{\min} E_{\min} \gamma^M} & \text{[Exponential growth BS and LR (13)]} \end{cases}$$

$$V_T \leq \begin{cases} \dfrac{2 K_{\max} E_{\max}(1 + c_2)\bar{\eta}^{2c_2}}{K_{\min} E_{\min} \underline{\eta}^{c_2} \underline{b}^{c_1} M^{1+c_2}} & \text{[Polynomial growth BS and LR (27)]} \\[4mm] \dfrac{K_{\max} \bar{E}_{\max} \eta_0 \delta}{K_{\min} E_{\min} b_0 (1 - \hat{\gamma}) \gamma^M} & \text{[Exponential growth BS and LR (13)]}. \end{cases}$$

*That is, Algorithm 1 has the convergence rate*

$$\min_{t \in [0:T-1]} \mathbb{E}\left[ \|\nabla f(\boldsymbol{\theta}_t)\| \right] = \begin{cases} O\left( \dfrac{1}{M^{\frac{1+c_2}{2}}} \right) & \text{[Polynomial growth BS and LR (27)]} \\[4mm] O\left( \dfrac{1}{\gamma^{\frac{M}{2}}} \right) & \text{[Exponential growth BS and LR (13)].} \end{cases}$$

*Proof of Theorem A.2:* Let $M \in \mathbb{N}$ and $T = \sum_{m=0}^M K_m E_m$, where $E_{\max} = \sup_{M \in \mathbb{N}} \sup_{m \in [0:M]} E_m < +\infty$, $K_{\max} = \sup_{M \in \mathbb{N}} \sup_{m \in [0:M]} K_m < +\infty$, $S_0 := \mathbb{N} \cap [0, K_0 E_0)$, and $S_m = \mathbb{N} \cap [\sum_{k=0}^{m-1} K_k E_k, \sum_{k=0}^m K_k E_k)$ $(m \in [M])$.

[Polynomial growth BS and LR (27)] We have that

$$\sum_{t \in S_m} \eta_t = \sum_{t \in S_m} \left( a_2 m \left\lceil \frac{t}{\sum_{k=0}^m K_k E_k} \right\rceil + \eta_0 \right)^{c_2} \geq \sum_{t \in S_m} (a_2 m + \eta_0)^{c_2},$$

which, together with $\underline{\eta} = \min\{a_2, \eta_0\}$, implies that

$$\sum_{t \in S_m} \eta_t \geq \underline{\eta}^{c_2} \sum_{t \in S_m} (m+1)^{c_2} \geq \underline{\eta}^{c_2} K_{\min} E_{\min}(m+1)^{c_2}.$$

Hence,

$$\sum_{m=0}^{M} \sum_{t \in S_m} \eta_t \geq \underline{\eta}^{c_2} K_{\min} E_{\min} \sum_{m=1}^{M+1} m^{c_2} \geq \frac{\underline{\eta}^{c_2} K_{\min} E_{\min}}{1 + c_2} M^{1+c_2}.$$

We also have that

$$\sum_{t \in S_m} \frac{\eta_t^2}{b_t} = \sum_{t \in S_m} \frac{\left( a_2 m \left\lceil \frac{t}{\sum_{k=0}^{m} K_k E_k} \right\rceil + \eta_0 \right)^{2c_2}}{\left( a_1 m \left\lceil \frac{t}{\sum_{k=0}^{m} K_k E_k} \right\rceil + b_0 \right)^{c_1}} \leq \sum_{t \in S_m} \frac{(a_2 m + \eta_0)^{2c_2}}{(a_1 m + b_0)^{c_1}}.$$

Let $\overline{\eta} = \max\{a_2, \eta_0\}$ and $\underline{b} = \min\{a_1, b_0\}$. Then,

$$\sum_{m=0}^{M} \sum_{t \in S_m} \frac{\eta_t^2}{b_t} \leq K_{\max} E_{\max} \frac{\overline{\eta}^{2c_2}}{\underline{b}^{c_1}} \sum_{m=0}^{M} \frac{(m+1)^{2c_2}}{(m+1)^{c_1}} \leq K_{\max} E_{\max} \frac{\overline{\eta}^{2c_2}}{\underline{b}^{c_1}} \sum_{m=1}^{M+1} \frac{1}{m^{c_1 - 2c_2}}$$

$$\leq \frac{2 K_{\max} E_{\max} \overline{\eta}^{2c_2}}{\underline{b}^{c_1}}.$$

Hence,

$$B_T = \frac{1}{\sum_{t=0}^{T-1} \eta_t} \leq \frac{1 + c_2}{\underline{\eta}^{c_2} K_{\min} E_{\min} M^{1+c_2}}$$

and

$$V_T = \frac{1}{\sum_{t=0}^{T-1} \eta_t} \sum_{t=0}^{T-1} \frac{\eta_t^2}{b_t} \leq \frac{2 K_{\max} E_{\max} (1 + c_2) \overline{\eta}^{2c_2}}{K_{\min} E_{\min} \underline{\eta}^{c_2} \underline{b}^{c_1} M^{1+c_2}}.$$

[Exponential growth BS and LR (13)] We have that

$$\sum_{m=0}^{M} \sum_{t \in S_m} \eta_t = \sum_{m=0}^{M} \sum_{t \in S_m} \gamma^{m \left\lceil \frac{t}{\sum_{k=0}^{m} K_k E_k} \right\rceil} \eta_0 \geq \eta_0 K_{\min} E_{\min} \sum_{m=0}^{M} \gamma^m$$

$$= \eta_0 K_{\min} E_{\min} \frac{\gamma^M - 1}{\gamma - 1} > \frac{\eta_0 K_{\min} E_{\min} \gamma^M}{\gamma^2} > \frac{\eta_0 K_{\min} E_{\min} \gamma^M}{\delta}$$

and

$$\sum_{m=0}^{M} \sum_{t \in S_m} \frac{\eta_t^2}{b_t} = \sum_{m=0}^{M} \sum_{t \in S_m} \frac{\gamma^{2m \left\lceil \frac{t}{\sum_{k=0}^{m} K_k E_k} \right\rceil} \eta_0^2}{\delta^{m \left\lceil \frac{t}{\sum_{k=0}^{m} K_k E_k} \right\rceil} b_0} \leq K_{\max} E_{\max} \frac{\eta_0^2}{b_0} \sum_{m=0}^{M} \frac{\gamma^{2m}}{\delta^m}$$

$$\leq K_{\max} E_{\max} \frac{\eta_0^2}{b_0} \sum_{m=0}^{M} \left( \frac{\gamma^2}{\delta} \right)^m \leq K_{\max} E_{\max} \frac{\eta_0^2}{b_0} \frac{1}{1 - \hat{\gamma}},$$

where $\hat{\gamma} = \frac{\gamma^2}{\delta} < 1$. Hence,

$$B_T = \frac{1}{\sum_{t=0}^{T-1} \eta_t} \leq \frac{\delta}{\eta_0 K_{\min} E_{\min} \gamma^M}$$

and

$$V_T = \frac{1}{\sum_{t=0}^{T-1} \eta_t} \sum_{t=0}^{T-1} \frac{\eta_t^2}{b_t} \leq \frac{K_{\max} E_{\max} \eta_0 \delta}{K_{\min} E_{\min} b_0 (1 - \hat{\gamma}) \gamma^M}.$$

$\square$

*Proof of Theorem 3.4:* Theorem 3.4 follows immediately from Theorems 3.2 and 3.3. $\square$

## A.4 Proofs of Convergence Results under Convexity

Using Lemma 2.1, we have the following lemma.

**Lemma A.1** *Suppose that Assumption 2.1 holds and $f$ is convex and consider the sequence $(\boldsymbol{\theta}_t)$ generated by Algorithm 1 with $\eta_t \in [\eta_{\min}, \eta_{\max}] \subset [0, \frac{2}{\bar{L}})$ satisfying $\sum_{t=0}^{T-1} \eta_t \neq 0$, where $\bar{L} := \frac{1}{n}\sum_{i \in [n]} L_i$, $\underline{f}^\star := \frac{1}{n}\sum_{i \in [n]} f_i^\star$, $\boldsymbol{\theta}^\star \in \mathbb{R}^d$ is a global minimizer of $f$, and $f^\star := f(\boldsymbol{\theta}^\star)$. Then, for all $T \in \mathbb{N}$,*

$$
\min_{t \in [0:T-1]} \mathbb{E}\left[f(\boldsymbol{\theta}_t) - f^\star\right]
$$

$$
\leq \left(\frac{\|\boldsymbol{\theta}_0 - \boldsymbol{\theta}^\star\|^2}{2} + \frac{\eta_{\max}(f(\boldsymbol{\theta}_0) - \underline{f}^\star)}{2 - \bar{L}\eta_{\max}}\right) \underbrace{\frac{1}{\sum_{t=0}^{T-1} \eta_t}}_{B_T} + \frac{\sigma^2}{2}\left(1 + \frac{\bar{L}\eta_{\max}}{2 - \bar{L}\eta_{\max}}\right) \underbrace{\frac{\sum_{t=0}^{T-1} \eta_t^2 b_t^{-1}}{\sum_{t=0}^{T-1} \eta_t}}_{V_T},
$$

*where $\mathbb{E}$ denotes the total expectation, defined by $\mathbb{E} := \mathbb{E}_{\boldsymbol{\xi}_0}\mathbb{E}_{\boldsymbol{\xi}_1} \cdots \mathbb{E}_{\boldsymbol{\xi}_t}$.*

*Proof:* Since $\|\boldsymbol{\theta}_1 - \boldsymbol{\theta}_2\|^2 = \|\boldsymbol{\theta}_1\|^2 - 2\langle\boldsymbol{\theta}_1, \boldsymbol{\theta}_2\rangle + \|\boldsymbol{\theta}_2\|^2$ holds for all $\boldsymbol{\theta}_1, \boldsymbol{\theta}_2 \in \mathbb{R}^d$, we have that, for all $t \in \mathbb{N}$,

$$
\|\boldsymbol{\theta}_{t+1} - \boldsymbol{\theta}^\star\|^2 = \|(\boldsymbol{\theta}_t - \boldsymbol{\theta}^\star) - \eta_t \nabla f_{B_t}(\boldsymbol{\theta}_t)\|^2
$$

$$
= \|\boldsymbol{\theta}_t - \boldsymbol{\theta}^\star\|^2 - 2\eta_t\langle\boldsymbol{\theta}_t - \boldsymbol{\theta}^\star, \nabla f_{B_t}(\boldsymbol{\theta}_t)\rangle + \eta_t^2\|\nabla f_{B_t}(\boldsymbol{\theta}_t)\|^2.
$$

Taking expectation conditioned on $\boldsymbol{\xi}_{t-1} = \hat{\boldsymbol{\xi}}_{t-1}$ on both sides of the above equation, Proposition A.1 and (19) ensure that, for all $t \in \mathbb{N}$,

$$
\mathbb{E}_{\boldsymbol{\xi}_t}\left[\|\boldsymbol{\theta}_{t+1} - \boldsymbol{\theta}^\star\|^2\Big|\hat{\boldsymbol{\xi}}_{t-1}\right]
$$

$$
= \|\boldsymbol{\theta}_t - \boldsymbol{\theta}^\star\|^2 - 2\eta_t\mathbb{E}_{\boldsymbol{\xi}_t}\left[\langle\boldsymbol{\theta}_t - \boldsymbol{\theta}^\star, \nabla f_{B_t}(\boldsymbol{\theta}_t)\rangle\Big|\hat{\boldsymbol{\xi}}_{t-1}\right] + \eta_t^2\mathbb{E}_{\boldsymbol{\xi}_t}\left[\|\nabla f_{B_t}(\boldsymbol{\theta}_t)\|^2\Big|\hat{\boldsymbol{\xi}}_{t-1}\right]
$$

$$
\leq \|\boldsymbol{\theta}_t - \boldsymbol{\theta}^\star\|^2 - 2\eta_t\langle\boldsymbol{\theta}_t - \boldsymbol{\theta}^\star, \nabla f(\boldsymbol{\theta}_t)\rangle + \eta_t^2\left(\frac{\sigma^2}{b_t} + \|\nabla f(\boldsymbol{\theta}_t)\|^2\right).
$$

Since convexity of $f$ implies that, for all $\boldsymbol{\theta}_1, \boldsymbol{\theta}_2 \in \mathbb{R}^d$, $f(\boldsymbol{\theta}_1) \geq f(\boldsymbol{\theta}_2) + \langle\boldsymbol{\theta}_1 - \boldsymbol{\theta}_2, \nabla f(\boldsymbol{\theta}_2)\rangle$, we have that

$$
\mathbb{E}_{\boldsymbol{\xi}_t}\left[\|\boldsymbol{\theta}_{t+1} - \boldsymbol{\theta}^\star\|^2\Big|\hat{\boldsymbol{\xi}}_{t-1}\right] \leq \|\boldsymbol{\theta}_t - \boldsymbol{\theta}^\star\|^2 - 2\eta_t(f(\boldsymbol{\theta}_t) - f^\star) + \eta_t^2\left(\frac{\sigma^2}{b_t} + \|\nabla f(\boldsymbol{\theta}_t)\|^2\right).
$$

Taking the total expectation on both sides of the above inequality gives that, for all $t \in \mathbb{N}$,

$$
\mathbb{E}\left[\|\boldsymbol{\theta}_{t+1} - \boldsymbol{\theta}^\star\|^2\right] \leq \mathbb{E}\left[\|\boldsymbol{\theta}_t - \boldsymbol{\theta}^\star\|^2\right] - 2\eta_t\mathbb{E}\left[f(\boldsymbol{\theta}_t) - f^\star\right] + \eta_t^2\left(\frac{\sigma^2}{b_t} + \mathbb{E}\left[\|\nabla f(\boldsymbol{\theta}_t)\|^2\right]\right).
$$

Let $T \in \mathbb{N}$. Summing the above inequality from $t = 0$ to $t = T - 1$ gives that

$$
2\sum_{t=0}^{T-1} \eta_t\mathbb{E}\left[f(\boldsymbol{\theta}_t) - f^\star\right] \leq \|\boldsymbol{\theta}_0 - \boldsymbol{\theta}^\star\|^2 + \sigma^2\sum_{t=0}^{T-1} \frac{\eta_t^2}{b_t} + \eta_{\max}\sum_{t=0}^{T-1} \eta_t\mathbb{E}\left[\|\nabla f(\boldsymbol{\theta}_t)\|^2\right],
$$

where $\eta_t \leq \eta_{\max}$ is used. Since Lemma 2.1 (see (20)) guarantees that

$$\sum_{t=0}^{T-1} \eta_t \mathbb{E}\left[\|\nabla f(\boldsymbol{\theta}_t)\|^2\right] \leq \frac{2(f(\boldsymbol{\theta}_0) - \underline{f}^\star)}{2 - \bar{L}\eta_{\max}} + \frac{\bar{L}\sigma^2}{2 - \bar{L}\eta_{\max}} \sum_{t=0}^{T-1} \frac{\eta_t^2}{b_t},$$

we have that

$$\sum_{t=0}^{T-1} \eta_t \mathbb{E}\left[f(\boldsymbol{\theta}_t) - f^\star\right] \leq \frac{\|\boldsymbol{\theta}_0 - \boldsymbol{\theta}^\star\|^2}{2} + \frac{\eta_{\max}(f(\boldsymbol{\theta}_0) - \underline{f}^\star)}{2 - \bar{L}\eta_{\max}} + \frac{\sigma^2}{2}\left(1 + \frac{\bar{L}\eta_{\max}}{2 - \bar{L}\eta_{\max}}\right) \sum_{t=0}^{T-1} \frac{\eta_t^2}{b_t}.$$

Therefore, from $\sum_{t=0}^{T-1} \eta_t \neq 0$, we have

$$\min_{t \in [0:T-1]} \mathbb{E}\left[f(\boldsymbol{\theta}_t) - f^\star\right]$$

$$\leq \left(\frac{\|\boldsymbol{\theta}_0 - \boldsymbol{\theta}^\star\|^2}{2} + \frac{\eta_{\max}(f(\boldsymbol{\theta}_0) - \underline{f}^\star)}{2 - \bar{L}\eta_{\max}}\right) \frac{1}{\sum_{t=0}^{T-1} \eta_t} + \frac{\sigma^2}{2}\left(1 + \frac{\bar{L}\eta_{\max}}{2 - \bar{L}\eta_{\max}}\right) \frac{\sum_{t=0}^{T-1} \eta_t^2 b_t^{-1}}{\sum_{t=0}^{T-1} \eta_t},$$

which completes the proof. □

The right column in Table 2 can be obtained by using Lemma A.1 and Theorems 3.1–3.4.

## A.5 Training ResNet-34 on ImageNet: Evaluating Case (iv)

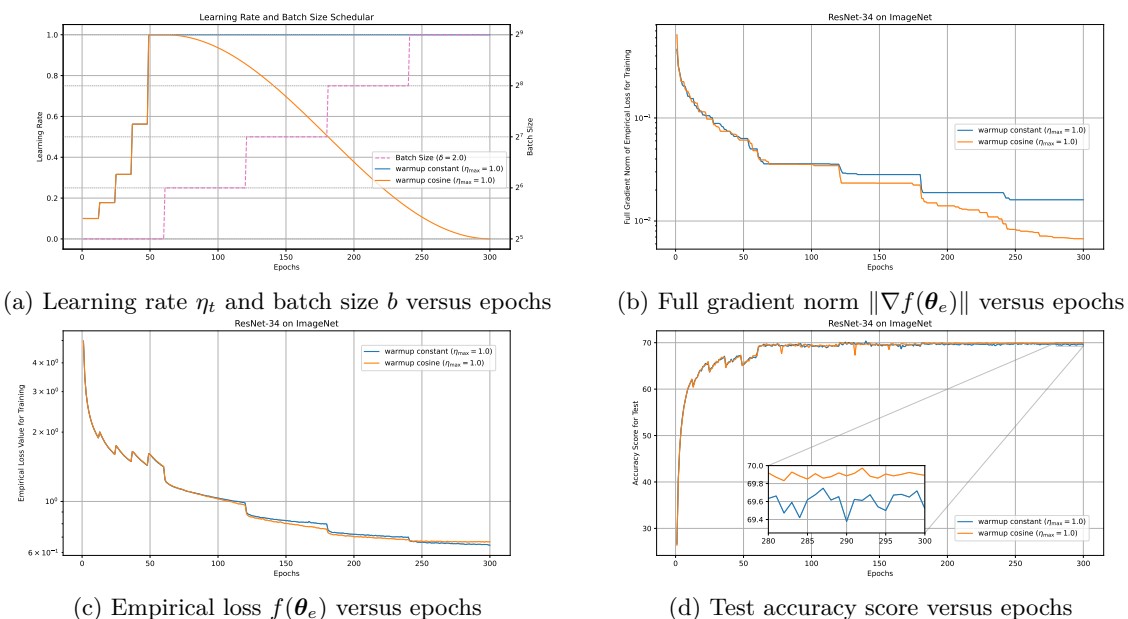

(a) Learning rate $\eta_t$ and batch size $b$ versus epochs

(b) Full gradient norm $\|\nabla f(\boldsymbol{\theta}_e)\|$ versus epochs

(c) Empirical loss $f(\boldsymbol{\theta}_e)$ versus epochs

(d) Test accuracy score versus epochs

Figure 7: (a) Warm-up decaying learning rates ($\eta_{\max} = 1.0$) and increasing batch sizes based on $\delta = 2$, (b) full gradient norm of empirical loss, (c) empirical loss value, and (d) accuracy score in testing for SGD to train ResNet-34 on ImageNet dataset.

## A.6 Training ResNet-18 on CIFAR10 and CIFAR100 using Doubling, Tripling, and Quadrupling Batch Sizes

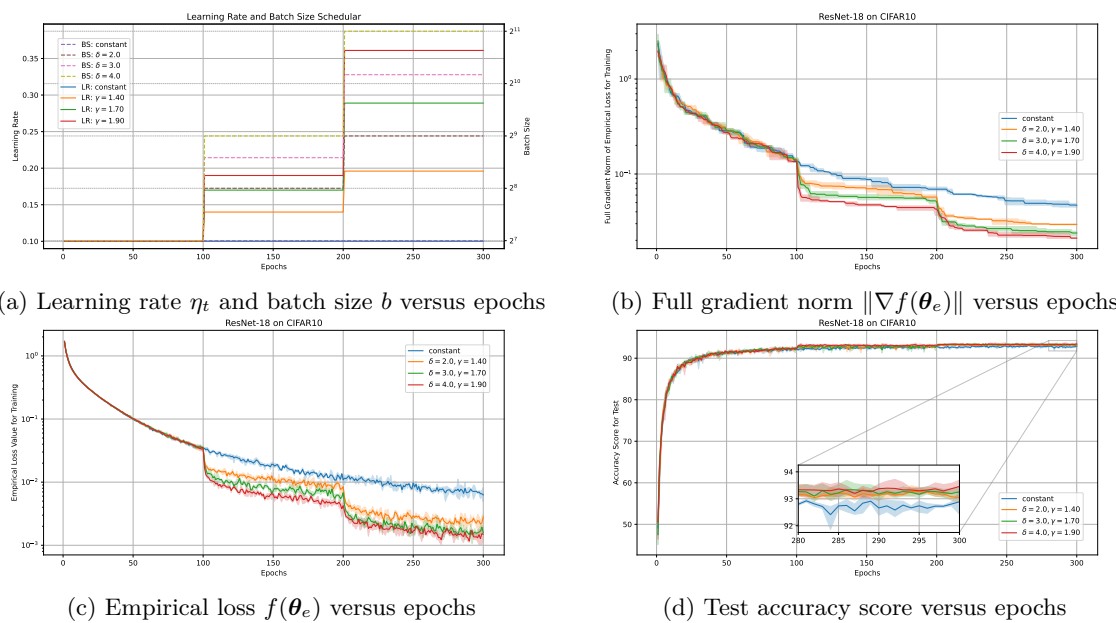

(a) Learning rate $\eta_t$ and batch size $b$ versus epochs

(b) Full gradient norm $\|\nabla f(\boldsymbol{\theta}_e)\|$ versus epochs

(c) Empirical loss $f(\boldsymbol{\theta}_e)$ versus epochs

(d) Test accuracy score versus epochs

Figure 8: (a) Increasing learning rates and doubling, tripling, and quadrupling batch sizes $((\delta, \gamma) = (2, 1.4), (3, 1.7), (4, 1.9)$ satisfying $\sqrt{\delta} > \gamma)$ every 100 epochs, (b) full gradient norm of empirical loss, (c) empirical loss value, and (d) accuracy score in testing for SGD to train ResNet-18 on CIFAR10 dataset.

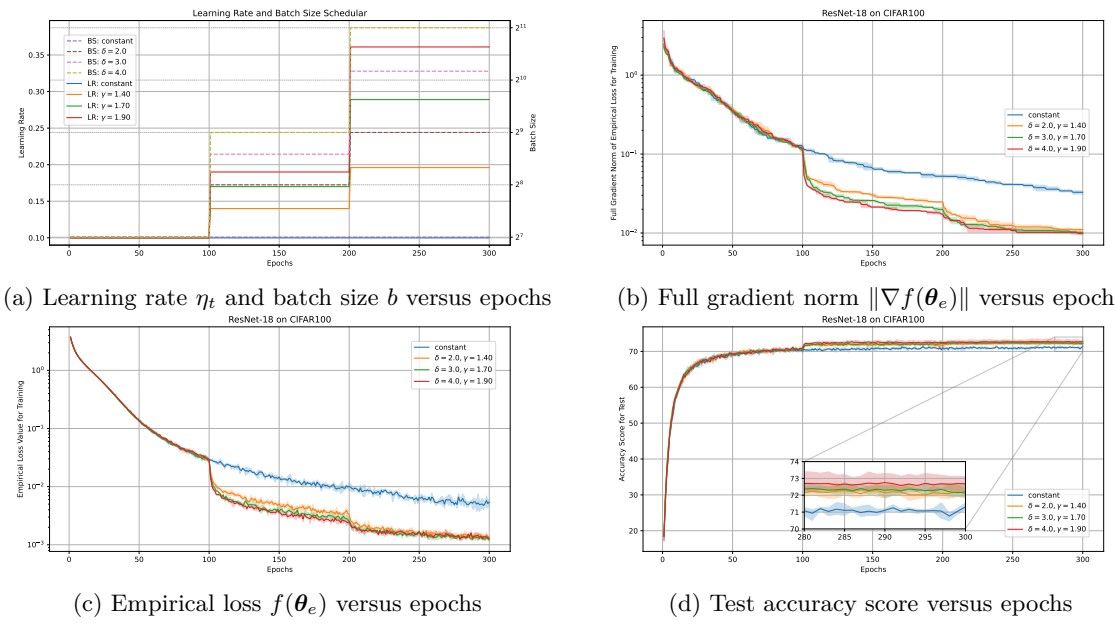

(a) Learning rate $\eta_t$ and batch size $b$ versus epochs

(b) Full gradient norm $\|\nabla f(\boldsymbol{\theta}_e)\|$ versus epochs

(c) Empirical loss $f(\boldsymbol{\theta}_e)$ versus epochs

(d) Test accuracy score versus epochs

Figure 9: (a) Increasing learning rates and doubling, tripling, and quadrupling batch sizes $((\delta, \gamma) = (2, 1.4), (3, 1.7), (4, 1.9)$ satisfying $\sqrt{\delta} > \gamma)$ every 100 epochs, (b) full gradient norm of empirical loss, (c) empirical loss value, and (d) accuracy score in testing for SGD to train ResNet-18 on CIFAR100 dataset.

### A.7 Comparisons of Case (ii) with Cases (iii) and (iv) for Training ResNet-18 on CIFAR100 using Increasing Batch Size based on $\delta = 3$

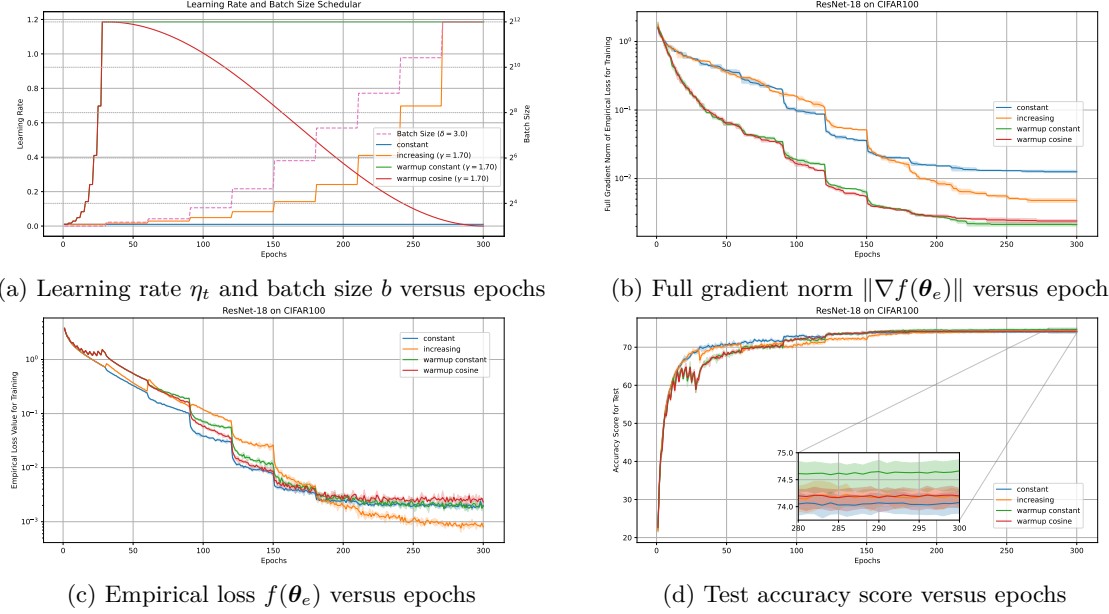

(a) Learning rate $\eta_t$ and batch size $b$ versus epochs

(b) Full gradient norm $\|\nabla f(\boldsymbol{\theta}_e)\|$ versus epochs

(c) Empirical loss $f(\boldsymbol{\theta}_e)$ versus epochs

(d) Test accuracy score versus epochs

Figure 10: (a) Increasing or warm-up decaying learning rates ($\eta_{\min} = 0.01$) and increasing batch sizes based on $\delta = 3$, (b) full gradient norm of empirical loss, (c) empirical loss value, and (d) accuracy score in testing for SGD to train ResNet-18 on CIFAR100 dataset.

Figures 2–4 compare Case (ii) with Cases (iii) and (iv) for training ResNet-18 on CIFAR100 using increasing batch size based on $\delta = 2$.

### A.8 Comparisons of With vs. Without Replacement for Training ResNet-18 on CIFAR100

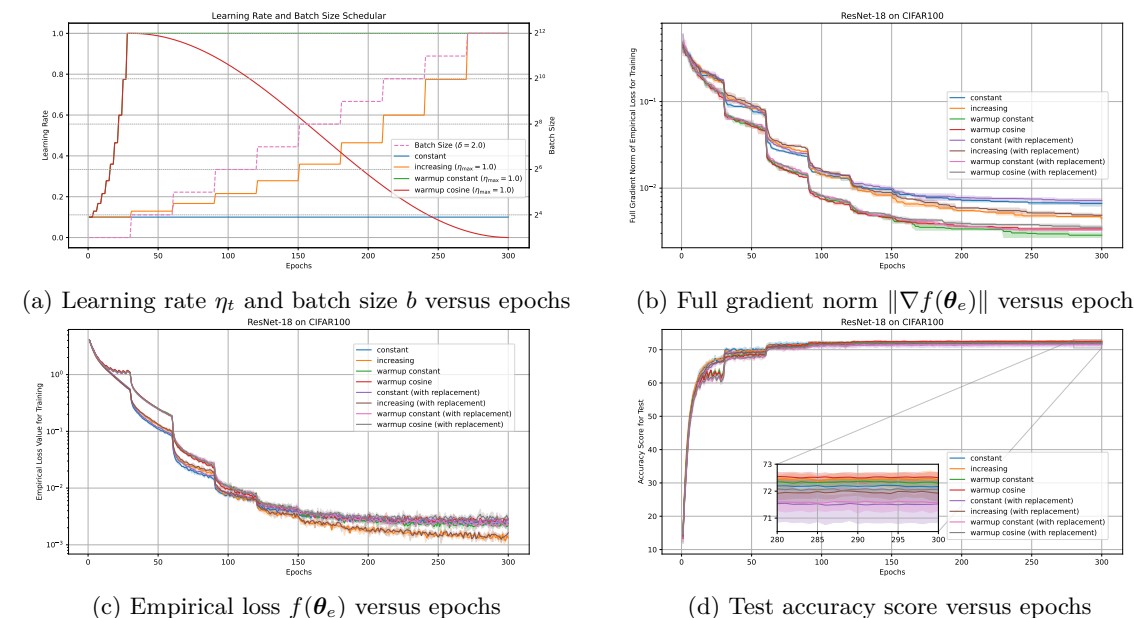

(a) Learning rate $\eta_t$ and batch size $b$ versus epochs

(b) Full gradient norm $\|\nabla f(\boldsymbol{\theta}_e)\|$ versus epochs

(c) Empirical loss $f(\boldsymbol{\theta}_e)$ versus epochs

(d) Test accuracy score versus epochs

Figure 11: (a) Increasing or warm-up decaying learning rates ($\eta_{\max} = 1.0$) and increasing batch sizes based on $\delta = 2$, (b) full gradient norm of empirical loss, (c) empirical loss value, and (d) accuracy score in testing for SGD to train ResNet-18 on CIFAR100 dataset.

## A.9 Training Wide-ResNet-28-10 on CIFAR100

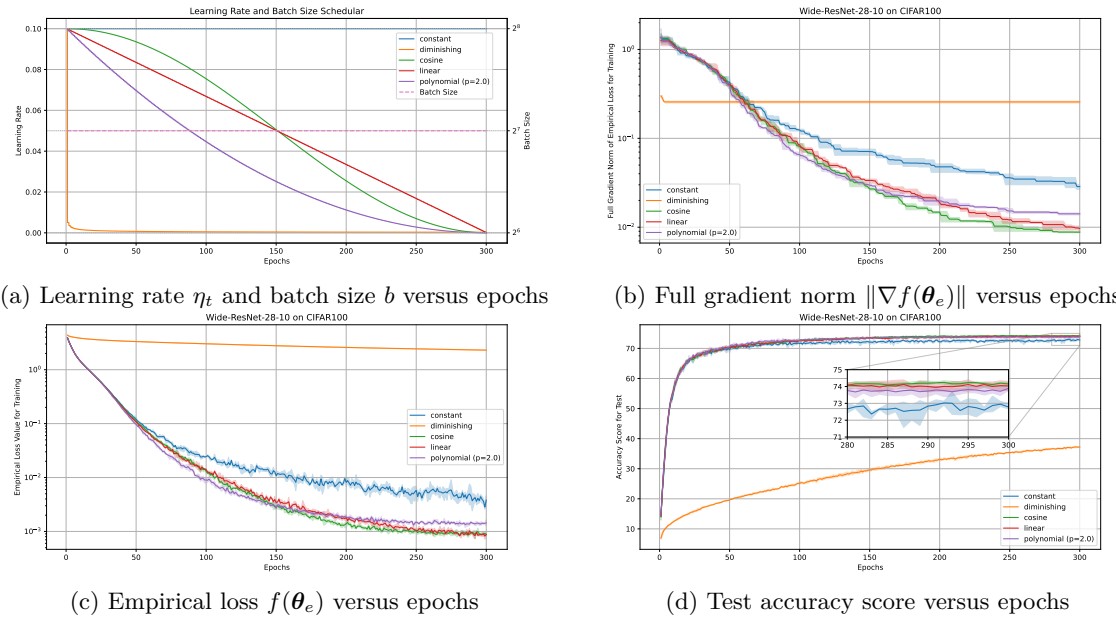

(a) Learning rate $\eta_t$ and batch size $b$ versus epochs

(b) Full gradient norm $\|\nabla f(\boldsymbol{\theta}_e)\|$ versus epochs

(c) Empirical loss $f(\boldsymbol{\theta}_e)$ versus epochs

(d) Test accuracy score versus epochs

Figure 12: (a) Decaying learning rates (constant, diminishing, cosine, linear, and polynomial) and constant batch size, (b) full gradient norm of empirical loss, (c) empirical loss value, and (d) accuracy score in testing for SGD to train Wide-ResNet-28-10 on CIFAR100 dataset.

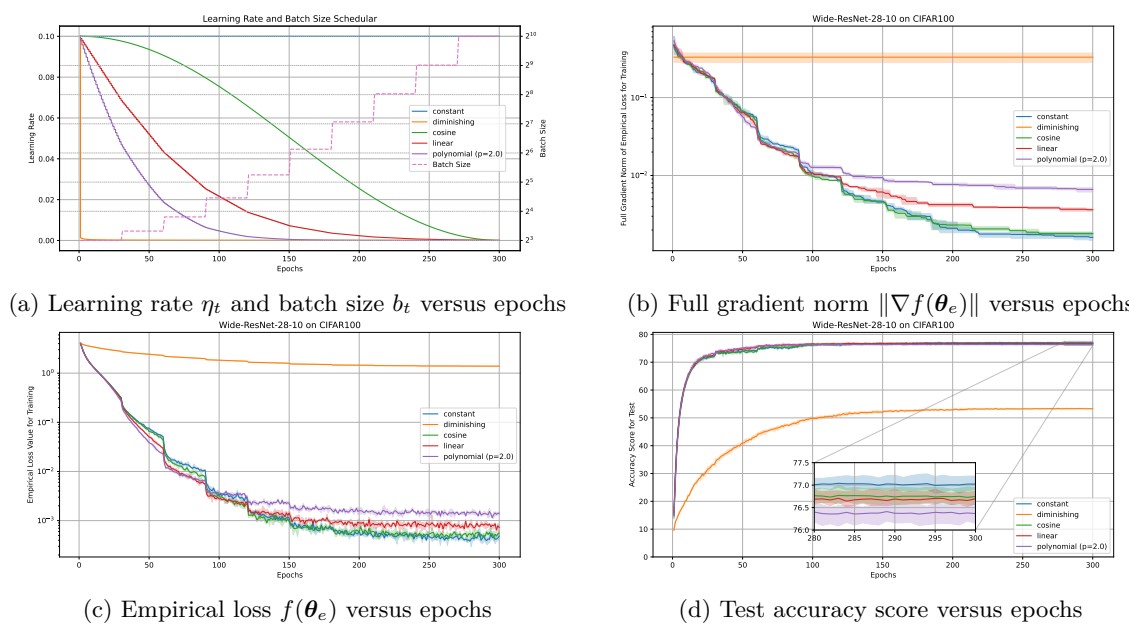

(a) Learning rate $\eta_t$ and batch size $b_t$ versus epochs

(b) Full gradient norm $\|\nabla f(\boldsymbol{\theta}_e)\|$ versus epochs

(c) Empirical loss $f(\boldsymbol{\theta}_e)$ versus epochs

(d) Test accuracy score versus epochs

Figure 13: (a) Decaying learning rates and increasing batch size every 30 epochs, (b) full gradient norm of empirical loss, (c) empirical loss value, and (d) accuracy score in testing for SGD to train Wide-ResNet-28-10 on CIFAR100 dataset.

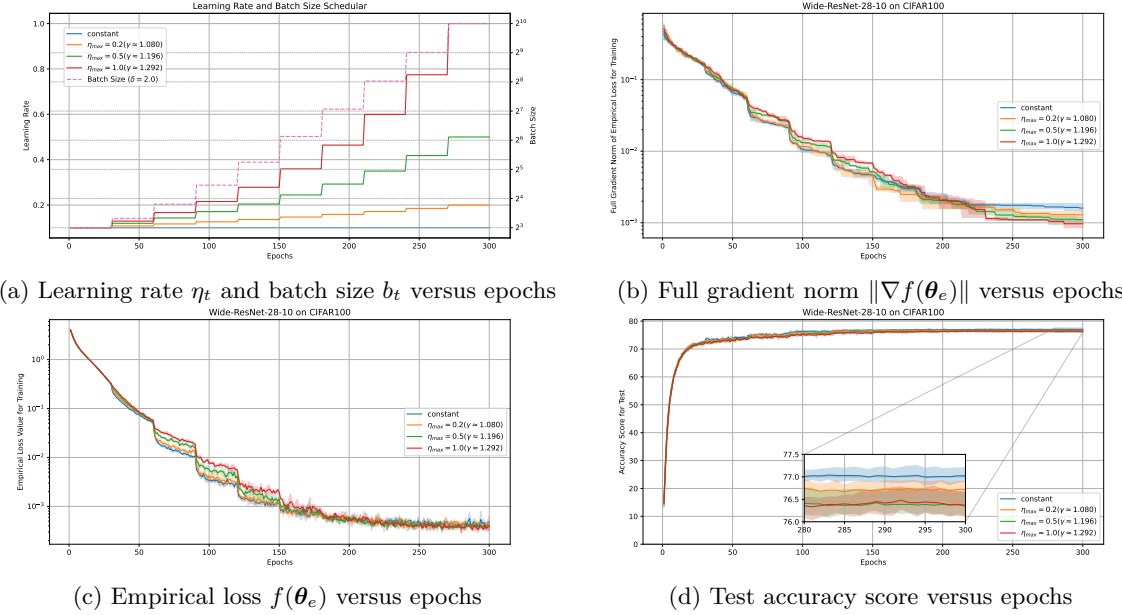

(a) Learning rate $\eta_t$ and batch size $b_t$ versus epochs

(b) Full gradient norm $\|\nabla f(\boldsymbol{\theta}_e)\|$ versus epochs

(c) Empirical loss $f(\boldsymbol{\theta}_e)$ versus epochs

(d) Test accuracy score versus epochs

Figure 14: (a) Increasing learning rates ($\eta_{\max} = 0.2, 0.5, 1.0$) and increasing batch size every 30 epochs, (b) full gradient norm of empirical loss, (c) empirical loss value, and (d) accuracy score in testing for SGD to train Wide-ResNet-28-10 on CIFAR100 dataset.

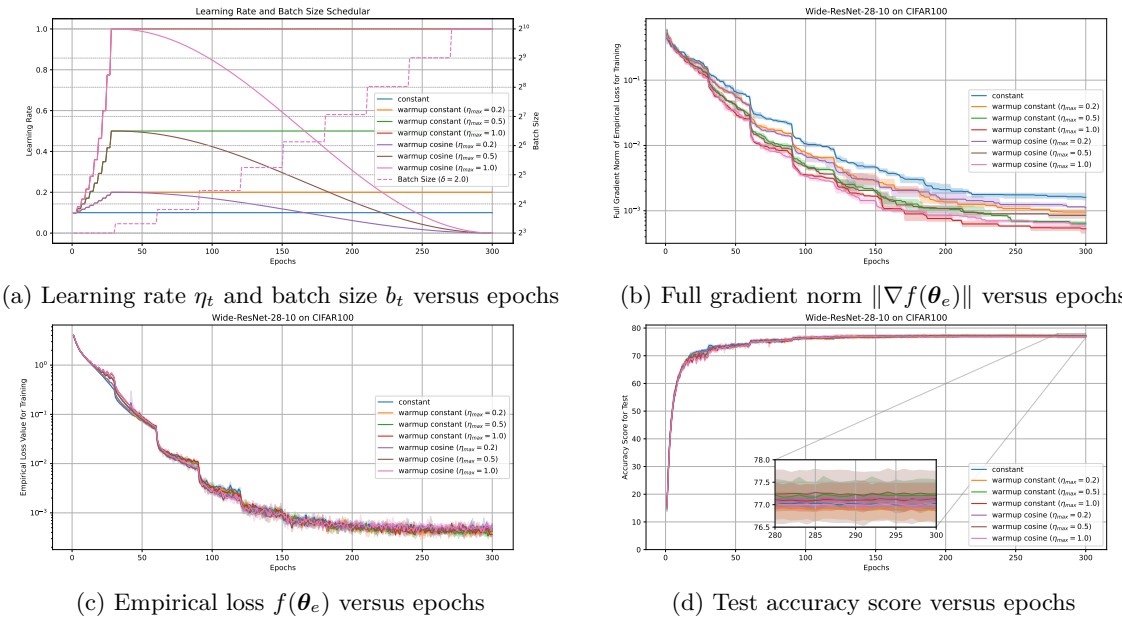

(a) Learning rate $\eta_t$ and batch size $b_t$ versus epochs

(b) Full gradient norm $\|\nabla f(\boldsymbol{\theta}_e)\|$ versus epochs

(c) Empirical loss $f(\boldsymbol{\theta}_e)$ versus epochs

(d) Test accuracy score versus epochs

Figure 15: (a) Warm-up learning rates and increasing batch size every 30 epochs, (b) full gradient norm of empirical loss, (c) empirical loss value, and (d) accuracy score in testing for SGD to train Wide-ResNet-28-10 on CIFAR100 dataset.

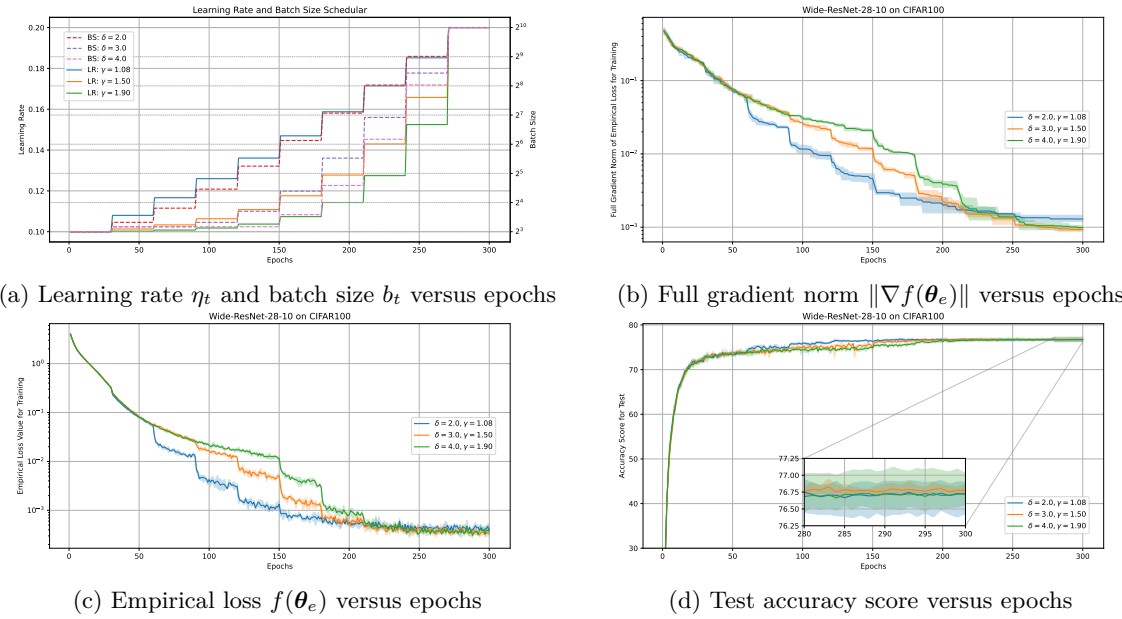

(a) Learning rate $\eta_t$ and batch size $b_t$ versus epochs

(b) Full gradient norm $\|\nabla f(\boldsymbol{\theta}_e)\|$ versus epochs

(c) Empirical loss $f(\boldsymbol{\theta}_e)$ versus epochs

(d) Test accuracy score versus epochs

Figure 16: (a) Increasing learning rates and increasing batch sizes based on $\delta = 2, 3, 4$, (b) full gradient norm of empirical loss, (c) empirical loss value, and (d) accuracy score in testing for SGD to train Wide-ResNet-28-10 on CIFAR100 dataset.

## A.10 Training ResNet-18 on Tiny ImageNet

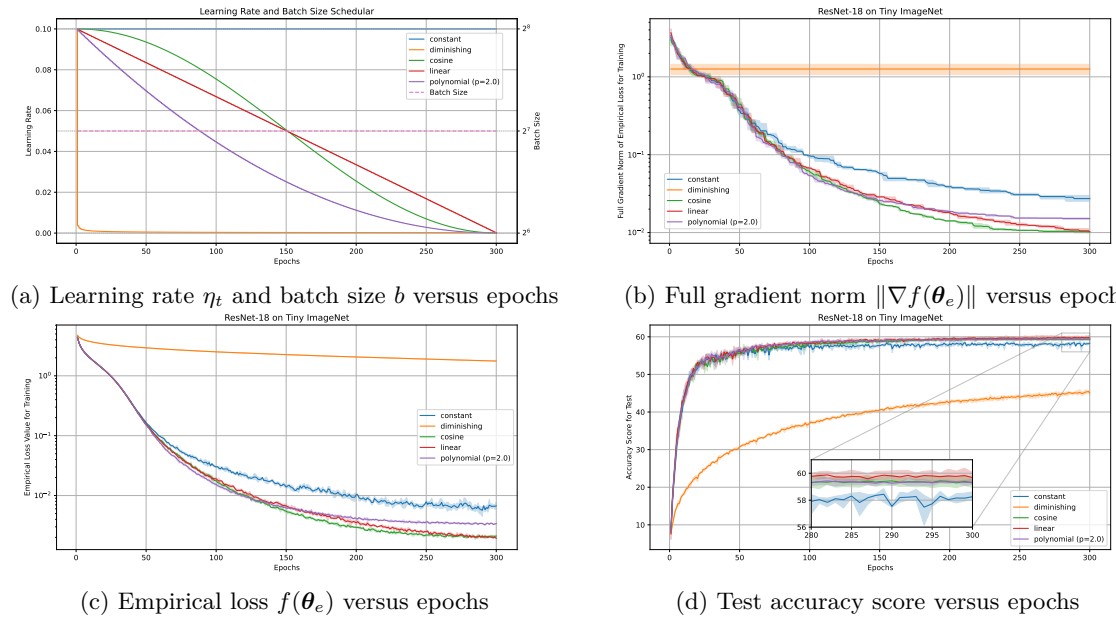

(a) Learning rate $\eta_t$ and batch size $b$ versus epochs

(b) Full gradient norm $\|\nabla f(\boldsymbol{\theta}_e)\|$ versus epochs

(c) Empirical loss $f(\boldsymbol{\theta}_e)$ versus epochs

(d) Test accuracy score versus epochs

Figure 17: (a) Decaying learning rates (constant, diminishing, cosine, linear, and polynomial) and constant batch size, (b) full gradient norm of empirical loss, (c) empirical loss value, and (d) accuracy score in testing for SGD to train ResNet-18 on Tiny ImageNet dataset.

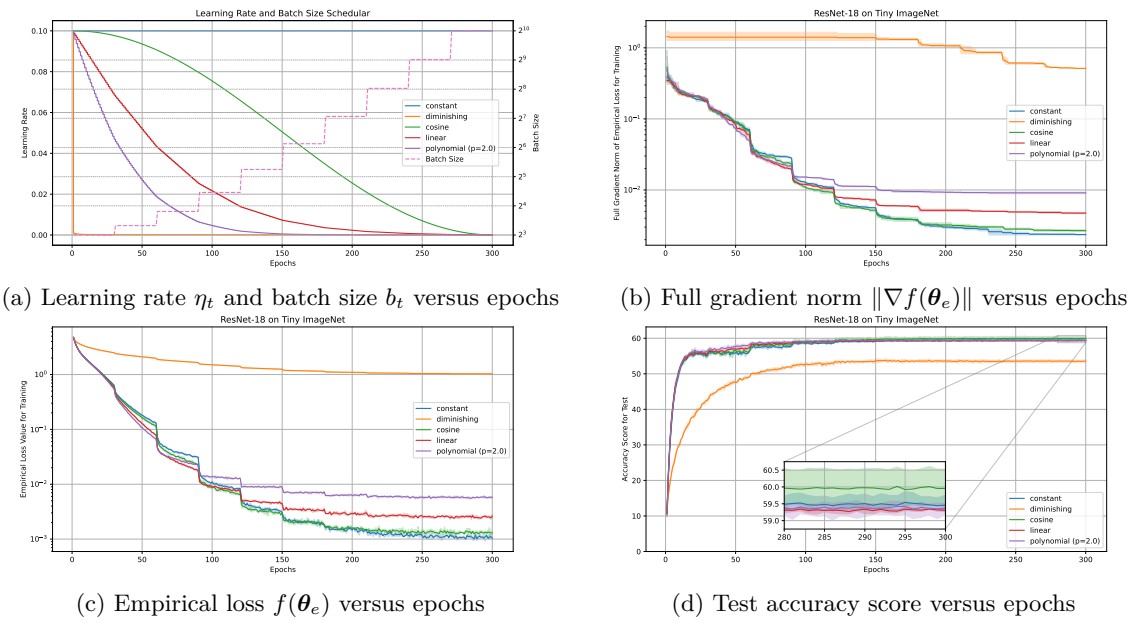

(a) Learning rate $\eta_t$ and batch size $b_t$ versus epochs

(b) Full gradient norm $\|\nabla f(\boldsymbol{\theta}_e)\|$ versus epochs

(c) Empirical loss $f(\boldsymbol{\theta}_e)$ versus epochs

(d) Test accuracy score versus epochs

Figure 18: (a) Decaying learning rates and increasing batch size every 30 epochs, (b) full gradient norm of empirical loss, (c) empirical loss value, and (d) accuracy score in testing for SGD to train ResNet-18 on Tiny ImageNet dataset.

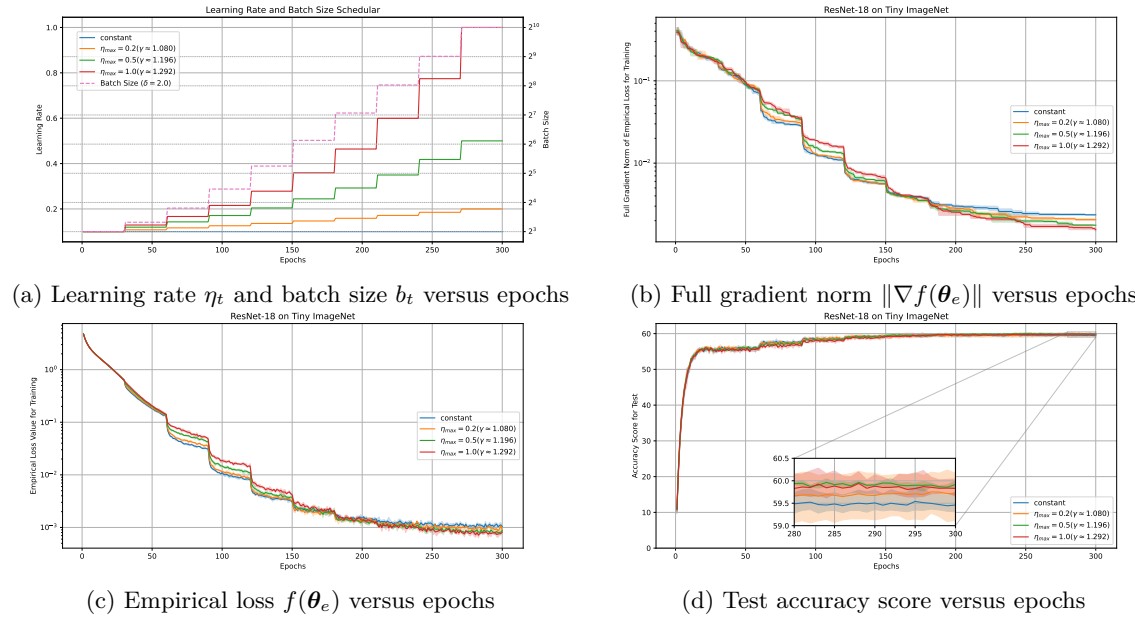

(a) Learning rate $\eta_t$ and batch size $b_t$ versus epochs

(b) Full gradient norm $\|\nabla f(\boldsymbol{\theta}_e)\|$ versus epochs

(c) Empirical loss $f(\boldsymbol{\theta}_e)$ versus epochs

(d) Test accuracy score versus epochs

Figure 19: (a) Increasing learning rates ($\eta_{\max} = 0.2, 0.5, 1.0$) and increasing batch size every 30 epochs, (b) full gradient norm of empirical loss, (c) empirical loss value, and (d) accuracy score in testing for SGD to train ResNet-18 on Tiny ImageNet dataset.

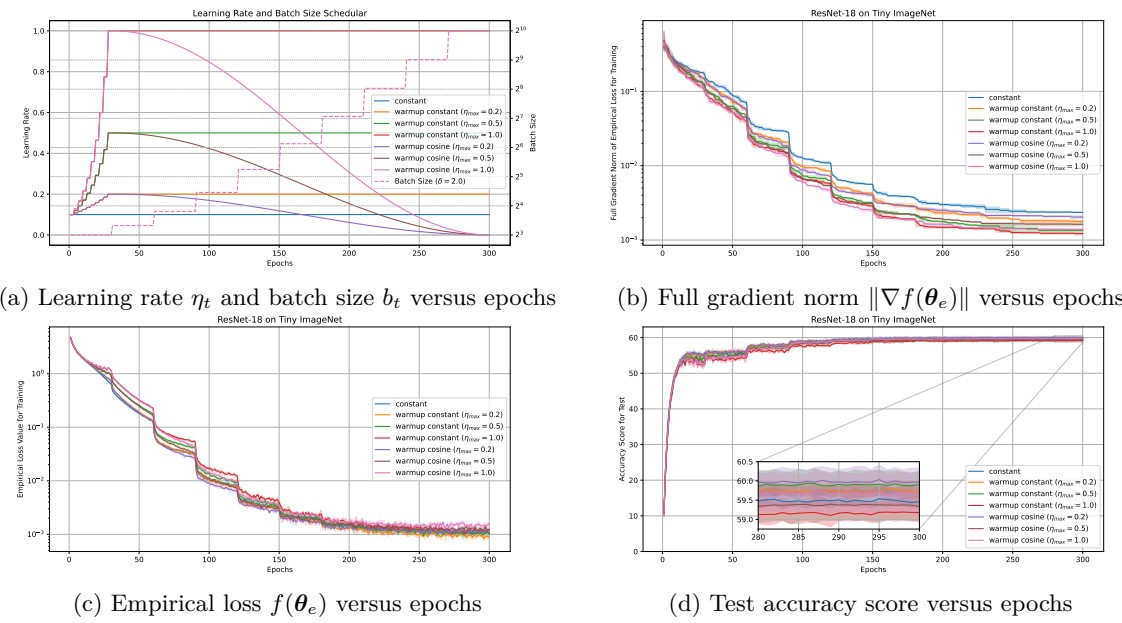

(a) Learning rate $\eta_t$ and batch size $b_t$ versus epochs

(b) Full gradient norm $\|\nabla f(\boldsymbol{\theta}_e)\|$ versus epochs

(c) Empirical loss $f(\boldsymbol{\theta}_e)$ versus epochs

(d) Test accuracy score versus epochs

Figure 20: (a) Warm-up learning rates and increasing batch size every 30 epochs, (b) full gradient norm of empirical loss, (c) empirical loss value, and (d) accuracy score in testing for SGD to train ResNet-18 on Tiny ImageNet dataset.

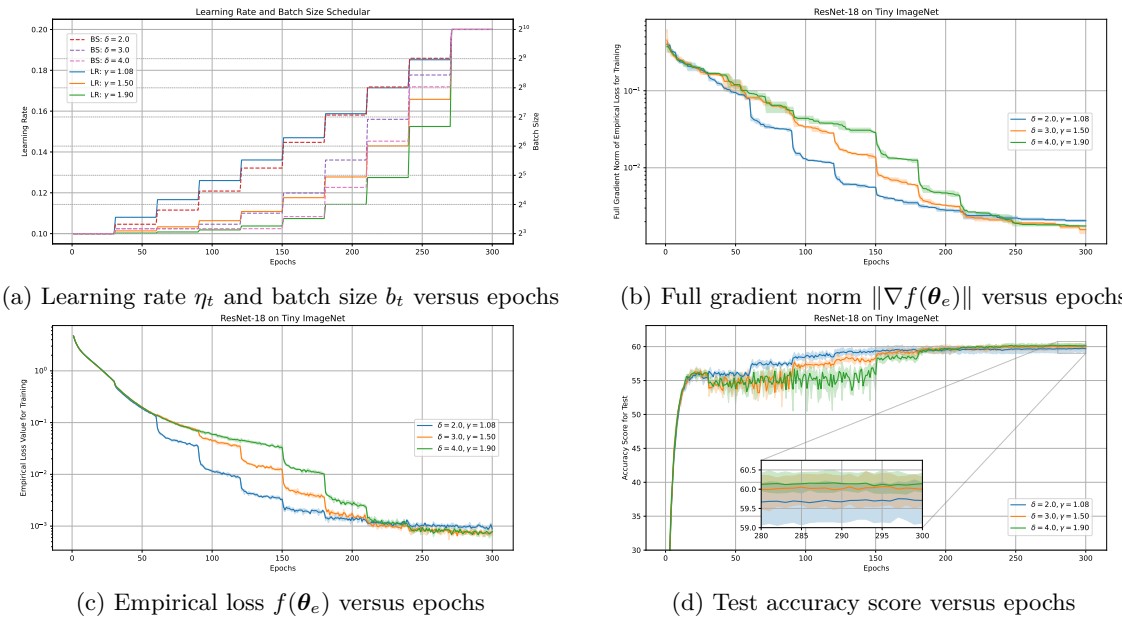

(a) Learning rate $\eta_t$ and batch size $b_t$ versus epochs

(b) Full gradient norm $\|\nabla f(\boldsymbol{\theta}_e)\|$ versus epochs

(c) Empirical loss $f(\boldsymbol{\theta}_e)$ versus epochs

(d) Test accuracy score versus epochs

Figure 21: (a) Increasing learning rates and increasing batch sizes based on $\delta = 2, 3, 4$, (b) full gradient norm of empirical loss, (c) empirical loss value, and (d) accuracy score in testing for SGD to train ResNet-18 on Tiny ImageNet dataset.

