# OpenReview forum: "Increasing Both Batch Size and Learning Rate Accelerates Stochastic Gradient Descent"
_TMLR — Accepted by TMLR_

### Review · Reviewer_onDZ · 2024-12-02

**Summary Of Contributions:**

This paper explores the theoretical and empirical impacts of different scheduling strategies for mini-batch stochastic gradient descent (SGD), focusing on variations in batch size and learning rate. Key contributions include:

1. Theoretical Analysis:

The paper provides a detailed examination of four scheduling strategies for batch size and learning rates, backed by rigorous theoretical analyses. For each strategy, the authors derive explicit inequalities that bound the gradient norm and explain the conditions under which convergence can be accelerated.

2. Practical Insights:

Demonstrates that increasing both batch size and learning rates can accelerate mini-batch SGD.

Highlights how warm-up in strategy (iv) ensure convergence with faster empirical loss reduction, combining the benefits of early acceleration and stable convergence.

3. Empirical Validation:

Experiments using ResNet-18 on the CIFAR-100 dataset confirm the theoretical findings.

Visualizations effectively illustrate the advantages of combining batch size and learning rate schedules, showing faster convergence and improved test accuracy compared to standard methods.

**Audience:**

Yes

**Broader Impact Concerns:**

This is a theoretical work. There is no such concern.

**Claims And Evidence:**

Yes

**Requested Changes:**

Please see weaknesses and address the raised concerns.

**Strengths And Weaknesses:**

Strengths:

1. Broad Applicability Without Convexity Assumption:

The paper provides theoretical guarantees and convergence analysis without relying on the often unrealistic assumption of convexity. This makes the results applicable to a broader range of real-world deep learning problems, which frequently involve nonconvex loss functions.

2. Thorough Theoretical Analysis of Scheduling Strategies:

The theoretical exploration of four distinct scheduling strategies for batch size and learning rates is comprehensive, offering rigorous mathematical insights supported by explicit inequalities. This analysis bridges theoretical advancements with practical utility.

3. Effective Presentation of Results:

The use of clear figures, tables, and theoretical bounds effectively conveys the impact of the different scheduling strategies, aiding understanding for both theoretical and applied audiences.

Weaknesses:

1. Independence Assumption in Assumption A3:

The independence assumption on $\xi_1, \dots, \xi_b$ could be replaced with a more practical setting where $\xi_1, \dots, \xi_b$ are drawn identically but with replacement. Under this modification, the conditional variance result in Proposition A.1 would require slight adjustments. Specifically:

a) When $b = n$, the conditional variance becomes exactly zero.

b) When $b \ll n$, the i.i.d. assumption remains a good approximation to the case of sampling with replacement.

Clarifying these distinctions and addressing the practical implications would strengthen the paper.

2. Comparison of Convergence Rates With and Without Convexity:

While the paper avoids assuming convexity, it would benefit from a direct comparison of the convergence rates under convex and nonconvex assumptions. Highlighting the differences would provide valuable insights into the trade-offs and implications of removing convexity assumptions.

3. Proofs of Proposition A.1 and Lemma 2.1 Rely on Established Results:

The proofs appear to depend heavily on standard results from the literature on SGD analysis. Including more detailed citations for foundational work would enhance the paper's scholarly rigor and appropriately acknowledge prior contributions. Additionally, clearly specifying the original contributions in the analysis would help distinguish this work from existing approaches.

---

> ### Author Response · Authors · 2024-12-27
> **Replies to Reviewer onDZ's comments**
>
> We appreciate your detailed assessments and helpful feedback. We have revised the manuscript to incorporate all of the recommendations, which has resulted in an improved presentation of our work. The revised parts of the manuscript are marked in red.
>
> Weaknesses and Requested Changes:
>
> **Comment 1 (Independence Assumption in Assumption A3):**
> The independence assumption on $\xi_1, …, \xi_b$ could be replaced with a more practical setting where $\xi_1, …, \xi_b$ are drawn identically but with replacement. Under this modification, the conditional variance result in Proposition A.1 would require slight adjustments. Specifically:
>
> a) When $b = n$, the conditional variance becomes exactly zero.
>
> b) When $b \ll n$, the i.i.d. assumption remains a good approximation to the case of sampling with replacement.
>
> Clarifying these distinctions and addressing the practical implications would strengthen the paper.
>
> **Reply:**
> Thank you for pointing out the limitations of the independence assumption in Assumption A3. To address this, we have clarified the assumptions and added a detailed discussion in Appendix A.1. Specifically, we now state:
>
> Assumption 2.1(A3) holds under sampling with replacement, where ${\xi} = (\xi\_{1}, \xi\_{2}, \cdots, \xi\_{b})^\top$ are independent and identically distributed (i.i.d.). In practice, however, sampling without replacement is commonly used to improve data efficiency and diversity. Under sampling without replacement, minor dependencies arise between samples, particularly when $b \approx n$. These dependencies reduce the conditional variance $\mathbb{V}\_{{\xi}\_t}[\nabla f\_{B_t}({\theta}\_t) |\hat{{\xi}}\_{t-1}]$, resulting in behavior that closely resembles full-batch gradient computation.On the other hand, when $b \ll n$, the dependencies introduced by sampling without replacement become negligible, and the behavior of the mini-batch gradient closely approximates that of i.i.d. samples. Under such conditions, the theoretical properties of Assumption 2.1(A3) are approximately satisfied.Therefore, in large-scale datasets, the approximation of i.i.d. is sufficient to ensure that the theoretical predictions remain valid under sampling without replacement.The experimental results (Appendix A.8) further confirm the validity of this approximation, as the observed behavior aligns closely with the theoretical predictions.
>
> Additionally, we have included new experiments in Appendix A.8 to compare sampling with and without replacement. These experiments use the CIFAR100 dataset, which contains 50000 training samples and batch size was increased up to 4096. The results show that, for a batch size of 4096 ($b \ll 50000$), the outcomes are nearly identical for both sampling methods, validating the practicality of the i.i.d. approximation.

---

> ### Author Response · Authors · 2024-12-27
> **Replies to Reviewer onDZ's comments**
>
> **Comment 2 (Comparison of Convergence Rates With and Without Convexity):**
> While the paper avoids assuming convexity, it would benefit from a direct comparison of the convergence rates under convex and nonconvex assumptions. Highlighting the differences would provide valuable insights into the trade-offs and implications of removing convexity assumptions.
>
> **Reply:**
> Thank you for suggesting the inclusion of a comparison between convergence rates under convex and nonconvex conditions. In response, we have added a new section, Section 3.6, titled "Comparisons of convergence rates under nonconvexity with ones under convexity." This section summarizes and compares the convergence rates under these two conditions. Specifically:
>
> While Sections 3.1 - 3.5 consider the case where $f$ is not always convex, this section considers the case where $f$ is convex and compares convergence rates under nonconvexity condition with ones under convexity condition. Table 2 summarizes the convergence rates under nonconvexity and convexity conditions. The left column in Table 2 is obtained by the results in Sections 3.1 - 3.4 (see also the table in Section 1). The well-known performance measure of Algorithm 1 when $f$ is convex is $\min\_{t\in [0:T-1]}\mathbb{E}[f({\theta}_t) - f^\star]$ (Garrigos & Gower, 2024, Section 9.1), where $f^\star$ is the optimal value of minimizing a convex function $f$. The right column in Table 2 is the results under the convexity condition of $f$. An upper bound of $\min\_{t\in [0:T-1]}\mathbb{E}[f({\theta}_t) - f^\star]$ for the sequence $({\theta}_t)$ generated by Algorithm 1 can be obtained by using Lemma 2.1 that is the result when $f$ is not always convex (The proof of the results for convexity condition is given in Appendix A.4). While the performance measure $\min\_{t\in [0:T-1]}\mathbb{E}[\|\nabla f ({\theta}_t)\|]$ for the nonconvexity condition of $f$ differs from the measure $\min\_{t\in [0:T-1]}\mathbb{E}[f({\theta}_t) - f^\star]$ for the convexity condition of $f$,  Algorithm 1 under, for example, Case (ii) with the convexity of $f$ satisfies that $\min\_{t\in [0:T-1]}\mathbb{E}[f({\theta}_t) - f^\star] = O(\frac{1}{T})$.

---

> ### Author Response · Authors · 2024-12-27
> **Replies to Reviewer onDZ's comments**
>
> **Comment 3 (Proofs of Proposition A.1 and Lemma 2.1 Rely on Established Results):**
> The proofs appear to depend heavily on standard results from the literature on SGD analysis. Including more detailed citations for foundational work would enhance the paper's scholarly rigor and appropriately acknowledge prior contributions. Additionally, clearly specifying the original contributions in the analysis would help distinguish this work from existing approaches.
>
> **Reply:**
> Thank you for noting that including more detailed citations for foundational work would enhance the paper's scholarly rigor and appropriately acknowledge prior contributions. In response, we have added references to "Handbook of Convergence Theorems for (Stochastic) Gradient Methods (Garrigos & Gower, 2024)", which provides a comprehensive overview of convergence proofs for gradient and stochastic gradient methods in Section 2.2. Specifically, we added the following statements after Lemma 2.1:
>
> The proof of Lemma 2.1 depends on standard results from the literature on SGD analysis (Garrigos & Gower, 2024, Section 5.4, Theorem 5.12) using the descent lemma. Theorem 5.12 in (Garrigos & Gower, 2024) leads to the convergence of SGD with a constant learning rate $\eta$ that is proven by using the following inequality (Garrigos & Gower, 2024, (27)):
>
> \begin{align*}
> \min\_{t\in [0:T-1]} \mathbb{E} [\| \nabla f({\theta}_t) \|^2]
> \leq \frac{f({\theta}\_0) - \underline{f}^\star}{\eta \sum\_{t=0}^{T-1} \alpha\_t} + \eta \bar{L} L\_{\max} \Delta\_f^*,
> \end{align*}
>
> where $(\alpha_t)$ is defined by $\alpha_t (1  +\eta^2 \bar{L} L_{\max}) = \alpha_{t-1}$, $L_{\max} := \max_{i\in [n]} L_i$, and $\Delta_f^* := f^\star - \underline{f}^\star$.
>
> Meanwhile, Theorem 3.2 in this paper leads to the convergence of mini-batch SGD with a constant learning rate $\eta$ and an increasing batch size $b_t$ that is proven by using the following inequality in Lemma 2.1:
>
> \begin{align*}
> \min\_{t\in [0:T-1]} \mathbb{E} [\|\nabla f({\theta}_t) \|^2]
> \leq \frac{2(f({\theta}_0) - \underline{f}^\star)}{2 - \bar{L} \eta}
> \frac{1}{\eta T} + \frac{\bar{L} \eta \sigma^2}{2 - \bar{L} \eta} \frac{\sum\_{t=0}^{T-1} b_t^{-1} }{T}.
> \end{align*}
>
> While the existing approach (Garrigos & Gower, 2024) uses the sequence $(\alpha_t)$ satisfying $\sum\_{t=0}^{T-1} \alpha_t \geq \frac{1}{2 \eta^2 \bar{L} L_{\max}}$, the paper uses an increasing batch size $b_t$ satisfying $\sum\_{t=0}^{T-1} \frac{1}{b_t} < + \infty$. As a result, SGD with $\eta = \sqrt{\frac{2}{L L_{\max} T}}$ satisfies $\min\_{t\in [0:T-1]} \mathbb{E} [\|\nabla f({\theta}_t)\|] = O(\frac{1}{T^{\frac{1}{4}}})$ ) (Garrigos & Gower, 2024, Theorem 5.12), and mini-batch SGD with $\eta > 0$ and an the exponential growth batch size $b_t$ satisfies that $\min\_{t\in [0:T-1]} \mathbb{E} [\|\nabla f({\theta}_t)\|] = O(\frac{1}{\sqrt{T}})$ (Theorem 3.2 in this paper).
>
> Section 3.5 compares our results with the existing ones. Please see also Section 3.5 in the revised manuscript.
>
> We believe that these revisions address the reviewer’s concerns comprehensively and enhance the clarity, rigor, and scholarly value of our manuscript. We sincerely thank the reviewer for their constructive feedback, which has significantly improved the quality of our work.

---

> > ### Comment · Reviewer_onDZ · 2025-01-16
> >
> > I appreciate the authors’ responses and have no further comments.

---

> > > ### Author Response · Authors · 2025-01-16
> > >
> > > We would like to thank you for your careful reading of our manuscript and for the important and helpful comments and suggestions.

---

### Review · Reviewer_Z3gv · 2024-12-10

**Summary Of Contributions:**

The authors analyzed the convergence rate of mini-batch SGD to reach an approximate first-order stationary point. In particular, they considered four cases: (i) constant batch size and decaying learning rate scheduler, (ii) increasing batch size and decaying learning rate scheduler, (iii) increasing batch size and increasing learning rate scheduler, and (iv) increasing batch size and warm-up decaying learning rate scheduler. They first provided Lemma 2.1 which shows how the minimum norm of gradient along the trajectory of optimization can be bounded by the terms that depend on the batch size $b_t$ and the learning rate $\eta_t$. Based on this lemma, the rates of convergence for the four cases mentioned above are derived. The theoretical findings are evaluated in the experiment of training ResNet-18 on the CIFAR100 dataset.

**Audience:**

Yes

**Claims And Evidence:**

Yes

**Requested Changes:**

-	Please revise the table on page 2 and compare cases (ii) and (iii) in more detail.
-	It would be nice to provide intuition why increasing batch-size and learning rate can provide the convergence rate of $O(1/\sqrt{T})$. Moreover, is it sensitive how we set the initial learning rate (as the learning rate is increasing in time)? Finally, is there any work in training deep neural networks using this scheme?
-	To my understanding, Lemma 2.1 can be proved fairly straightforward. If not, it is good to mention what are the key challenges in proving Lemma 2.1. The main theorems are just based on applying this lemma to the four cases.
-	It is good to do experiments on complex tasks (such as training on ImageNet dataset) to see whether these trends hold there too.
-	I think it is good to pick the best method for each case and then put them in the same plot.

**Strengths And Weaknesses:**

Strength:

-	The paper provided the convergence rate of four different schemes in mini-batch SGD.

Weaknesses:

-	It is hard to follow the main results of the paper. In the table on page 2, for increasing batch-size and decreasing learning rate (case (ii)), the convergence rate is written as $O(1/\sqrt{T})$, $O(1/\sqrt{M})$. Do the authors mean $\max(O(1/\sqrt{T}), O(1/\sqrt{M}))$? The other issue is that the convergence rate for case (ii) is $ O(1/\sqrt{T})$ (except for diminishing LR). For the case (iii), the convergence rate is $O(1/\gamma^{M/2})$ where $1\leq \gamma^2 \leq 2$ and $M$ is the number of times that the batch-size increases. In the table, it is mentioned the $O(1/\gamma^{M/2})$ converges to $O(1/\sqrt{T})$ (which is not clear to me why). Even if it is the case, there is no improvement over case (ii) as there, we also have the convergence rate of $ O(1/\sqrt{T})$ (except for diminishing LR).
-	I think one of the main contributions of the current work is to validate the theoretical findings in training deep neural networks. However, as also mentioned by the authors, the experiments are very limited on a relatively simple dataset (CIFAR100). Therefore, it is not clear whether these trends can be observed in more complex tasks.

---

> ### Author Response · Authors · 2024-12-27
> **Replies to Reviewer Z3gv's comments**
>
> We appreciate your detailed assessments and helpful feedback. We have revised the manuscript to incorporate all of the recommendations, which has resulted in an improved presentation of our work. The revised parts of the manuscript are marked in red.
>
> Requested Changes:
>
> **Comment 1:**
> Please revise the table on page 2 and compare cases (ii) and (iii) in more detail.
>
> **Reply:**
> Thank you for your comment. Regarding Case (ii), the convergence rates $O(1 / \sqrt{T})$ and $O(1 / \sqrt{M})$ are both presented because they reflect the relationship between the total number of steps $T$ and the number of batch size increases $M$. As noted below the table, the total number of steps $T$ is given by $T(M) = \sum_{m=0}^M \lceil \frac{n}{b_m}\rceil E \geq M E$ which implies $T(M) \geq ME$. Hence, it is valid to express the convergence rate in terms of both $O(1 / \sqrt{T})$ and $O(1 / \sqrt{M})$. For Case (iii), the rate of convergence is expressed as $O(1 / \gamma^{M/2})$, which is formulated using $M$. To enable a direct comparison between Case (ii) and Case (iii), we include both $O(1 / \sqrt{T})$ and $O(1 / \sqrt{M})$ for Case (ii).
>
> Regarding the notation $O \left( \frac{1}{\gamma^{M/2}} \right) \to O \left( \frac{1}{\sqrt{T}} \right) $ in Case (iv), this does not signify convergence but rather reflects the transition from the rate of Case (iii) to the rate of Case (ii). Specifically, warm-up (Case (iv)) combines the increasing phase (Case (iii)) with the decay phase of Case (ii). To prevent misunderstanding, we have added the annotation of $O \left( \frac{1}{\gamma^{M/2}} \right) \to O \left( \frac{1}{\sqrt{T}} \right)$ for clarity as “$O \left( \frac{1}{\gamma^{M/2}}  \right) \to O \left( \frac{1}{\sqrt{T}} \right)$ implies that the convergence rate changes from $O \left( \frac{1}{\gamma^{M/2}}  \right)$ to $O \left( \frac{1}{\sqrt{T}} \right)$ when the learning rate $\eta_t$ changes from increasing learning rate to decaying learning rate.”.
>
> Let us compare cases (ii) and (iii). For simplicity, let us use a scheduler tripling increasing batch size, i.e., $b_t$ is multiplied by $\delta = 3$ per step and consider Case (ii) with a constant learning rate $\eta_t = \eta$ satisfying $\eta_{t+1} \leq \eta_t$ ($t\in \{0\} \cup \mathbb{N}$) and Case (iii) with the learning rate $\eta_t$ that is multiplied by $\gamma$ ($< \sqrt{\delta} = \sqrt{3}$) per step.
> $B_T$ in Case (ii) is such that
> \begin{align*}
>     B_T = \frac{1}{\sum_{t=0}^{T-1} \eta_t}
>     = \frac{1}{\sum_{t=0}^{T-1} \eta}
>     = O \left(\frac{1}{T} \right),
> \end{align*}
> while $B_T$ in Case (iii) is such that
> \begin{align*}
>     B_T = \frac{1}{\sum_{t=0}^{T-1} \eta_t}
>     = \frac{1}{\sum_{t=0}^{T-1} \gamma^t \eta_0}
>     = \frac{1}{\eta_0 \frac{\gamma^T - 1}{\gamma - 1}}
>     = O \left(\frac{1}{\gamma^T} \right).
> \end{align*}
> $V_T$ in Case (ii) is such that
> \begin{align*}
>     V_T = \frac{1}{\sum_{t=0}^{T-1} \eta_t} \sum_{t=0}^{T-1} \frac{\eta_t^2}{b_t}
>     = \frac{1}{\sum_{t=0}^{T-1} \eta} \sum_{t=0}^{T-1} \frac{\eta^2}{b_t}
>     = \frac{\eta}{T} \sum_{t=0}^{T-1} \frac{1}{\delta^t b_0}
>     \leq
>     \frac{\eta}{b_0 T} \frac{1}{1 - \frac{1}{\delta}}
>     = O \left(\frac{1}{T} \right).
> \end{align*}
> Meanwhile, $V_T$ in Case (iii) is such that
> \begin{align*}
>     V_T = \frac{1}{\sum_{t=0}^{T-1} \eta_t} \sum_{t=0}^{T-1} \frac{\eta_t^2}{b_t}
>     = \frac{1}{\sum_{t=0}^{T-1} \eta_t}
>     \sum_{t=0}^{T-1} \frac{\gamma^{2t} \eta_0^2}{\delta^t b_0}
>     = \frac{1}{\eta_0 \frac{\gamma^T - 1}{\gamma - 1}} \frac{\eta_0^2}{b_0} \sum_{t=0}^{T-1} \left(\frac{\gamma^2}{\delta} \right)^t
>     \leq
>     \frac{1}{\frac{\gamma^T - 1}{\gamma - 1}} \frac{\eta_0}{b_0} \frac{1}{1 - \frac{\gamma^2}{\delta}}
>     = O\left(\frac{1}{\gamma^T} \right),
> \end{align*}
> where $\gamma^2 < \delta$ is used to guarantee $\sum\_{t=0}^{+ \infty} (\frac{\gamma^2}{\delta} )^t < + \infty$.
> Therefore, Case (iii) has $\min\_{t\in [0:T-1]} \mathbb{E}[\|\nabla f({\theta}_t) \|] = O(\frac{1}{\gamma^{T/2}})$, which is better than Case (ii) with $\min\_{t\in [0:T-1]} \mathbb{E}[\|\nabla f({\theta}_t) \|] = O(\frac{1}{\sqrt{T}})$.

---

> ### Author Response · Authors · 2024-12-27
> **Replies to Reviewer Z3gv's comments**
>
> **Comment 2:**
> It would be nice to provide intuition why increasing batch-size and learning rate can provide the convergence rate of $O(1/ \sqrt{T})$. Moreover, is it sensitive how we set the initial learning rate (as the learning rate is increasing in time)? Finally, is there any work in training deep neural networks using this scheme?
>
> **Reply:**
> As mentioned above, the convergence rate of Case (ii) is $O(1 / \sqrt{M})$. Comparing this with Case (iii), where the rate is $O(1 / \gamma^{M/2})$, it becomes evident that Case (iii) achieves faster convergence. The intuition why increasing batch size and learning rate can provide the fast convergence rate $O(1/\gamma^{M/2})$ is the following:
> (1) Increasing batch sizes decreases the variance of stochastic gradient, since the upper bound of the variance is inversely proportional to the batch size (see also Proposition A.1 in the revised manuscript). Hence, increasing batch sizes leads to finding stationary points of the empirical loss $f$. This fact is based on Case (i) such that the upper bound of $\min\_{t \in [0:T-1]} \mathbb{E}[\| \nabla f({\theta}_t)  \|]$ is $O(\sqrt{ 1/T + 1/b })$, where $b$ is the batch size.
> (2) SGD does not work when learning rates are small at an early stage of training. This fact is supported by the numerical results in Figure 1 (Case (i)) indicating that using the decaying learning rate $\eta_t = \eta\_{\max}/\sqrt{t+1}$ does not train DNN. Hence,  keeping learning rates large implies that SGD works well.
> (3) From (1) and (2), increasing batch size and learning rate can provide fast convergence of SGD.
>
> Additionally, in Case (iv), the learning rate increases more rapidly over fewer epochs, leading to a steeper reduction in the full gradient norm of the empirical loss. However, continuing this rapid increase could violate the condition $\eta_t \in [\eta_{\min}, \eta_{\max}] \subset [0, \frac{2}{\bar{L}})$. Therefore, careful control of the upper bound on the learning rate is essential for maintaining stability. In practice, 10% of the total training epochs is used as the warm-up period (He et al., 2016; Goyal et al., 2018; Gotmare et al., 2019; He et al., 2019).
>
> Our results (Case (iv)) are numerically supported by the previously reported results (He et al., 2016; Goyal et al., 2018; Gotmare et al., 2019; He et al., 2019) indicating that using warm-up learning rate is useful for training deep neural networks, such as ResNets and Transformer networks (Vaswani et al., 2017).

---

> ### Author Response · Authors · 2024-12-27
> **Replies to Reviewer Z3gv's comments**
>
> **Comment 3:**
> To my understanding, Lemma 2.1 can be proved fairly straightforward. If not, it is good to mention what are the key challenges in proving Lemma 2.1. The main theorems are just based on applying this lemma to the four cases.
>
> **Reply:**
> Thank you for your valuable comment. The proof of Lemma 2.1 depends on standard results from the literature on SGD analysis (Garrigos & Gower, 2024, Section 5.4, Theorem 5.12) using the descent lemma. Theorem 5.12 in (Garrigos & Gower, 2024) leads to the convergence of SGD with a constant learning rate $\eta$ that is proven by using the following inequality (Garrigos & Gower, 2024, (27)):
>
> \begin{align*}
> \min\_{t\in [0:T-1]} \mathbb{E} [\|\nabla f({\theta}_t)\|^2]
> \leq \frac{f({\theta}_0) - \underline{f}^\star}{\eta \sum\_{t=0}^{T-1} \alpha\_t} + \eta \bar{L} L\_{\max} \Delta_f^*,
> \end{align*}
>
> where $(\alpha_t)$ is defined by $\alpha_t (1  +\eta^2 \bar{L} L_{\max}) = \alpha_{t-1}$, $L_{\max} := \max_{i\in [n]} L_i$, and $\Delta_f^* := f^\star - \underline{f}^\star$.
> Meanwhile, Theorem 3.2 in this paper leads to the convergence of mini-batch SGD with a constant learning rate $\eta$ and an increasing batch size $b_t$ that is proven by using the following inequality in Lemma 2.1:
>
> \begin{align*}
> \min\_{t\in [0:T-1]} \mathbb{E} [\|\nabla f({\theta}_t)\|^2]
> \leq
> \frac{2(f({\theta}_0) - \underline{f}^\star)}{2 - \bar{L} \eta}
> \frac{1}{\eta T} + \frac{\bar{L} \eta \sigma^2}{2 - \bar{L} \eta} \frac{\sum\_{t=0}^{T-1} b_t^{-1}}{T}.
> \end{align*}
>
> While the existing approach (Garrigos & Gower, 2024) uses the sequence $(\alpha_t)$ satisfying $\sum\_{t=0}^{T-1} \alpha_t \geq \frac{1}{2 \eta^2 \bar{L} L_{\max}}$, the paper uses an increasing batch size $b_t$ satisfying $\sum\_{t=0}^{T-1} \frac{1}{b_t} < + \infty$. As a result, SGD with $\eta = \sqrt{\frac{2}{L L_{\max} T}}$ satisfies $\min\_{t\in [0:T-1]} \mathbb{E} [\|\nabla f({\theta}_t)\|] = O(\frac{1}{T^{\frac{1}{4}}})$ (Garrigos & Gower, 2024, Theorem 5.12), and mini-batch SGD with $\eta > 0$ and an the exponential growth batch size $b_t$ satisfies that $\min\_{t\in [0:T-1]} \mathbb{E} [\|\nabla f({\theta}_t)\|] = O(\frac{1}{\sqrt{T}})$ (Theorem 3.2 in this paper).
>
> Please see also Section 3.5 in the revised manuscript for comparisons of our convergence rate results with existing ones.

---

> ### Author Response · Authors · 2024-12-27
> **Replies to Reviewer Z3gv's comments**
>
> **Comment 4:**
> It is good to do experiments on complex tasks (such as training on ImageNet dataset) to see whether these trends hold there too.
>
> **Reply:**
> Thank you for your valuable suggestion. While we have already included experiments on Tiny-ImageNet in Appendix A.10, we have now added experiments on ImageNet in Appendix A.5 using the scheduler from Case (iv), which was identified as the best-performing case in Figure 6 (as discussed in Requested Changes 5). These new experiments confirm that the warm-up scheduling in Case (iv) effectively accelerates convergence on ImageNet as well. Overall, Case (iv) demonstrates its ability to reduce the full gradient norm of the empirical loss most effectively, validating its superior performance even on large-scale tasks.
>
>
> **Comment 5:**
> I think it is good to pick the best method for each case and then put them in the same plot.
>
> **Reply:**
> We appreciate this suggestion and have added a new plot (Figure 6) that compares the best-performing configurations of Case (i) through Case (iv) in terms of the reduction of the full gradient norm of the empirical loss. This visualization clearly shows that Case (iv) outperforms the other cases, with the steepest and most consistent decline in the gradient norm.
>
> We believe that these revisions address the reviewer’s concerns comprehensively and enhance the clarity, rigor, and scholarly value of our manuscript. We sincerely thank the reviewer for their constructive feedback, which has significantly improved the quality of our work.

---

> > ### Comment · Reviewer_Z3gv · 2025-01-16
> >
> > I thank the authors for addressing my comments. I have no further comment.

---

> > > ### Author Response · Authors · 2025-01-16
> > >
> > > We would like to thank you for your careful reading of our manuscript and for the important and helpful comments and suggestions.

---

### Review · Reviewer_YADH · 2024-12-17

**Summary Of Contributions:**

The manuscript presents an analysis of the scheduler within mini-batch SGD, including both theoretical and empirical aspects. The authors propose a clear convergence bound for mini-batch SGD under the assumptions of Lipschitz continuity and bounded functions, as well as bounded variance. The theoretical results might be sound with super-martingale-like manipulations and the detailed empirical results show great efforts of authors.

**Audience:**

Yes

**Claims And Evidence:**

Yes

**Requested Changes:**

1. More discussion about the assumption should be made (like Lipschitz, lower bounded f and variance bounded...) How are they employed in others' work and how is it closed to the reality/empirical cases.
2. More comparisons should be made with other work theoretically. There should be lots of work providing convergence bounds for miniSGD and please state how to differentiate you and them
3. Maybe more proof sketch needs to be added to reflect some insights and technical contributions.
4. If possible, more numerical studies can be made with large datasets and networks to improve the value of the work in the training process   today.

**Strengths And Weaknesses:**

Pros:
1. Authors provide detailed and complex theoretical math manipulations for the convergence bound of the different learning schedulers,
bringing more theoretical insights for the choice of learning schedulers.
2. The complete empirical results are provided and analyzed.

Cons:
1. Assumptions need to be discussed more. I think the assumption is not weak in reality (or for the learning cases in the empirical cases).
2. The contribution in theoretical techniques should be re-evaluated. I can not clearly identify the novelty theoretically.
3. More comparisons in both theoretical and empirical should be made for better positioning the work in the optimization community,

---

> ### Author Response · Authors · 2024-12-27
> **Replies to Reviewer YADH's comments**
>
> We appreciate your detailed assessments and helpful feedback. We have revised the manuscript to incorporate all of the recommendations, which has resulted in an improved presentation of our work. The revised parts of the manuscript are marked in red.
>
> Requested Changes:
>
> **Comment 1:**
> More discussion about the assumption should be made (like Lipschitz, lower bounded f and variance bounded...) How are they employed in others' work and how is it closed to the reality/empirical cases.
>
> **Reply:**
> Thank you for your valuable comment. In the revision, we added explanations about Assumption 2.1(A1)-(A3) to Section 2.1.
>
> The $L_i$-smoothness of $f_i$ in Assumption (A1) is used to analyze mini-batch SGD (Assumption 4.3; Garrigos and Gower, 2024), since almost of the analyses of mini-batch SGD are based on the descent lemma (Lemma 5.7; Beck, 2017) that is satisfied under the smoothness of $f_i$. If $f_i^\star := \inf\_{{\theta} \in \mathbb{R}^d}  f_i({\theta})  = - \infty$ holds, then the loss function $f_i$ corresponding to the $i$-th labeled training data $({x}_i, {y}_i)$ does not have any global minimizer, which implies that the empirical loss $f$ satisfies $f^\star := \inf\_{{\theta} \in \mathbb{R}^d} f({\theta}) = - \infty$. Hence, the interpolation property (Section 4.3.1; Garrigos and Gower, 2024) (i.e., there exists ${\theta}^\star \in \mathbb{R}^d$ such that, for all $i\in [n]$, $f_i ({\theta}^\star) = f_i^\star \in \mathbb{R}$) does not hold, while the interpolation property holds for optimization of a linear model with the squared hinge loss for binary classification on linearly separable data (Section 2; Vaswani et al., 2019).
> Moreover, in the case where $f$ is convex with $f^\star = - \infty$, there is no stationary point of $f$, which implies that any algorithms never find stationary points of $f$. Accordingly, the condition $f_i^\star := \inf\_{{\theta} \in \mathbb{R}^d}  f_i({\theta})  \in \mathbb{R}$ in (A1) is natural under training DNNs including the case where the empirical loss $f$ is cross entropy with ${\theta}^\star \in \mathbb{R}^d$ such that  $f({\theta}^\star) = \inf\_{{\theta} \in \mathbb{R}^d} f({\theta}) = 0$. Assumption (A2) is satisfied when (A1) holds and the random variable $\xi$ follows uniform distribution that is used to train DNNs in practice (please see Appendix A.1 for details).
> Assumption (A3) holds under sampling with replacement (please see Appendix A.2 for details).

---

> ### Author Response · Authors · 2024-12-27
> **Replies to Reviewer YADH's comments**
>
> **Comment 2:**
> More comparisons should be made with other work theoretically. There should be lots of work providing convergence bounds for miniSGD and please state how to differentiate you and them.
>
> **Reply:**
> Thank you for your valuable comment. In the revision, we compared our results with the existing convergence analyses of SGD (Section 3.5 in the revised manuscript).
>
> Let us consider the case where a learning rate $\eta_t$ is constant, i.e., $\eta_t = \eta > 0$. Theorem 11 in (Scaman and Malherbe, 2020) indicated that SGD with $\eta = O(\frac{1}{\sqrt{\bar{L} T}})$ satisfies
> $\min\_{t\in [0:T-1]} \mathbb{E}[\|\nabla f ({\theta}_{t})\|] = O(\frac{1}{T^{1/4}})$.
>
> Corollary 1 in  (Khaled & Richtárik, 2023) showed that, under the weaker condition (the expected smoothness (Assumption 2; Khaled & Richtárik, 2023) than (A2), SGD with $\eta = O(\frac{1}{\sqrt{\bar{L} T}})$ satisfies $\min\_{t\in [0:T-1]} \mathbb{E}[\|\nabla f ({\theta}_{t})\|] = O(\frac{1}{\sqrt{T}})$.
>
> Meanwhile, Theorem 2 in the paper indicates that SGD with $\eta = O(\frac{1}{\bar{L}})$ and an increasing batch size $b_t$ satisfies $\min_{t\in [0:T-1]} \mathbb{E}[\|\nabla f ({\theta}_{t})\|] = O(\frac{1}{\sqrt{T}})$. For example, let us consider training DNN on CIFAR-100 dataset ($n = 50000$) during the number of epochs $E =200$. When the batch size $b_0$ is $2^5$, the number of steps per epoch is $K = \lceil \frac{n}{b_0}\rceil = 1563$. Hence, we have $T = KE = 312600$.
> Since the Lipschitz constant $\bar{L}$ of $\nabla f$ would be large, SGD with a small enough learning rate $\eta = O(\frac{1}{\sqrt{\bar{L} T}})$ would not work in practice.
> Meanwhile, since the learning rate $\eta = O(\frac{1}{\bar{L}})$ is constant without depending on $T$, SGD with $\eta = O(\frac{1}{\bar{L}})$ will work well.
>
> The previously reported results in  (Vaswani et al., 2019; Wang et al., 2021; Loizou et al., 2021) showed convergence of SGD with specialized learning rates, such as the Armijo line search learning rate, step decay learning rate, and stochastic Polyak learning rate. Our results indicate that SGD using practical learning rates, such as the cosine-annealing and polynomial decay learning rates, minimizes $\min_{t \in [0:T-1]} \mathbb{E}[\|\nabla f({\theta}_t)\|]$ in the sense of the rate of convergence $O(\frac{1}{\sqrt{T}})$ (Theorem 2). In addition, we would like to emphasize that, under using an increasing batch size, SGD with an increasing learning rate accelerates SGD with constant learning rate (Theorem 3). The acceleration of SGD is guaranteed during $\eta_t < \frac{2}{\bar{L}}$ and the convergence of SGD is not guaranteed during $\eta_t > \frac{2}{\bar{L}}$.
> Therefore, using warm-up constant learning rate (Case (iv)) is appropriate to guarantee fast convergence of SGD.
>
> The comparisons are summarized in Table 1 in the revised manuscript.

---

> ### Author Response · Authors · 2024-12-27
> **Replies to Reviewer YADH's comments**
>
> **Comment 3:**
> Maybe more proof sketch needs to be added to reflect some insights and technical contributions.
>
> **Reply:**
> Thank you again for your valuable comment. In the revision, we added the sketch of the proof of Lemma 2.1 that is essential to provide our results. Please see Page 3 in the revised manuscript.
>
> We also compared our theoretical techniques with the existing ones. The existing technique of SGD is based on the following standard inequality ((27); Garrigos and Gower, 2024) using the descent lemma:
>
> \begin{align*}
> \min\_{t\in [0:T-1]} \mathbb{E} \left[\|\nabla f({\theta}_t)\|^2 \right]
> \leq
> \frac{f({\theta}_0) - \underline{f}^\star}{\eta \sum\_{t=0}^{T-1} \alpha_t} + \eta \bar{L} L\_{\max} \Delta\_f^*,
> \end{align*}
>
> where $(\alpha_t)$ is defined by $\alpha_t (1  +\eta^2 \bar{L} L\_{\max}) = \alpha_{t-1}$, $L\_{\max} := \max_{i\in [n]} L_i$, $f^\star$ is the optimal value of $f$ over $\mathbb{R}^d$, and $\Delta_f^* := f^\star - \underline{f}^\star$.
> While the existing approach (Garrigos and Gower, 2024) uses the sequence $(\alpha_t)$ satisfying $\sum\_{t=0}^{T-1} \alpha_t \geq \frac{1}{2 \eta^2 \bar{L} L_{\max}}$, the paper uses a learning rate $\eta_t$ satisfying $\sum\_{t=0}^{T-1} \eta_t \geq O(T)$ and an increasing batch size $b_t$ satisfying $\sum\_{t=0}^{T-1} \frac{1}{b_t} < + \infty$ in Lemma 2.1. As a result, mini-batch SGD with $\eta > 0$ and the increasing batch size $b_t$ satisfies that
>
> $\min\_{t\in [0:T-1]} \mathbb{E} [\|\nabla f({\theta}_t)\|] = O(\frac{1}{\sqrt{T}})$ (Theorem 2),
>
> which is better than the existing result such that SGD with $\eta = \sqrt{\frac{2}{\bar{L} L\_{\max} T}}$ satisfies $\min_{t\in [0:T-1]} \mathbb{E} [\|\nabla f({\theta}_t)\|] = O(\frac{1}{T^{1/4}})$ (Theorem 5.12; Garrigos and Gower, 2024).
> Section 3.5 in the revised manuscript compares our results with the existing ones in details.
>
>
> **Comment 4:**
> If possible, more numerical studies can be made with large datasets and networks to improve the value of the work in the training process today.
>
> **Reply:**
> Thank you for your valuable suggestion. While we have already included experiments on Tiny-ImageNet in Appendix A.10, we have now added experiments on ImageNet in Appendix A.5 using the scheduler from Case (iv), which was identified as the best-performing case in Figure 6. Figure 6 is a newly created plot that compares the best-performing configurations of Case (i) through Case (iv) in terms of the reduction of the full gradient norm of the empirical loss. The results of the new ImageNet experiments confirm that the warm-up scheduling in Case (iv) effectively accelerates convergence on ImageNet as well. Overall, Case (iv) demonstrates its superior ability to reduce the full gradient norm of the empirical loss, validating its effectiveness even on large-scale tasks.
>
> We believe that these revisions address the reviewer’s concerns comprehensively and enhance the clarity, rigor, and scholarly value of our manuscript. We sincerely thank the reviewer for their constructive feedback, which has significantly improved the quality of our work.

---

> ### Author Response · Authors · 2025-01-23
>
> Dear Reviewer YADH,
>
> We appreciate your detailed assessments and helpful feedback. We replied to your comments on 27 Dec 2024. If you have any questions, please let us know.
>
> We look forward to your valuable comments.
>
> Sincerely yours,
>
> Paper3686 Authors

---

> > ### Comment · Reviewer_YADH · 2025-01-24
> >
> > Sorry for later replying. I appreciate the efforts for the clarfication of the assumptions and theoretical contri. I have no further comments and I think the authors address my concerns. Thanks!

---

> > > ### Author Response · Authors · 2025-01-24
> > >
> > > We would like to thank you for your careful reading of our manuscript and for the important and helpful comments and suggestions.

---

### Decision · Action_Editor_nzud · 2025-02-03

**Recommendation:** Accept with minor revision

**Comment:**

The authors have analyzed convergence of mini-batch SGD under four batch-size and step-size schedulers. They consider empirical risk minimization to optimize a smooth function with unbiased stochastic gradients considering mini batches where indices are drawn i.i.d., i.e., sampling with replacement.

Lemma 2.1 is the key result where an upper bound on the expected full gradient norm (the best iterate) is obtained for a general batch-size and step-size scheduler, and the following rates for four special cases are obtained using this lemma.

The reviewers had major concerns regarding theoretical contributions and comparison with current results. The reviewers’ concerns have been properly addressed after discussing with the authors, and all reviewers are positive with the revised version.The reviewers highlighted detailed empirical studies and providing theoretical insights for the choice hyperparameters' schedulers as strengths of this paper. I’d share the same view as the reviewers that the revised manuscript can be accepted after addressing these comments:

Schedule i) This is an upper bound and you show that the upper bound is larger than 0 no matter how large $T$ becomes. It does not necessarily imply that “mini-batch SGD does not converge to a stationary point.” BTW, even in other cases where the upper bound goes to zero as $T$ increases, you essentially show error bounds for the best iterate within $T$ iterates, not the last iterate. For the last iterate, none of the key results in Page 2 guarantee that the mini-batch SGD always converges to the stationary point.

When describing the second row of the table in Page 2  (Case (ii)), please clarify what $M$ is. Also it is not clear what you mean by $O(\frac{1}{\sqrt{T}}),\quad O(\frac{1}{\sqrt{M}})$

**Audience:**

Yes

**Claims And Evidence:**

Yes

---

> ### Author Response · Authors · 2025-02-12
> **Replies to Action Editor nzud's comments**
>
> We appreciate your valuable feedback and the opportunity to further clarify our manuscript. Below, we provide a detailed response to each of your comments and describe the corresponding modifications made to the original manuscript.
>
> 1. Clarification of Schedule (i) and the Interpretation of the Upper Bound
>
> We acknowledge the validity of your concern regarding the statement "mini-batch SGD does not always converge to a stationary point" in the description of Schedule (i). As you pointed out, our analysis establishes an upper bound on the expected full gradient norm of the best iterate within $T$ iterations.
> However, this does not necessarily imply a failure of convergence to a stationary point in the long run. Instead, this bound characterizes how the expected gradient norm scales with the batch size $b$ under the considered learning rate schedules.
> Importantly, this bound is derived for the best iterate over $T$ iterations and does not guarantee the convergence of the last iterate.
> To address this issue, we have revised the corresponding statement in the manuscript to avoid any misleading implications. Specifically, we have modified the description of Schedule (i) in the Introduction section as follows:
>
> Using a constant batch size and practical decaying learning rates, such as constant, cosine-annealing, and polynomial decay learning rates, satisfies that, for a sufficiently large step $T$, the upper bound on
> $\min_{t\in [0:T-1]} \mathbb{E} [\|\nabla f({\theta}_t)\| ]$ becomes approximately  $O(\frac{1}{\sqrt{b}}) > 0$.
> This result provides an upper bound on the best iterate within $T$ iterations and characterizes how the expected gradient norm scales with the batch size $b$ under the considered learning rate schedules. However, this bound does not necessarily mean that mini-batch SGD does not converge to a stationary point in the long run, as it does not characterize the last iterate.
> Meanwhile, the analysis indicates that using the cosine-annealing and polynomial decay learning rates would decrease $\mathbb{E}[\|\nabla f({\theta}_t)\|]$ faster than using a constant learning rate (see (8)), which is supported by the numerical results in Figure 1.

---

> ### Author Response · Authors · 2025-02-12
> **Replies to Action Editor nzud's comments**
>
> 2. Clarification of $M$ and the Two Convergence Rates in Case (ii)
>
> In Case (ii), we analyze mini-batch SGD with increasing batch sizes and decaying learning rates. To formalize this setting, we define $M$ as the number of times the batch size is increased during training, which allows us to express the convergence rate in terms of $M$ rather than the total number of optimization steps $T$. This provides a more intuitive framework for understanding the impact of batch size scaling on convergence.
> The total number of training steps when the batch size increases $M$ times satisfies $T(M) = \sum_{m=0}^M \lceil \frac{n}{b_m}\rceil E \geq M E$.
> Since $T(M)$ is at least proportional to $ME$, the convergence rates can be analyzed in terms of either $T$ or $M$, giving $O(\frac{1}{\sqrt{T}})$ or $O(\frac{1}{\sqrt{M}})$, respectively.
> This reformulation in terms of $M$ instead of $T$ also enables a direct comparison with Case (iii), where geometric batch size scheduling achieves a convergence rate of $O(\gamma^{-\frac{M}{2}})$.
> By expressing the convergence rate in terms of $M$, we establish a unified framework that facilitates comparison of different batch size scheduling strategies in mini-batch SGD.
> To enhance clarity and explicitly connect the two convergence rates, we have revised the explanation of Case (ii) in the Introduction section as follows:
>
> Although convergence analyses of SGD
> were presented in  (Vaswani et al., 2019; Fehrman et al., 2020; Scaman & Malherbe, 2020; Loizou et al., 2021; Wang et al., 2021; Khaled & Richtárik, 2023),
> providing the theoretical performance of mini-batch SGD with increasing batch sizes that have been used in practice may not be sufficient.
> The present paper shows that mini-batch SGD has an $O(\frac{1}{\sqrt{T}})$ rate of convergence,
> where $T$ represents the total number of optimization steps, and $M$ represents the number of times the batch size is increased during training.
> Since the total number of training steps when batch size increases $M$ times satisfies $T(M) = \sum_{m=0}^M \lceil \frac{n}{b_m}\rceil E \geq M E$, the convergence rates can be analyzed either in terms of $T$ or $M$, giving $O(\frac{1}{\sqrt{T}})$ or $O(\frac{1}{\sqrt{M}})$, respectively.
> This result highlights that increasing batch sizes affects the variance reduction in mini-batch SGD.
> By analyzing the convergence rate in terms of $M$ instead of $T$, we are able to make a more direct comparison with Case (iii), where the geometric batch size scheduling achieves an $O(\frac{1}{\gamma^{\frac{M}{2}}})$ convergence rate.
> Increasing batch size every $E$ epochs makes the polynomial decay and linear learning rates
> become small at an early stage of training Figure 2(a).
> Meanwhile, the cosine-annealing and constant learning rates are robust to  increasing batch sizes (Figure 2(a)).
> Hence, it is desirable for mini-batch SGD using increasing batch sizes to use the cosine-annealing and constant learning rates, which is supported by the numerical results in Figure 2.

---

> > ### Comment · Action_Editor_nzud · 2025-02-12
> > **Camera Ready Verification**
> >
> > Dear Authors,
> >
> > Thanks for clarifying and addressing the comments within the camera ready version. I am pleased to verify the camera ready version!
> >
> > Regards,
> >
> > Action Editor